# Chemical characterization of long-range transport biomass burning emissions to the Himalayas: insights from high-resolution aerosol mass spectrometry

**Xinghua Zhang[1,2,3], Jianzhong Xu[1], Shichang Kang[1], Yanmei Liu[1,3], Qi Zhang[4]**

[1]State Key Laboratory of Cryospheric Sciences, Northwest Institute of Eco-Environment and Resources, Chinese Academy of Sciences, Lanzhou 730000, China

[2]Key Laboratory of Arid Climatic Change and Reducing Disaster of Gansu Province, Key Laboratory of Arid Climatic Change and Disaster Reduction of CMA, Institute of Arid Meteorology, China Meteorological Administration, Lanzhou 730020, China

[3]University of Chinese Academy of Sciences, Beijing 100049, China

[4]Department of Environmental Toxicology, University of California, Davis, CA 95616, USA

*Correspondence to*: Jianzhong Xu (jzxu@lzb.ac.cn)

## Abstract

An intensive field measurement was conducted at a remote, background, and high-altitude site (Qomolangma station, QOMS, 4276 m a.s.l.) in the northern Himalayas, using an Aerodyne high-resolution time-of-flight aerosol mass spectrometer (HR-ToF-AMS) along with other collocated instruments. The field measurement was performed from April 12 to May 12, 2016 to chemically characterize the high time-resolved submicron particulate matter ($PM_1$) and obtain the dynamic processes (emissions, transport, and chemical evolution) of biomass burning (BB), frequently transported from South Asia to the Himalayas during pre-monsoon season. Overall, the average ($\pm$ $1\sigma$) $PM_1$ mass concentration was 4.44 ($\pm 4.54$) $\mu g\ m^{-3}$ for the entire study, comparable with those observed at other remote sites worldwide. Organic aerosol (OA) was the dominant $PM_1$ species (accounting for 54.3% of total $PM_1$ on average) followed by black carbon (BC) (25.0%), sulfate (9.3%), ammonium (5.8%), nitrate (5.1%), and chloride (0.4%). The average size distributions of $PM_1$ species all peaked at an overlapping accumulation mode ($\sim 500$ nm), suggesting that aerosol particles were internally well-mixed and aged during long-range transport. Positive matrix factorization (PMF) analysis on the high-resolution organic mass spectra identified three distinct OA factors, including a BB-related OA (BBOA, 43.7%), a nitrogen-containing OA (NOA, 13.9%) and a more-oxidized oxygenated OA (MO-OOA, 42.4%). Two polluted episodes with enhanced $PM_1$ mass loadings and elevated BBOA contributions from the west and southwest of QOMS during the study were observed. A typical BB plume was investigated in detail to illustrate the chemical evolution of aerosol characteristics under distinct air mass origins, meteorological conditions and atmospheric oxidation processes.

## 1 Introduction

The Tibetan Plateau and Himalayas (TPH), generally called the "third pole", is the highest (average altitude of more than 4000 m a.s.l.) and largest ($\sim 2\ 500\ 000\ km^2$) plateau in the world. This region has been recognized as one of the most pristine region in the world due to its high altitude, sparse population and minor influence of anthropogenic activities (Yao et al., 2012a).

Consideration on the intense dynamical and thermal forcing effects, the TPH not only plays a key
role in the formation of Asian monsoon systems, but also impacts the large-scale atmospheric
circulation, hydrological cycle, as well as global climate (Duan and Wu, 2005; Wu et al., 2007).
Over the past decades, more attentions have been paid to the environment and climate change in
the TPH since this region is very susceptibility to the global climate change such as fast air
temperature rise and dramatic glacier shrinkage (Xu et al., 2009; Kang et al., 2010; Yao et al.,
2012b; Yang et al., 2014). Atmospheric environment in the TPH, albeit which is one of the most
pristine region in the world, has been thought to be influenced variably due to the worse air
pollution in its surrounding countries (Hou et al., 2003; Lau et al., 2008). For example, polluted air
mass, particularly from South and Southeast Asia regions, had been observed frequently to
transport to the Himalayas (Bonasoni et al., 2010; Cong et al., 2015), heat the aloft air masses over
the TPH (Lau et al., 2006; Ramanathan and Carmichael, 2008) and decline the surface albedo after
its deposition onto snow and glacier (Xu et al., 2009). As a consequence, characterizing the
aerosol physicochemical properties in the TPH, including mass loading, chemical composition,
size distribution and source, are of great importance to better understand the aerosol chemistry,
estimate the aerosol radiative forcing, and finally evaluate the effect of polluted air mass on the
ecology and environment in the TPH region.
Numerous aerosol measurements have been conducted in the TPH region in past decades to
characterize the physicochemical properties, sources and transport pathways of ambient aerosol
(Liu et al., 2008; Decesari et al., 2010; Marcq et al., 2010; Marinoni et al., 2013; Putero et al.,
2014; Xu et al., 2017; Zhang et al., 2017a). South and Southeast Asia are two major polluted
regions due to their intense biomass burning (BB) activities from natural forest fires and
traditional human burning activities for residential heating and cooking (Engling et al., 2011;
Yadav et al., 2017). The polluted feature of South and Southeast Asia during April 12 to May 12,
2016 can be further revealed by the distribution of average aerosol optical depth in Fig. 1. During
the pre-monsoon period, atmospheric pollutants associated with BB emissions in South Asia are
generally advected by regional and long-range transport (e.g., westerlies and South Asian
monsoon system) to Himalayas and built up in the southern foothills, then pollutants are lifted up
to high altitude by the Himalayan topography and the typical valley wind circulation (Zhao et al.,
2013; Cong et al., 2015; Liu et al., 2017). However, the chemical properties of aerosol particles
are still not well understood and limited in the Himalayas region due to its remote and harsh
environments, challenging weather conditions and logistic difficulties. In addition, most of the
available studies are mainly based on the off-line filter sampling of ambient aerosol or snow/ice
samples following by laboratory analyses (Decesari et al., 2010; Ram et al., 2010; Li et al., 2016;
Wan et al., 2017). These studies usually had a relatively low-time resolution (days to weeks).
Therefore, real-time consecutive field measurement, especially focusing on the high-resolution
size-resolved chemical characteristics of aerosol particles, is of great importance and necessary to
give insight into the sources and the dynamic chemical evolution of ambient aerosol.
Online real-time instrument such as Aerodyne aerosol mass spectrometer (AMS), which can
be used to characterize the chemical properties and sources of submicron aerosol particles with
high time resolution and sensitivity, has been greatly developed and widely implemented
worldwide (Canagaratna et al., 2007; Zhang et al., 2007a; Jimenez et al., 2009; Li et al., 2017).
Although the deployments of the AMS in China have started since 2006, most of these studies in
China are conducted in urban areas, including Beijing−Tianjin−Hebei (Sun et al., 2013; Sun et al.,
2016), Yangtze River Delta (Wang et al., 2016a; Wang et al., 2016b), Pearl River Delta regions
(Huang et al., 2011), and Lanzhou (Xu et al., 2014; Xu et al., 2016; Zhang et al., 2017b) as shown
in Fig. S1, whereas just few studies deployed in remote sites so far, such as Menyuan (Du et al.,
2015), Mt. Yulong (Zheng et al., 2017), and Nam Co (Xu et al., 2017; Wang et al., 2017). In this
paper, an Aerodyne high-resolution time-of-flight mass spectrometer (HR-ToF-AMS) was
deployed at the Qomolangma Station for Atmospheric and Environmental Observation and
Research (QOMS) in the north slope of the Himalayas to fill the vacancy of real-time mass
spectrometer measurement at high elevation site and evaluate the significant impacts of BBs from
polluted areas in the South Asia on the TPH aerosol properties during the pre-monsoon season.
Here, we report an overview of the 5-min real-time chemical and physical characteristics of
submicron aerosols (PM$_1$), including mass loading, composition, size distribution, acidity as well
as temporal and diurnal variations. The sources of organic aerosols (OA) are also investigated
using positive matrix factorization analysis on the high-resolution OA mass spectrum. BB
influence and chemical evolution of aerosols in polluted plume are examined via combining back
trajectory analysis of air masses and fire hotspots information, respectively.

## 2 Experimental methods

### 2.1 Sampling site

The QOMS (28.36 °N, 86.95 °E, 4276 m a.s.l.; Fig. 1), which is located in the northern slope of
Mt. Everest (~ 30 km away), was established for atmospheric and environmental observation since
2005 (Ma et al., 2008). The geomorphic and climate features around the QOMS are typical alpine
cold and arid areas covered by sandy soil with sparse vegetation. The QOMS is located in a long
river valley and isolated from residential areas due to its harsh environment with a small village
(with a population of ~ 300) to the south (~ 10 km). The closest town, Dingri County, is ~ 100 km
south from the QOMS. A freeway is located at the front of the QOMS for tourism with increased
tourist during summer. The measurements were conducted from April 12 to May 12, 2016. Since
this period was within the typical pre-monsoon season of the TPH, the large-scale atmospheric
circulation pattern was dominated by westerly or southwesterly winds with limited precipitation.
Owning to a distinct thermal forcing from the southern mountains and glaciers, the QOMS was
locally dominated by strongly mountain-valley circulation with down-slope wind prevailing
during the daytime, especially in the afternoon (Fig. 1c and S2) (Zou et al., 2008), which would
make the valley as an efficient channel for the down transport of air mass form high-altitude
troposphere.

### 2.2 Instrumentation

A suite of real-time instruments were co-located to measure the physiochemical properties of fine
particles at the QOMS, including an Aerodyne HR-ToF-AMS (Aerodyne Research Inc., Billerica,
MA, USA) for 5-min size-resolved chemical compositions (organics, sulfate, nitrate, ammonium,
and chloride) of non-refractory submicron particulate matter (NR-PM$_1$), a scanning mobility
particle sizer (SMPS, model 3936, TSI Inc., Shoreview, MN, USA) for 5-min particle number
concentration and size distribution between 14.6 and 661.2 nm in mobility diameter ($D_m$), and a
photoacoustic extinctiometer (PAX, DMT Inc., Boulder, CO, USA) for particle light absorption
and scattering coefficient ($b_{abs}$ and $b_{scat}$) at 405 nm and further deriving black carbon (BC) mass
concentration. All instruments were placed in an air-conditioned room with temperature
maintaining at ~ 20 ℃. Ambient aerosol particles were introduced through a 0.5 inch copper tube
which stemmed out of the rooftop by about 1.5 m. A PM$_{2.5}$ cyclone (model URG-2000-30EH,
URG Corp., Chapel Hill, NC, USA) was used in front of the sampling inlet for removing coarse
particles with size cutoffs of 2.5 μm in aerodynamic diameter ($D_{va}$). A diffusion dryer was placed
following the cyclone to dry the ambient air and eliminate potential humidity effect on particles.
The total length of the sampling line was about 5 m and the retention time of particles was less
than 2.5 s in the whole inlet. The total air flow rate from the sampling inlet was about 10 L min$^{-1}$,
with part of flow shared by the HR-ToF-AMS and the SMPS while the remaining flow exhausted
by an external pump. The meteorology data including wind speed (WS), wind direction (WD),
relative humidity (RH), temperature ($T$), and solar radiation (SR) during this study were obtained
from a Vantage Pro2 weather station (Davis Instruments Corp., Hayward, CA, USA). Note that all
the date and time used in this study are reported in Beijing Time (BJT: UTC + 8 h).
**2.3 HR-ToF-AMS operation and data analysis**
**2.3.1 HR-ToF-AMS operation**
A detailed instrumental description of the Aerodyne HR-ToF-AMS can be found elsewhere
(DeCarlo et al., 2006) and only a brief summary is provided here. Briefly, the HR-ToF-AMS
consists of three main parts: an aerosol sampling inlet, a particle sizing vacuum chamber, and a
particle composition detection section (Jimenez et al., 2003). Ambient particles are sampled into
the instrument through a critical orifice (130 μm in this study for enhancing the transmission
efficiency at the high-altitude area) and focus into a concentrated and narrow beam through an
aerodynamic lens. Then particles are accelerated into the sizing vacuum chamber and obtain
different velocities for particles with different sizes due to the supersonic expansion induced by
different pressure between the two chambers. Meanwhile, a mechanical chopper with two radial
slits located 180 ° apart is used to intercept the focused particle, and then the time of flight (P-ToF)
from the chopper to the vaporizer is measured to obtain the aerodynamic size of particles. After
passing through the sizing chamber, particles are directed onto a resistively heated surface (~
600 ℃) under a high vacuum and ionized by a 70 eV electron impact, and finally detect by the
high-resolution time-of-flight mass spectrometer. In this study, the HR-ToF-AMS was only
toggled under the high sensitive V-mode (detection limits ~ 10 ng m$^{-3}$). Under the V-mode
operation, the instrument also switched between the mass spectrum (MS) mode and the particle P-
ToF mode every 15 s, spending 6 and 9 s on each, to obtain the mass concentrations and size
distributions of the non-refractory species, respectively.
The HR-ToF-AMS was calibrated for ionization efficiency (IE) and particle sizing at the
beginning, in the middle, and at the end of this study according to the standard protocols (Jayne et
al., 2000). Both the calibrations of IE and particle sizing were performed using mono-dispersed
ammonium nitrate particles with nominal diameters of 70–300 nm. Default relative ionization
efficiency (RIE) values were assumed in this study as 1.1 for nitrate, 1.3 for chloride, and 1.4 for
organics. The RIE values of 3.9 and 4.2 were used for ammonium based on the results of two IE
calibrations at the beginning and in the middle of this study, while RIE values of 1.6 and 1.4 were
determined similarly for sulfate by using mono-dispersed ammonium sulfate particles,
respectively.

### 2.3.2 HR-ToF-AMS data analysis

The mass concentrations and size distributions of NR-PM$_1$ species and the ion-speciated mass
spectra, composition and elemental composition of organics were determined from the HR-ToF-
AMS data by using the standard ToF-AMS analysis toolkit SQUIRREL (v1.56) and PIKA (v1.15c)
modules written in Igor Pro (Wavemetrics Inc., Lake Oswego, OR, USA). An empirical particle
collection efficiency (CE) of 0.5 was used to compensate for the incomplete transmission and
detection of particles due to particle bouncing at the vaporizer and partial transmission through the
aerodynamic lens, which has been widely used in field studies employing AMS with a dryer
installed in front of the inlet (Xu et al., 2014; Xu et al., 2016). The elemental ratios of oxygen-to-
carbon (O/C), hydrogen-to-carbon (H/C), nitrogen-to-carbon (N/C), and organic mass-to-organic
carbon (OM/OC) for this study were determined using the "improved-ambient" method (referred
as I-A method) (Canagaratna et al., 2015), which increased O/C on average by 34%, H/C on
average by 15%, and OM/OC on average by 17% (Fig. S3) compared with those determined from
the "Aiken ambient" method (referred as A-A method) (Aiken et al., 2008).
Positive matrix factorization (PMF) analysis using the PMF2.exe algorithm (v4.2) (Paatero
and Tapper, 1994) in robust mode was conducted on the high resolution mass spectra (HRMS) to
determine distinct OA components in this study. The analysis was performed using an Igor Pro-
based PMF Evaluation Tool (PET, v2.03) (Ulbrich et al., 2009), downloaded from the webpage
(http://cires.colorado.edu/jimenez-group/wiki/index.php/PMF-AMS_Analysis_Guide). The data
and error matrices input into the PMF analysis were generated from analyzing the V-mode data via
PIKA fitting. Detailed PMF analysis was thoroughly evaluated following the procedures
summarized in Table 1 of Zhang et al. (2011). Isotopic ions were generally excluded and the four
ions of O$^+$, HO$^+$, H$_2$O$^+$, and CO$^+$ were downweighted in PMF analysis, because they were
determined according to the relationship with CO$_2^+$ signal (Ulbrich et al., 2009). The "bad" ions
with $S/N$ less than 0.2 were removed from the HRMS data and error matrices before PMF analysis,
and "weak" ions with $S/N$ between 0.2 and 2 were downweighted by increasing their errors. In
addition, some runs with huge mass loading spikes were also removed from the data and error
matrices. The detailed matrix preparation and data pretreatment can also refer to Xu et al. (2014).
A summary of key diagnostic plots of the PMF results for this study is presented in Fig. S4.
Overall, the PMF solutions were investigated for 1 to 8 factors and for the rotational parameter
(fPeak) varying from −1 to 1 with a step of 0.1. Besides examining the model residuals, scaled
residuals, and the Q/Q$_{exp}$ contributions for each $m/z$ and time following procedures detailed in
Table 1 of Zhang et al. (2011), the optimum solution can also be evaluated via comparing the mass
spectra of individual factors with reference spectra from specific sources or other ambient AMS
measurements, comparing the time series of individual factors with the known external tracers,
and analyzing the diurnal variations of individual factors. Finally, the 3-factor solution with fPeak
= 0 was chosen in this work. The direct comparisons of the mass spectra, time series, and diurnal
variations for 2-factor and 4-factor solution were also shown in Fig. S5 and S6, respectively. The
2-factor solution does not resolve the small, yet distinct nitrogen-containing OA, while the 4-
factor solution shows a splitting factor from the BB OA resolved in the 3-factor solution and
seems just like a simple separation of the two BB polluted episodes.

## 2.4 Other relevant data

The Hybrid Single Particle Lagrangian Integrated Trajectory (HYSPLIT4) model developed by the National Oceanic and Atmospheric Administration (NOAA) (Draxler and Rolph, 2003) was used to investigate the origins of air masses in this study, using the meteorological data from the NOAA Global Data Assimilation System (GDAS). The back trajectories were calculated every 6 h at an ending height of 500 m above ground level at the QOMS during the entire campaign, and then clustered them according to their similarity in spatial distribution. Finally, a four-cluster solution was adopted according to its small total spatial variance.

Aerosol optical depth (AOD) at 550 nm was derived from the observations made by National Aeronautics and Space Administration (NASA) Moderate Resolution Imaging Spectroradiometer (MODIS) onboard the Terra satellite. The distribution of average aerosol optical depth (AOD) in a large range areas (20 °–45 °N, 60 °–110 °E) around the TPH during the entire period of this study is given in Fig. 1d.

Various active fire hotspots were detected over South and Southeast Asia by the Fire Information for Resource Management System (FIRMS) provided by MODIS satellite (https://firms.modaps.eosdis.nasa.gov), demonstrating the possibility that active wildfires or BBs from South and Southeast Asia may have significant impacts on the air conditions in the TPH region.

The aerosol liquid water content (ALWC) was estimated with the Extended AIM (E-AIM) Aerosol Thermodynamics Model (http://www.aim.env.uea.ac.uk/aim/aim.php). The input data included the concentrations of sulfate, nitrate, ammonium, and chloride measured by the HR-ToF-AMS as well as the relative humidity (RH) and temperature of ambient air.

## 3 Results and discussion

## 3.1 Overview of the study

### 3.1.1 Meteorological conditions

The measurement period in our study was within the typical pre-monsoon season of the TPH. The meteorological conditions were therefore characterized by a relatively cold, dry and windy weather, and the westerlies dominated the large-scale atmospheric circulation patterns with little precipitation, as displayed in Fig. 2. During the study, the averaged diurnal air temperature ranged from −2.0 to 12.5 ℃ with an average ($\pm 1\sigma$) of 5.7 ($\pm 5.0$) ℃, and the RH ranged from 15.3 to 67.5% with an average of 39.8 $\pm$ 18.8%. Only two light precipitation events (1 and 0.5 mm d$^{-1}$) occurred on 1 and 8 May, respectively. The WDs at QOMS were predominantly by southwesterly, which were mainly associated with the thermally driven mountain-valley winds and glacier winds (Zou et al., 2008). For the diurnal variation of wind conditions, a nearly calm wind period (hourly average WS less than 2 m s$^{-1}$) was observed in the early morning time; after sunrise to noon time, there was a weak up-slope wind period (from the north); the diurnal wind cycles in the rest time were dominated by the down-slope wind (from the southwest) with the maximum value of hourly average WS up to 7 m s$^{-1}$ (Fig. 2b and S2).

### 3.1.2 Inter-comparisons between different instruments

An inter-comparison of the total $PM_1$ (NR-$PM_1$ + BC) mass concentrations measured by the HR-
ToF-AMS (CE = 0.5) and the PAX with particle volumes (assuming spherical particles)
determined from the SMPS is shown in Fig. S7. Overall, the $PM_1$ mass is closely correlated ($R^2$ =
0.97) with that of SMPS particle volume during the entire campaign, with a linear regression slope
of 2.86. This slope is significantly higher than the estimated average $PM_1$ density of 1.44 g $cm^{-3}$,
which is calculated based on the measured particle compositions in this study and the assumed
particle densities of 1.2 for organics, 1.78 for $(NH_4)_2SO_4$, 1.72 for $NH_4NO_3$, 1.52 for $NH_4Cl$ and
1.8 g $cm^{-3}$ for BC (Zhang et al., 2005b; Xu et al., 2016). This discrepancy is likely introduced by
various factors, including different transmission sizes between HR-ToF-AMS and SMPS (up to ~
1.0 μm in $D_{va}$ for AMS vs. limited size range of 14.6−661.2 nm in $D_m$ for SMPS), rough
calculation of $PM_1$ density using assumed composition densities and spherical shape without
consideration the particle porosity, as well as the using of empirical and constant CE value of 0.5
in this study. This phenomenon was also observed at other sites in previous studies (Ge et al., 2012;
Huang et al., 2012; Xu et al., 2014; Du et al., 2015).
**3.1.3 Mass concentration and chemical composition of $PM_1$**
As shown in Fig. 2, the mass concentrations of $PM_1$ and all $PM_1$ species, as well as their mass
fractions in $PM_1$ varied dynamically throughout this study. Two polluted periods (PP1 and PP2)
were identified according to their high $PM_1$ mass concentrations (daily average $PM_1$ mass is larger
than 5 μg $m^{-3}$), high contributions from BBOA and unique back trajectories. The rest periods
characterized by low $PM_1$ mass concentrations were considered as clear periods (CP1 and CP2).
The 5-min total $PM_1$ mass concentration ranged from 0.18 to 27.97 μg $m^{-3}$ for the study, with an
average ($\pm 1\sigma$) value of 4.44 $\pm 4.54$ μg $m^{-3}$. This average value was more than two times lower
than most of the $PM_1$ mass concentrations measured with Aerodyne AMS or aerosol chemical
speciation monitor (ACSM) instruments at various urban, suburban, rural or background sites in
China (10.9−138.8 μg $m^{-3}$) (Fig. S1), except slightly lower than that at Mt. Yulong (5.7 μg $m^{-3}$)
located at the southeastern edge of the TPH, whereas higher than that at Nam Co Station (2.0 μg
$m^{-3}$) located in the central of the TPH. Moreover, as shown in table S1, the $PM_1$ mass
concentration in this study was also lower than those measured at the three remote island sites in
Asia which were frequently influenced by outflow from China, Korea and Japan (i.e., 7.9 μg $m^{-3}$
for Okinawa island, 12.0 μg $m^{-3}$ for Fukue island in Japan, and 10.7 μg $m^{-3}$ for Jeju island in
Korea) (Takami et al., 2005; Jimenez et al., 2009), as well as the $PM_1$ mass concentration (15.1 μg
$m^{-3}$) obtained at the Bachelor mountain in United States which was heavily impacted by wildfire
smoke plumes (Zhou et al., 2017). However, it was higher than those reported at other coastal,
high elevation, forest or remote background sites in North America and Europe (0.55−2.91 μg $m^{-3}$)
(Zhang et al., 2007a; Sun et al., 2009; Fröhlich et al., 2015). Although these measurements
mentioned above were conducted at various sites worldwide during different seasons, these
comparisons further demonstrate that QOMS is a typical high elevation and remote background
site in Asia.
Overall, organics and BC were the two dominant $PM_1$ species (averagely contributed 54.3%
and 25.0% to the total $PM_1$ mass, respectively) followed by sulfate (9.3%), ammonium (5.8%),
nitrate (5.1%), and chloride (0.4%) (Fig. 3a). The high contributions of organics and BC at QOMS
were significantly associated with the active BB emissions by long-range transport from polluted
areas in South Asia. Organic compounds and BC have been revealed as two dominant components
of BB aerosols and generally used to identify BB events in previous studies (Bond et al., 2004;
Bougiatioti et al., 2014). In addition, biomass burning at high elevation regions of Himalayas and
south Asia was more incomplete burning and could emit amount of BC. This conclusion can be
further revealed by their enhanced mass concentrations and contributions, especially for organics,
during the two distinct polluted episodes influenced by active BB plumes. Figure 3b showed the
mass contributions of $PM_1$ species as a function of total $PM_1$ mass concentrations. The $PM_1$ mass
loadings in this study were mostly below 6 μg m$^{-3}$ (accounted for ~ 77%); The mass contribution
of organics increased significantly with the increase of total $PM_1$ mass loading whereas the rest
species showed relatively stable or decrease trends, suggesting the dominant contributions of
organics in the polluted episodes at QOMS.

**3.1.4 Acidity and size distributions of submicron aerosols**

To evaluate the bulk acidity of NR-$PM_1$ in this study, we calculated the $NH_4^+$ concentration
($NH_{4\,calc}^+$) based on the mass concentrations of sulfate, nitrate and chloride measured by the HR-
ToF-AMS and assumed full neutralization of these anions by ammonium (Zhang et al., 2007b).
The scatter plot of the measured $NH_4^+$ ($NH_{4\,meas}^+$) concentration versus the $NH_{4\,calc}^+$ concentration
for the entire campaign was shown in Fig. S7. A tight correlation ($R^2 = 0.97$) existed between
$NH_{4\,meas}^+$ and $NH_{4\,calc}^+$ with a linear regression slope of 1.2, indicating that there were excess of
ammonium in the submicron particle. This slightly high $NH_{4\,meas}^+/NH_{4\,calc}^+$ ratio was quite different
with those results from various urban and rural sites in China, where bulk aerosols were overall
neutralized or acidic due to the enrich gaseous precursors of $SO_2$ and $NO_x$ that could be further
oxidized to sulfate and nitrate (Sun et al., 2013; Xu et al., 2014; Du et al., 2015; Zhang et al.,
2017b). The excess ammonium at QOMS might relate to the important contributions of organic
acids in this area (Cong et al., 2015), which could underestimate the $NH_{4\,calc}^+$ due to the neglect of
organic acids in the ion-balance calculation, and the non-negligible contributions of nitrogen-
containing organic compounds to $NH_x^+$ which finally overestimated the $NH_{4\,meas}^+$ (Sun et al., 2009;
Ge et al., 2012). As mentioned above, atmospheric aerosols in the TPH region were significantly
influenced by BB emissions from South Asia during the sampling periods. BBs would emit large
amounts of nitrogen-containing organic compounds (Fleming et al., 2017; Zhou et al., 2017) and
as discussed in section 3.2.
Figure 4 shows the average size distributions of NR-$PM_1$ species and their mass contributions
as the function of size distribution. Overall, all chemical species showed a nearly consistent but
narrow accumulation mode peaking at ~ 500 nm in $D_{va}$, indicating the well internal-mixed and
aged aerosol particles at QOMS. Ultrafine particles (particles with diameter less than 100 nm)
were dominated by organics (more than 70%), while the mass contributions of chemical species at
the major peak (~ 500 nm) were organics (~ 65%), sulfate (~ 13%), nitrate (~ 11%), ammonium
(~ 10%), and chloride (~ 1%). The contribution of organics decreased with the increase of size
mode, while the contributions of three major inorganic species (sulfate, nitrate and ammonium)
slightly increased with the increasing sizes (Fig. 4b).

**3.1.5 Diurnal variations of chemical species**

The average diurnal cycles of meteorological parameters as well as the $PM_1$ species and their mass
fractions for the entire campaign were shown in Fig. 5. All $PM_1$ species presented a similar diurnal
pattern with lower concentrations in the daytime whereas higher concentrations in the nighttime.
The mass concentrations reached the minimum values at around 15:00. This pattern was
accompanied with the enhanced wind speed and the increased air temperature in the afternoon
which could related with the dynamics of planetary boundary layer (PBL). After that, the mass
concentrations began to build up and reached to high levels in the nighttime. Note that the mass
concentrations of chloride and BC also existed a slight peak during the early morning, which
corresponded with the calm wind conditions and the lowest air temperature of the day and could
associated with the enhanced local emissions at QOMS in the morning. The diurnal cycles of mass
contributions of each $PM_1$ species were relatively stable for the entire campaign, besides the slight
increase of BC from 24% at ~ 08:00 to 30% at ~ 10:00. Overall, organics dominated $PM_1$
throughout the day (49−57%), followed by BC (23−30%), sulfate (9−10%), ammonium (5−6%),
nitrate (4−6%), and chloride (0.3−0.8%).

## 3.2 Bulk characteristics of OA

Figure 6a and b showed the average mass contributions of the four elements and the six ion
categories to total organics, respectively. The organic mass was on average composed of 36.8%
carbon, 57.9% oxygen, 4.0% hydrogen, and 1.3% nitrogen. For ionic categories, $C_xH_yO_1^+$ ions
dominated the total OA accounting for 41.3%, followed by $C_xH_yO_2^+$ (24.9%), $C_xH_y^+$ (23.9%),
$H_yO_1^+$ (6.1%), $C_xH_yN_p^+$ (2.9%) and $C_xH_yO_zN_p^+$ (0.9%). The contributions of oxygen and the two
major oxygenated ion fragments ($C_xH_yO_z^+ = C_xH_yO_1^+ + C_xH_yO_2^+$) at QOMS were quite higher than
those obtained at other urban or rural sites in China, whereas carbon and $C_xH_y^+$ ions had relative
lower contributions, e.g., 38% of $C_xH_yO_z^+$ and 21% of oxygen versus 56% of $C_xH_y^+$ and 70% of
carbon in urban Lanzhou (Xu et al., 2014), and 37.4% of $C_xH_yO_z^+$ versus 51.2% of $C_xH_y^+$ in urban
Nanjing (Wang et al., 2016a), suggesting that OA at QOMS were highly aged. Correspondingly,
the average high-resolution OA mass spectrum (Fig. 6c) also showed significantly high
contribution (~ 25%) at $m/z$ 44 signal (one of the most reliable marker of oxygenated OA)
compared with other ion fragments, e.g., 5% at $m/z$ 43 (indicator for less oxidized compounds),
1.7% at $m/z$ 55 (important COA fragment), and 0.4% at $m/z$ 57 (tracer for traffic-related emission)
(Alfarra et al., 2004; Zhang et al., 2005a). The average O/C ratio was 1.07 during this study, which
was much higher than those observed at various urban and rural sites in China using the I-A
method, e.g., 0.37 in Beijing (Sun et al., 2016), 0.36 in Lanzhou (Xu et al., 2016), 0.35 in Nanjing
(Wang et al., 2016a), and 0.65 in Ziyang (Hu et al., 2016)). Moreover, the average O/C ratio was
even higher than that of 0.98 at the background site of Mt. Wuzhi in southern China (Zhu et al.,
2016), indicating that OA at QOMS was more oxidized and aged during long-range transport. The
average H/C, N/C and OM/OC ratios were on average 1.29, 0.026 and 2.55 in this study,
determined a nominal chemical formula of OA as $C_1H_{1.29}O_{1.07}N_{0.026}$.
For the diurnal cycles, O/C ratio had two peaks in the early morning and late afternoon, likely
related to the production of secondary organic aerosol (SOA) via aqueous-phase reactions or
photochemical oxidation processes during these two periods. H/C and N/C ratios yet showed
inverse diurnal cycles with that of O/C, namely peaked at around 08:00−10:00 in the morning.
The Van Krevelen diagram (H/C versus O/C), which had been used widely to probe the oxidation
reaction mechanisms for bulk OA (Heald et al., 2010), showed an apparent anticorrelation ($R^2$ =
0.57) with a slope of −0.48 at QOMS (Fig. S8). Ng et al. (2011b) have suggested that a slope of −
0.5 indicate a net change in chemical composition from the addition of both acid and
alcohol/peroxide functional groups without fragmentation, and/or carboxylic acid groups with
fragmentation.

### 3.3 Organic aerosol source apportionment

Source apportionment via PMF analysis on the high-resolution OA mass spectrum identified three
distinct factors in this campaign according to their unique temporary variations, mass spectrum
(MS) profiles, element ratios, correlations with tracers, and diurnal patterns, i.e., a BB-related OA
(BBOA), a nitrogen-containing OA (NOA) and a more-oxidized oxygenated OA (MO-OOA).
Detailed discussion on each factor is given in the following subsections.

### 3.3.1 BBOA

Although significant high contribution at $m/z$ 44 (mostly $CO_2^+$) was found in all of the three OA
components, the BBOA MS was also characterized by contributions at $m/z$ 60 (mainly $C_2H_4O_2^+$)
and tiny $m/z$ 73 (mainly $C_3H_5O_2^+$) (Fig. 7g), which were generally regarded as well-known tracers
for BB emissions (Alfarra et al., 2007). The average fraction of the signal at $m/z$ 60 (referred as $f_{60}$)
in the BBOA mass spectrum was 0.61%, which was higher than the typical value of ∼ 0.3% in the
absence of BB impacts (Cubison et al., 2011). The time series of BBOA correlated tightly with
those of $C_2H_4O_2^+$ ($R^2 = 0.91$) and $C_3H_5O_2^+$ ($R^2 = 0.87$) as well as BC ($R^2 = 0.72$) and nitrate ($R^2 =$
0.75) (Fig. 7a and Table S2). If ignoring the influence of high contribution at $m/z$ 44, the BBOA
mass spectrum in this study correlated well ($R^2 = 0.5$–0.9) with those BBOA mass spectrum
identified at other sites worldwide (Ng et al., 2011a; Mohr et al., 2012; Saarikoski et al., 2012;
Crippa et al., 2013; Crippa et al., 2014; Xu et al., 2016), as shown in Fig. S9. The average mass
concentration of BBOA was 1.05 μg m$^{-3}$ for the entire study and contributed a large fraction
(43.7%) of the total OA mass on average (Fig. 8a), indicating that BBOA was an important
components of OA during the pre-monsoon season at the QOMS. The diurnal cycle of BBOA
showed high concentrations during nighttime whereas relatively low concentrations during
daytime (Fig. 7d). Correspondingly, the mass contributions of BBOA to total OA mass decreased
distinctly from ∼ 55% at 00:00 to 28% at 15:00 (Fig. 8b). In addition, higher mass concentrations
and contributions of BBOA were found during the two polluted episodes (PP1 and PP2) than those
during the clear periods, further indicating the important contribution of BBOA to OA in this
region. Figure 8c showed the mass fractions of the three OA components as a function of total OA
mass during the entire campaign. A continuously increased trend was found for the BBOA
contributions with the increasing OA mass, which contributed ∼ 15% when the total OA mass was
less than 0.3 μg m$^{-3}$, whereas it reached up to more than 75% with the OA mass increased to 9 μg
m$^{-3}$. This dominant contribution of BBOA during the polluted periods was consistent with those
results in previous studies that BB emission were an important source of aerosol to the southern
TPH (Engling et al., 2011; Xia et al., 2011; Putero et al., 2014; Cong et al., 2015). The O/C ratio
(0.85) of BBOA in this study was quite higher than those BBOA factors identified at other
urban/rural sites in previous studies (Aiken et al., 2009; Huang et al., 2011; Mohr et al., 2012; Sun
et al., 2016; Xu et al., 2016), suggesting its long-range transport feature. This aged BBOA feature
was similar with those obtained at other remote sites worldwide, such as a remote forest site in
Finland (Raatikainen et al., 2010), a remote background site in Greece (Bougiatioti et al., 2014),
and a national air quality background sites in southern China (Zhu et al., 2016), where OA were
generally highly oxidized.

### 3.3.2 NOA

Besides the two highest signals at $m/z$ 43.99 ($CO_2^+$) and 27.995 ($CO^+$) which together contributed
half of the total NOA signal due to the highly aged OA nature at QOMS, the NOA MS was also
characterized by some nitrogen-containing fragments, such as $m/z$ 27.011 ($CHN^+$), 41.027
($C_2H_3N^+$), and 43.006 ($CHON^+$). In total, these three fragments could comprise nearly half of the
nitrogen-containing signals in the NOA factor and finally contributed 5% of the total NOA signal.
The average O/C ratio of NOA for the entire campaign was 1.10 with the highest N/C ratio (0.068)
among the three OA components. This high N/C ratio at QOMS was comparable with those
nitrogen-containing OA factor identified in previous studies, such as 0.06 in Mexico City (Aiken
et al., 2009), 0.078 in Po Valley, Italy (Saarikoski et al., 2012), and 0.053 in New York (Sun et al.,
2012). The time series of NOA showed tightly correlation ($R^2$ = 0.62) with that of estimated
organic nitrates, whereas relatively weak correlations with $PM_1$ species and OA ions (Table S2). In
addition, the $f_{60}$ value (~ 0.37%) was also slightly higher than the background $f_{60}$ (0.3%) of BB
aerosols (Fig. 7h). These results together suggested that this oxygenated OA factor was likely a
nitrogen-containing OA and might be related to the aged BB emissions, consistent with the results
in previous studies that large amounts of nitrogen-containing organic compounds were found from
BB aerosols (Laskin et al., 2009; Gautam et al., 2016; Wang et al., 2017). In recently, Fleming et
al. (2017) found dung burning, a very popular activities in Himalayas and India for residential
cooking and heating, could emit much more nitrogen-containing OA than wood burning. Our filter
samples during high BBOA period analyzed by Fourier Transform Ion Cyclotron Resonance Mass
Spectrometry (FTICR-MS) also found amount of ON molecular (in preparation). As shown in Fig.
7e and 8b, both the diurnal cycles of mass concentrations and fractions of NOA had distinct
increase in the morning, similar with the diurnal patterns of chloride, element ratios of H/C and
N/C, and the estimated organic nitrates. This diurnal feature of NOA at QOMS was quite
consistent with those NOA factors identified in Po Valley, Italy (Saarikoski et al., 2012) and in
Mexico City (Aiken et al., 2009), or less-oxidized oxygenated OA (LO-OOA) in southeastern
USA (Xu et al., 2015) where have active BB emissions. NOA contributed ~ 14% of the total OA
mass on average, with an average mass concentration of 0.34 μg m$^{-3}$ for the entire study (Fig. 8a).

### 3.3.3 MO-OOA

An obvious more oxygenated OA factor was also identified in this study according to its
significant high signal at $m/z$ 44 (~ 25%) and the high average O/C ratio of 1.34 (Fig. 7i). The time
series of MO-OOA correlated closely ($R^2$ = 0.7) with sulfate and nitrate (Fig. 7c and Table S2).
Moreover, the mass spectrum of MO-OOA in this study resembled tightly to those more aged and
low-volatility oxygenated OA (LV-OOA) observed using AMS instruments at various sites
worldwide (Fig. S9), e.g., with $R^2$ of 0.89 and 0.97 to those in Lanzhou, China (Xu et al., 2014;
Zhang et al., 2017b), 0.96 to that in Paris, France (Crippa et al., 2013), 0.95 to that in Barcelona,
Spain (Mohr et al., 2012), as well as 0.70 and 0.71 to the standard LV-OOA mass spectrums
obtained from abundant AMS data sets by Ng et al. (2011a) and Crippa et al. (2014). The diurnal
variation of MO-OOA was mainly driven by the dynamic of PBL height, with high concentrations
during the nighttime yet relatively low concentrations during the daytime (Fig. 7f). This pattern
was quite different with those observed in previous studies that LV-OOA generally showed
elevated concentrations during the afternoon in accordance with strong photochemical activities,
suggesting that SOA at QOMS were mainly oxidized and aged during the long-range transport. On
average, MO-OOA contributed by 42.4% of the total OA mass, with an average mass
concentration of 1.02 μg m$^{-3}$ for the entire study (Fig. 8a). As shown in Fig. 2f and 8c, MO-OOA
also displayed enhanced mass contributions during the clear periods, especially for period after
May 2 when the average mass fraction of MO-OOA increased up to ~ 68% of the total OA mass.

## 3.4 Impact of BB emissions on aerosol characteristics

### 3.4.1 Sources of BB aerosols

In order to understand the transport pathways and the potential source areas of aerosol, 3-day back
trajectories of air mass were calculated at an ending height of 500 m above ground level every 6 h
at the QOMS from April 12 to May 12, 2016. A four-cluster solution and the wildfire hotspots
around the QOMS during the entire measurement period were presented in Fig. 9. Cluster 1 and 2
(C1 and C2), which originated from the west of the QOMS and passed over many hotspot areas
(e.g., Indo-Gangetic Plain and Nepal), represented two polluted clusters. On the contrary, C3 and
C4, which accounted for half of the total back trajectories, were identified as clear clusters. C3
traveled a short distance from the southwest of the QOMS, whereas C4 was from the north of the
QOMS and passed over the inland of the TPH. The average PM$_1$ mass concentrations for C1 and
C2 were 5.17 and 6.61 μg m$^{-3}$, respectively, which were 2−3 times higher than those for the two
clear clusters (2.74 and 2.21 μg m$^{-3}$). The mass contributions of OA and BBOA during C1 and C2
were up to more than 55% and 25% of the total PM$_1$ mass on average, whereas weak contributions
were found for the clear clusters (C3 and C4), indicating the significant impacts of BB emissions
from South Asia on aerosol loadings at QOMS.

### 3.4.2 Comparison of aerosol characteristics and air mass origins during different episodes

As shown in Fig. 2, the mass concentrations and compositions of PM$_1$ varied dynamically during
the entire sampling period. Two polluted periods (PP1 and PP2) and two clear periods (CP1 and
CP2) were identified. The comparisons of average mass concentrations and other indicators for the
four different episodes were presented in box plots in Fig. 10, whereas the corresponding back
trajectories of air masses and MODIS fire hotspots belong to each episode period were given in
Fig. 11, respectively.
During the two polluted periods, PM$_1$ mass concentrations were much higher than those in
clear periods (8.06 and 7.87 μg m$^{-3}$ for PP1 and PP2 vs. 2.76 and 1.82 μg m$^{-3}$ for CP1 and CP2;
similarly hereinafter), with higher contributions from OA (60.1% and 57.5% vs. 48.1% and 43.9%)
and BBOA (38.3% and 36.6% vs. 14.3% and 7.5%) (Fig. 10 and 11). In addition, $f_{60}$ were also
higher during polluted periods than those for clear periods (0.34% and 0.34% vs. 0.26% and 0.22%
on average) (Fig. 10). Air masses during PP1 and PP2 generally originated form long-range
transport to the west of the QOMS, which would pass through intense wildfires areas in South
Asia (e.g., Indo-Gangetic Plain and Nepal where showed high AOD values in Fig. 1d and active
fire hotspots in Fig. 9 and 11). The fire hotspot number around the air mass trajectories during PP2
was more than three times higher than those during other periods. Although the hotspot number
around the air mass trajectories during PP1 was not as abundant as that during PP2 and even
slightly lower than that during CP1, it was just collected within 3 days for PP1 whereas 8-10 days
for another periods. Hence, the BB activities were also more frequent and intense during the short
PP1 and finally resulted in the highest average $PM_1$ mass concentration among these periods.
Back trajectories in CP1 also originated from the west of QOMS and passed over the northern
India and Nepal, however, both the intensity of fire hotspot number (1089 hotspots in ~ 8 days)
and average FRP (19.6) were obvious lower than that in PP2. CP2 was the most clear period, of
which average $PM_1$ mass concentration was more than four times lower than those in polluted
periods. Back trajectories during CP2 period were from either the north of QOMS which passed
over inland areas of the TPH or the south of QOMS with quite short distance and low WS. These
results together suggested the significant roles of air mass sources and BB emissions to aerosol
characteristics at QOMS.

### 3.4.3 Case study on the chemical evolution of BB emission aerosols

In order to examine how atmospheric aging affects the aerosol chemistry characteristics at QOMS,
a typical evolution process of BB aerosol plume (referred as BB evolution case) was analyzed
from April 30 at 15:00 when a fresh BB plume occurred to May 1 at 18:00 when the BB plume
was highly aged after undergoing various atmospheric oxidation processes. The temporal
variations of meteorological parameters, mass concentrations and mass contributions of each $PM_1$
species and OA components as well as other chemistry parameters before and during this BB
evolution case were all shown in Fig. 12.
Before the BB evolution case, all the mass concentrations decreased slowly and synchronously
from 00:00 to 10:00 on April 30, which were consistent with the nearly stable trends of mass
contributions and other chemistry parameters, indicating the relatively unified air mass sources
and stable atmospheric conditions. After that, the wind circulations changed from the thermally-
driven down-slope winds (mostly southwest) to the weak up-slope winds (northeast). In this
period, BBOA and $f_{60}$ values kept relatively stable in contrast to other species likely due to the
weak of air dilution and local sources. All the species reached the minimum at around 15:00 due to
the lift of PBL.
The BB evolution case in this study was further divided into three different situations (as
marked with arrows in Fig. 12 and 13), including the arriving of the fresh BBOA plume (from
15:00 to 24:00 on April 30), followed by the aqueous-phase oxidation in the nighttime (from 2:30
to 7:10 on May 1) and photochemical oxidation in the daytime (from 10:00 to 18:00 on May 1).
All the mass concentrations began to increase from 15:00 and finally reached the maximum $PM_1$
mass loading of 18.4 μg m$^{-3}$ at 24:00, which was about four times higher than the average $PM_1$
mass during the entire campaign. Thus continuous increase was mainly dominated by the dramatic
increase of BBOA, which reached up to 10.8 μg m$^{-3}$ and contributed 88% of the total OA mass
and 50% of the total $PM_1$ at 24:00 (Fig. 13a), suggesting a distinct presence of BB emissions
during this period. In contrast, the total OA mass was comprised by 78% of MO-OOA and 12% of
BBOA at 15:00. Similar continuous increase trend could also be found for the mass concentration
of calculated organic nitrate in this stage. In addition, nine aerosol chemistry parameters were
presented as a function of BBOA mass concentrations during this period (Fig. 13b). The mass
contributions of OA to $PM_1$ ($f_{Org}$) and BBOA to total OA ($f_{BBOA}$), $f_{60}$, and H/C ratio were all
increased with the increasing BBOA mass, whereas the mass contribution of MO-OOA to total

OA ($f_{MO\text{-}OOA}$), O/C ratio, carbon oxidation state ($OS_c = 2 \times O/C - H/C$) of OA, and aerosol single scattering albedo (SSA) were decreased obviously, indicating the fresh nature of this BB plume. The significant impacts of fresh BB plume during this period was mainly associated with the unique wind circulation and the long-range transport of air masses. As displayed in Fig. 12b, the wind circulation changed from the weak up-slope winds to the strong down-slope glacier winds on April 30 at 15:00, with the WS increased from ~ 2 to 8 m s$^{-1}$. Meanwhile, the back trajectories in this period also presented that the long-range transport of air masses passed over the northern India and Nepal where active wildfires occurred, then air masses would accumulate and uplift to cross the Himalayas and finally downward to QOMS with the strong glacier winds.

A distinct aqueous-phase oxidation process was found in the nighttime from 02:30 to 07:10 on May 1. Although the total PM$_1$ and its species showed nearly stable mass concentrations during this period, the BBOA mass decreased gradually (from 82% to 70%) whereas MO-OOA increased constantly (from 14% to 20%) with the significant increase of RH (up to 91%) and aerosol liquid water content (ALWC) (Fig. 12). The scattering plots of the aerosol chemistry parameters versus the logarithmic values of cumulative ALWC, which could be used for the aqueous-phase oxidation during transport, also showed apparent increase trends for $f_{MO\text{-}OOA}$, O/C ratio, $OS_c$, and SSA that generally indicated the aerosol aging extent. All of these together suggested a distinct aqueous-phase oxidation of BBOA in the nighttime.

Since sunrise, all the mass concentrations decreased gradually, mainly related to the increasing PBL height and the clear air mass dilution. The back trajectories indicated that air masses during this period firstly went into the inland of the north of QOMS where had rare wildfires. Moreover, the BB plume would further undergo strong photochemical oxidation in the daytime due to the strong solar radiation. MO-OOA just contributed 26% of the total OA mass at 10:00, but it could increase to 74% at 18:00 after long-time photochemical oxidation. In contrast, BBOA mass contribution decreased from 49% to 20%. The cumulative solar radiation, which denoted the total amount of solar radiation that the plumes were exposed to during transport, could be used as an indicator for the extent of photochemical aging in the daytime (Zhou et al., 2017). Clear increased trend were found for $f_{MO\text{-}OOA}$, O/C ratio, $OS_c$, and SSA values with the increasing of cumulative solar radiation, whereas decreased trend in $f_{BBOA}$, $f_{NOA}$, H/C ratio, and $f_{60}$ values, suggesting a possible oxidation mechanism that the relatively fresh BBOA and NOA oxidized to aged MO-OOA in the daytime. Another interesting phenomenon was the continuous increase of SSA during both the aqueous-phase and photochemical oxidation periods on May 1 (Fig. 12e and 13b), indicating the potential influence of atmospheric aging to aerosol optical property at QOMS.

## 4 Conclusions

A comprehensive characterization of submicron aerosol chemical compositions and sources was investigated at the QOMS during the pre-monsoon season in 2016. The average mass concentration of PM$_1$ (NR-PM$_1$ + BC) was 4.44 ($\pm$ 4.54) μg m$^{-3}$ for the entire study, which was much lower than those observed in various sites in China. OA was the dominant PM$_1$ species (accounted for 54.3% of the total mass on average) and its contributions increased with the increase PM$_1$ mass loading. The average size distributions of all PM$_1$ species displayed an overlapping and narrow accumulation mode at ~ 500 nm, indicating the internally well-mixed and aged aerosol particles at QOMS. All species presented similar diurnal cycles, with lower

concentrations in the daytime whereas higher concentrations at the nighttime, mainly attributed to the dynamic variations of PBL height. Three OA factors were identified by PMF analysis on the high-resolution OA mass spectrum, including a relatively fresh BB-related OA (BBOA), a nitrogen-containing OA (NOA) and a more-oxidized oxygenated OA (MO-OOA). BBOA and MO-OOA could respectively account for 43.7% and 42.4% of OA mass on average, however, their contributions to OA showed completely opposite variation trends with the increase of OA mass. A continuously increased trend could be found for BBOA with the increasing OA, suggesting the key role of BBOA during polluted periods when frequent and intense wildfires were observed in South Asia. The significant impact of BB emissions on aerosol characteristics at QOMS have been also illustrated for different air mass origins and periods, respectively. Elevated $PM_1$ mass concentrations and high contributions of BBOA were found for both polluted clusters and polluted periods. A case study of typical evolution process of BB aerosol plume was investigated in detail to illustrate the chemical evolution of aerosol characteristics at QOMS. The fresh BB plume occurred in the afternoon on April 30 and finally resulted in highly $PM_1$ mass loading of 18.4 μg m$^{-3}$, which was about four times higher than the average $PM_1$ mass during the entire campaign. Obvious aqueous-phase oxidation and photochemical oxidation processes were analyzed in the nighttime and daytime on May 1, respectively, both suggesting the oxidation mechanism that fresh BBOA to aged MO-OOA. The continuous increase of SSA during the two oxidation periods suggested the potential influence of atmospheric aging to aerosol optical property at QOMS.

*Acknowledgements.* This research was supported by grants from the National Natural Science Foundation of China (41771079, 41421061), the Key Laboratory of Cryospheric Sciences Scientific Research Foundation (SKLCS-ZZ-2017-01), and the Chinese Academy of Sciences Hundred Talents Program. The authors thank the colleagues for continuing support and discussion, and thank the NOAA Air Resources Laboratory, NASA MODIS and FIRMS teams for providing the HYSPLIT trajectory model, AOD and fire hotspots datasets.

# References

Aiken, A. C., DeCarlo, P. F., Kroll, J. H., Worsnop, D. R., Huffman, J. A., Docherty, K. S., Ulbrich, I. M., Mohr, C., Kimmel, J. R., Sueper, D., Sun, Y., Zhang, Q., Trimborn, A., Northway, M., Ziemann, P. J., Canagaratna, M. R., Onasch, T. B., Alfarra, M. R., Prevot, A. S. H., Dommen, J., Duplissy, J., Metzger, A., Baltensperger, U., and Jimenez, J. L.: O/C and OM/OC ratios of primary, secondary, and ambient organic aerosols with high-resolution time-of-flight aerosol mass spectrometry, Environ. Sci. Technol., 42, 4478-4485, doi:10.1021/es703009q, 2008.

Aiken, A. C., Salcedo, D., Cubison, M. J., Huffman, J. A., DeCarlo, P. F., Ulbrich, I. M., Docherty, K. S., Sueper, D., Kimmel, J. R., Worsnop, D. R., Trimborn, A., Northway, M., Stone, E. A., Schauer, J. J., Volkamer, R. M., Fortner, E., de Foy, B., Wang, J., Laskin, A., Shutthanandan, V., Zheng, J., Zhang, R., Gaffney, J., Marley, N. A., Paredes-Miranda, G., Arnott, W. P., Molina, L. T., Sosa, G., and Jimenez, J. L.: Mexico City aerosol analysis during MILAGRO using high resolution aerosol mass spectrometry at the urban supersite (T0)–Part 1: Fine particle composition and organic source apportionment, Atmos. Chem. Phys., 9, 6633-6653, doi:10.5194/acp-9-6633-2009, 2009.

Alfarra, M. R., Coe, H., Allan, J. D., Bower, K. N., Boudries, H., Canagaratna, M. R., Jimenez, J. L., Jayne, J. T., Garforth, A. A., Li, S.-M., and Worsnop, D. R.: Characterization of urban and rural organic particulate in the Lower Fraser Valley using two Aerodyne Aerosol Mass Spectrometers, Atmos. Environ., 38, 5745-5758, doi:10.1016/j.atmosenv.2004.01.054, 2004.

Alfarra, M. R., Prevot, A. S. H., Szidat, S., Sandradewi, J., Weimer, S., Lanz, V. A., Schreiber, D., Mohr, M., and Baltensperger, U.: Identification of the Mass Spectral Signature of Organic Aerosols from Wood Burning Emissions, Environ. Sci. Technol., 41, 5770-5777, doi:10.1021/es062289b, 2007.

Bonasoni, P., Laj, P., Marinoni, A., Sprenger, M., Angelini, F., Arduini, J., Bonafè, U., Calzolari, F., Colombo, T., Decesari, S., Di Biagio, C., di Sarra, A. G., Evangelisti, F., Duchi, R., Facchini, M. C., Fuzzi, S., Gobbi, G. P., Maione, M., Panday, A., Roccato, F., Sellegri, K., Venzac, H., Verza, G. P., Villani, P., Vuillermoz, E., and Cristofanelli, P.: Atmospheric Brown Clouds in the

Himalayas: first two years of continuous observations at the Nepal Climate Observatory-Pyramid (5079 m), Atmos. Chem.
Phys., 10, 7515-7531, doi:10.5194/acp-10-7515-2010, 2010.
Bond, T. C., Streets, D. G., Yarber, K. F., Nelson, S. M., Woo, J.-H., and Klimont, Z.: A technology-based global inventory of
black and organic carbon emissions from combustion, J. Geophys. Res., 109, doi:10.1029/2003jd003697, 2004.
Bougiatioti, A., Stavroulas, I., Kostenidou, E., Zarmpas, P., Theodosi, C., Kouvarakis, G., Canonaco, F., Prévôt, A. S. H., Nenes,
A., Pandis, S. N., and Mihalopoulos, N.: Processing of biomass-burning aerosol in the eastern Mediterranean during
summertime, Atmos. Chem. Phys., 14, 4793-4807, doi:10.5194/acp-14-4793-2014, 2014.
Canagaratna, M. R., Jayne, J. T., Jimenez, J. L., Allan, J. D., Alfarra, M. R., Zhang, Q., Onasch, T. B., Drewnick, F., Coe, H.,
Middlebrook, A., Delia, A., Williams, L. R., Trimborn, A. M., Northway, M. J., DeCarlo, P. F., Kolb, C. E., Davidovits, P., and
Worsnop, D. R.: Chemical and microphysical characterization of ambient aerosols with the aerodyne aerosol mass
spectrometer, Mass Spectrom. Rev., 26, 185-222, doi:10.1002/mas.20115, 2007.
Canagaratna, M. R., Jimenez, J. L., Kroll, J. H., Chen, Q., Kessler, S. H., Massoli, P., Hildebrandt Ruiz, L., Fortner, E., Williams,
L. R., Wilson, K. R., Surratt, J. D., Donahue, N. M., Jayne, J. T., and Worsnop, D. R.: Elemental ratio measurements of organic
compounds using aerosol mass spectrometry: characterization, improved calibration, and implications, Atmos. Chem. Phys.,
15, 253-272, doi:10.5194/acp-15-253-2015, 2015.
Cong, Z., Kang, S., Kawamura, K., Liu, B., Wan, X., Wang, Z., Gao, S., and Fu, P.: Carbonaceous aerosols on the south edge of
the Tibetan Plateau: concentrations, seasonality and sources, Atmos. Chem. Phys., 15, 1573-1584, doi:10.5194/acp-15-1573-
2015, 2015.

Crippa, M., DeCarlo, P. F., Slowik, J. G., Mohr, C., Heringa, M. F., Chirico, R., Poulain, L., Freutel, F., Sciare, J., Cozic, J., Di
Marco, C. F., Elsasser, M., Nicolas, J. B., Marchand, N., Abidi, E., Wiedensohler, A., Drewnick, F., Schneider, J., Borrmann, S.,
Nemitz, E., Zimmermann, R., Jaffrezo, J. L., Prévôt, A. S. H., and Baltensperger, U.: Wintertime aerosol chemical composition
and source apportionment of the organic fraction in the metropolitan area of Paris, Atmos. Chem. Phys., 13, 961-981,
doi:10.5194/acp-13-961-2013, 2013.
Crippa, M., Canonaco, F., Lanz, V. A., Äijälä, M., Allan, J. D., Carbone, S., Capes, G., Ceburnis, D., Dall'Osto, M., Day, D. A.,
DeCarlo, P. F., Ehn, M., Eriksson, A., Freney, E., Hildebrandt Ruiz, L., Hillamo, R., Jimenez, J. L., Junninen, H., Kiendler-
Scharr, A., Kortelainen, A. M., Kulmala, M., Laaksonen, A., Mensah, A. A., Mohr, C., Nemitz, E., O'Dowd, C., Ovadnevaite,
J., Pandis, S. N., Petäjä, T., Poulain, L., Saarikoski, S., Sellegri, K., Swietlicki, E., Tiitta, P., Worsnop, D. R., Baltensperger, U.,
and Prévôt, A. S. H.: Organic aerosol components derived from 25 AMS data sets across Europe using a consistent ME-2
based source apportionment approach, Atmos. Chem. Phys., 14, 6159-6176, doi:10.5194/acp-14-6159-2014, 2014.
Cubison, M. J., Ortega, A. M., Hayes, P. L., Farmer, D. K., Day, D., Lechner, M. J., Brune, W. H., Apel, E., Diskin, G. S., Fisher,
J. A., Fuelberg, H. E., Hecobian, A., Knapp, D. J., Mikoviny, T., Riemer, D., Sachse, G. W., Sessions, W., Weber, R. J.,
Weinheimer, A. J., Wisthaler, A., and Jimenez, J. L.: Effects of aging on organic aerosol from open biomass burning smoke in
aircraft and laboratory studies, Atmos. Chem. Phys., 11, 12049-12064, doi:10.5194/acp-11-12049-2011, 2011.
DeCarlo, P. F., Kimmel, J. R., Trimborn, A., Northway, M. J., Jayne, J. T., Aiken, A. C., Gonin, M., Fuhrer, K., Horvath, T.,
Docherty, K. S., Worsnop, D. R., and Jimenez, J. L.: Field-Deployable, High-Resolution, Time-of-Flight Aerosol Mass
Spectrometer, Anal. Chem., 78, 8281-8289, doi:10.1021/ac061249n, 2006.
Decesari, S., Facchini, M. C., Carbone, C., Giulianelli, L., Rinaldi, M., Finessi, E., Fuzzi, S., Marinoni, A., Cristofanelli, P.,
Duchi, R., Bonasoni, P., Vuillermoz, E., Cozic, J., Jaffrezo, J. L., and Laj, P.: Chemical composition of PM10 and PM1 at the
high-altitude Himalayan station Nepal Climate Observatory-Pyramid (NCO-P) (5079 m a.s.l.), Atmos. Chem. Phys., 10, 4583-
4596, doi:10.5194/acp-10-4583-2010, 2010.
Draxler, R. R., and Rolph, G. D.: HYSPLIT (HYbrid Single-Particle Lagrangian Integrated Trajectory) model access via NOAA
ARL READY website (http://www.arl.noaa.gov/ready/hysplit4.html). NOAA Air Resources Laboratory, Silver Spring, MD,
USA, 2003.
Du, W., Sun, Y. L., Xu, Y. S., Jiang, Q., Wang, Q. Q., Yang, W., Wang, F., Bai, Z. P., Zhao, X. D., and Yang, Y. C.: Chemical
characterization of submicron aerosol and particle growth events at a national background site (3295 m a.s.l.) on the Tibetan
Plateau, Atmos. Chem. Phys., 15, 10811-10824, doi:10.5194/acp-15-10811-2015, 2015.
Duan, A. M., and Wu, G. X.: Role of the Tibetan Plateau thermal forcing in the summer climate patterns over subtropical Asia,
Climate Dynamics, 24, 793-807, doi:10.1007/s00382-004-0488-8, 2005.
Engling, G., Zhang, Y. N., Chan, C. Y., Sang, X. F., Lin, M., Ho, K. F., Li, Y. S., Lin, C. Y., and Lee, J. J.: Characterization and
sources of aerosol particles over the southeastern Tibetan Plateau during the Southeast Asia biomass-burning season, Tellus B,
63, 117-128, doi:10.1111/j.1600-0889.2010.00512.x, 2011.
Fleming, L. T., Lin, P., Laskin, A., Laskin, J., Weltman, R., Edwards, R. D., Arora, N. K., Yadav, A., Meinardi, S., Blake, D. R.,
Pillarisetti, A., Smith, K. R., and Nizkorodov, S. A.: Molecular Composition of Particulate Matter Emissions from Dung and
Brushwood Burning Household Cookstoves in Haryana, India, Atmos. Chem. Phys. Discuss., 1-35, doi:10.5194/acp-2017-784,
2017.

Fröhlich, R., Cubison, M. J., Slowik, J. G., Bukowiecki, N., Canonaco, F., Croteau, P. L., Gysel, M., Henne, S., Herrmann, E.,
Jayne, J. T., Steinbacher, M., Worsnop, D. R., Baltensperger, U., and Prévôt, A. S. H.: Fourteen months of on-line
measurements of the non-refractory submicron aerosol at the Jungfraujoch (3580 m a.s.l.) – chemical composition, origins and
organic aerosol sources, Atmos. Chem. Phys., 15, 11373-11398, doi:10.5194/acp-15-11373-2015, 2015.
Gautam, S., Edwards, R., Yadav, A., Weltman, R., Pillarsetti, A., Arora, N. K., and Smith, K. R.: Probe-based measurements of
moisture in dung fuel for emissions measurements, Energy for Sustainable Development, 35, 1-6,
doi:10.1016/j.esd.2016.09.003, 2016.
Ge, X., Zhang, Q., Sun, Y., Ruehl, C. R., and Setyan, A.: Effect of aqueous-phase processing on aerosol chemistry and size
distributions in Fresno, California, during wintertime, Environ. Chem., 9, 221, doi:10.1071/en11168, 2012.
Heald, C. L., Kroll, J. H., Jimenez, J. L., Docherty, K. S., DeCarlo, P. F., Aiken, A. C., Chen, Q., Martin, S. T., Farmer, D. K., and
Artaxo, P.: A simplified description of the evolution of organic aerosol composition in the atmosphere, Geophys. Res. Lett., 37,
L08803, doi:10.1029/2010gl042737, 2010.
Hou, S., Qin, D., Zhang, D., Kang, S., Mayewski, P. A., and Wake, C. P.: A 154a high-resolution ammonium record from the
Rongbuk Glacier, north slope of Mt. Qomolangma (Everest), Tibet–Himal region, Atmos. Environ., 37, 721-729,
doi:10.1016/S1352-2310(02)00582-4, 2003.
Hu, W., Hu, M., Hu, W.-W., Niu, H., Zheng, J., Wu, Y., Chen, W., Chen, C., Li, L., Shao, M., Xie, S., and Zhang, Y.:
Characterization of submicron aerosols influenced by biomass burning at a site in the Sichuan Basin, southwestern China,
Atmos. Chem. Phys., 16, 13213-13230, doi:10.5194/acp-16-13213-2016, 2016.
Huang, X. F., He, L. Y., Hu, M., Canagaratna, M. R., Kroll, J. H., Ng, N. L., Zhang, Y. H., Lin, Y., Xue, L., Sun, T. L., Liu, X. G.,
Shao, M., Jayne, J. T., and Worsnop, D. R.: Characterization of submicron aerosols at a rural site in Pearl River Delta of China
using an Aerodyne High-Resolution Aerosol Mass Spectrometer, Atmos. Chem. Phys., 11, 1865-1877, doi:10.5194/acp-11-
1865-2011, 2011.

Huang, X. F., He, L. Y., Xue, L., Sun, T. L., Zeng, L. W., Gong, Z. H., Hu, M., and Zhu, T.: Highly time-resolved chemical
characterization of atmospheric fine particles during 2010 Shanghai World Expo, Atmos. Chem. Phys., 12, 4897-4907,
doi:10.5194/acp-12-4897-2012, 2012.
Jayne, J. T., Leard, D. C., Zhang, X. F., Davidovits, P., Smith, K. A., Kolb, C. E., and Worsnop, D. R.: Development of an aerosol
mass spectrometer for size and composition analysis of submicron particles, Aerosol Sci. Technol., 33, 49-70,
doi:10.1080/027868200410840, 2000.
Jimenez, J. L., Jayne, J. T., Shi, Q., Kolb, C. E., Worsnop, D. R., Yourshaw, I., Seinfeld, J. H., Flagan, R. C., Zhang, X., Smith, K.
A., Morris, J. W., and Davidovits, P.: Ambient aerosol sampling using the Aerodyne Aerosol Mass Spectrometer, J. Geophys.
Res., 108, doi:10.1029/2001jd001213, 2003.
Jimenez, J. L., Canagaratna, M. R., Donahue, N. M., Prevot, A. S., Zhang, Q., Kroll, J. H., DeCarlo, P. F., Allan, J. D., Coe, H.,
Ng, N. L., Aiken, A. C., Docherty, K. S., Ulbrich, I. M., Grieshop, A. P., Robinson, A. L., Duplissy, J., Smith, J. D., Wilson, K.
R., Lanz, V. A., Hueglin, C., Sun, Y. L., Tian, J., Laaksonen, A., Raatikainen, T., Rautiainen, J., Vaattovaara, P., Ehn, M.,
Kulmala, M., Tomlinson, J. M., Collins, D. R., Cubison, M. J., Dunlea, E. J., Huffman, J. A., Onasch, T. B., Alfarra, M. R.,
Williams, P. I., Bower, K., Kondo, Y., Schneider, J., Drewnick, F., Borrmann, S., Weimer, S., Demerjian, K., Salcedo, D.,
Cottrell, L., Griffin, R., Takami, A., Miyoshi, T., Hatakeyama, S., Shimono, A., Sun, J. Y., Zhang, Y. M., Dzepina, K., Kimmel,
J. R., Sueper, D., Jayne, J. T., Herndon, S. C., Trimborn, A. M., Williams, L. R., Wood, E. C., Middlebrook, A. M., Kolb, C. E.,
Baltensperger, U., and Worsnop, D. R.: Evolution of organic aerosols in the atmosphere, Science, 326, 1525-1529,
doi:10.1126/science.1180353, 2009.
Kang, S., Xu, Y., You, Q., Flügel, W.-A., Pepin, N., and Yao, T.: Review of climate and cryospheric change in the Tibetan Plateau,
Environ. Res. Lett., 5, 015101, doi:10.1088/1748-9326/5/1/015101, 2010.
Laskin, A., Smith, J. S., and Laskin, J.: Molecular Characterization of Nitrogen-Containing Organic Compounds in Biomass
Burning Aerosols Using High-Resolution Mass Spectrometry, Environ. Sci. Technol., 43, 3764-3771, doi:10.1021/es803456n,
2009.

Lau, K. M., Kim, M. K., and Kim, K. M.: Asian summer monsoon anomalies induced by aerosol direct forcing: the role of the
Tibetan Plateau, Climate Dynamics, 26, 855-864, doi:10.1007/s00382-006-0114-z, 2006.
Lau, K. M., Tsay, S. C., Hsu, C., Chin, M., Ramanathan, V., Wu, G. X., Li, Z., Sikka, R., Holben, B., Lu, D., Chen, H., Tartari, G.,
Koudelova, P., Ma, Y., Huang, J., Taniguchi, K., and Zhang, R.: The Joint Aerosol–Monsoon Experiment: A New Challenge
for Monsoon Climate Research, Bulletin of the American Meteorological Society, 89, 369-383, doi:10.1175/bams-89-3-369,
2008.

Li, C., Bosch, C., Kang, S., Andersson, A., Chen, P., Zhang, Q., Cong, Z., Chen, B., Qin, D., and Gustafsson, O.: Sources of
black carbon to the Himalayan-Tibetan Plateau glaciers, Nat. Commun., 7, 12574, doi:10.1038/ncomms12574, 2016.
Li, Y. J., Sun, Y., Zhang, Q., Li, X., Li, M., Zhou, Z., and Chan, C. K.: Real-time chemical characterization of atmospheric
particulate matter in China: A review, Atmos. Environ., 158, 270-304, doi:10.1016/j.atmosenv.2017.02.027, 2017.
Liu, B., Cong, Z., Wang, Y., Xin, J., Wan, X., Pan, Y., Liu, Z., Wang, Y., Zhang, G., Wang, Z., Wang, Y., and Kang, S.:
Background aerosol over the Himalayas and Tibetan Plateau: observed characteristics of aerosol mass loading, Atmos. Chem.
Phys., 17, 449-463, doi:10.5194/acp-17-449-2017, 2017.
Liu, Z., Liu, D., Huang, J., Vaughan, M., Uno, I., Sugimoto, N., Kittaka, C., Trepte, C., Wang, Z., Hostetler, C., and Winker, D.:
Airborne dust distributions over the Tibetan Plateau and surrounding areas derived from the first year of CALIPSO lidar
observations, Atmos. Chem. Phys., 8, 5045-5060, doi:10.5194/acp-8-5045-2008, 2008.
Ma, Y., Kang, S., Zhu, L., Xu, B., Tian, L., and Yao, T.: ROOF OF THE WORLD: Tibetan Observation and Research Platform,
Bulletin of the American Meteorological Society, 89, 1487-1492, doi:10.1175/2008bams2545.1, 2008.

Marcq, S., Laj, P., Roger, J. C., Villani, P., Sellegri, K., Bonasoni, P., Marinoni, A., Cristofanelli, P., Verza, G. P., and Bergin, M.:
Aerosol optical properties and radiative forcing in the high Himalaya based on measurements at the Nepal Climate
Observatory-Pyramid site (5079 m a.s.l.), Atmos. Chem. Phys., 10, 5859-5872, doi:10.5194/acp-10-5859-2010, 2010.

Marinoni, A., Cristofanelli, P., Laj, P., Duchi, R., Putero, D., Calzolari, F., Landi, T. C., Vuillermoz, E., Maione, M., and
Bonasoni, P.: High black carbon and ozone concentrations during pollution transport in the Himalayas: Five years of
continuous observations at NCO-P global GAW station, J. Environ. Sci., 25, 1618-1625, doi:10.1016/S1001-0742(12)60242-3,
2013.

Mohr, C., DeCarlo, P. F., Heringa, M. F., Chirico, R., Slowik, J. G., Richter, R., Reche, C., Alastuey, A., Querol, X., Seco, R.,
Peñuelas, J., Jiménez, J. L., Crippa, M., Zimmermann, R., Baltensperger, U., and Prévôt, A. S. H.: Identification and
quantification of organic aerosol from cooking and other sources in Barcelona using aerosol mass spectrometer data, Atmos.
Chem. Phys., 12, 1649-1665, doi:10.5194/acp-12-1649-2012, 2012.

Ng, N., Canagaratna, M., Jimenez, J., Zhang, Q., Ulbrich, I., and Worsnop, D.: Real-time methods for estimating organic
component mass concentrations from aerosol mass spectrometer data, Environ. Sci. Technol., 45, 910-916,
doi:10.1021/es102951k, 2011a.

Ng, N. L., Canagaratna, M. R., Jimenez, J. L., Chhabra, P. S., Seinfeld, J. H., and Worsnop, D. R.: Changes in organic aerosol
composition with aging inferred from aerosol mass spectra, Atmos. Chem. Phys., 11, 6465-6474, doi:10.5194/acp-11-6465-
2011, 2011b.

Paatero, P., and Tapper, U.: Positive matrix factorization: A non-negative factor model with optimal utilization of error estimates
of data values, Environmetrics, 5, 111-126, doi:10.1002/env.3170050203, 1994.

Putero, D., Landi, T. C., Cristofanelli, P., Marinoni, A., Laj, P., Duchi, R., Calzolari, F., Verza, G. P., and Bonasoni, P.: Influence
of open vegetation fires on black carbon and ozone variability in the southern Himalayas (NCO-P, 5079 m a.s.l.), Environ.
Pollut., 184, 597-604, doi:10.1016/j.envpol.2013.09.035, 2014.

Raatikainen, T., Vaattovaara, P., Tiitta, P., Miettinen, P., Rautiainen, J., Ehn, M., Kulmala, M., Laaksonen, A., and Worsnop, D. R.:
Physicochemical properties and origin of organic groups detected in boreal forest using an aerosol mass spectrometer, Atmos.
Chem. Phys., 10, 2063-2077, doi:10.5194/acp-10-2063-2010, 2010.

Ram, K., Sarin, M. M., and Hegde, P.: Long-term record of aerosol optical properties and chemical composition from a high-
altitude site (Manora Peak) in Central Himalaya, Atmos. Chem. Phys., 10, 11791-11803, doi:10.5194/acp-10-11791-2010,
2010.

Ramanathan, V., and Carmichael, G.: Global and regional climate changes due to black carbon, Nature Geoscience, 1, 221-227,
doi:10.1038/ngeo156, 2008.

Saarikoski, S., Carbone, S., Decesari, S., Giulianelli, L., Angelini, F., Canagaratna, M., Ng, N. L., Trimborn, A., Facchini, M. C.,
Fuzzi, S., Hillamo, R., and Worsnop, D.: Chemical characterization of springtime submicrometer aerosol in Po Valley, Italy,
Atmos. Chem. Phys., 12, 8401-8421, doi:10.5194/acp-12-8401-2012, 2012.

Sun, Y., Zhang, Q., Macdonald, A., Hayden, K., Li, S., Liggio, J., Liu, P., Anlauf, K., Leaitch, W., and Steffen, A.: Size-resolved
aerosol chemistry on Whistler Mountain, Canada with a high-resolution aerosol mass spectrometer during INTEX-B, Atmos.
Chem. Phys., 9, 3095-3111, doi:10.5194/acp-9-3095-2009, 2009.

Sun, Y. L., Zhang, Q., Schwab, J. J., Chen, W. N., Bae, M. S., Hung, H. M., Lin, Y. C., Ng, N. L., Jayne, J., Massoli, P., Williams,
781       L. R., and Demerjian, K. L.: Characterization of near-highway submicron aerosols in New York City with a high-resolution
aerosol mass spectrometer, Atmos. Chem. Phys., 12, 2215-2227, doi:10.5194/acp-12-2215-2012, 2012.

Sun, Y., Wang, Z., Fu, P., Jiang, Q., Yang, T., Li, J., and Ge, X.: The impact of relative humidity on aerosol composition and
evolution processes during wintertime in Beijing, China, Atmos. Environ., 77, 927-934, doi:10.1016/j.atmosenv.2013.06.019,
2013.

Sun, Y., Du, W., Fu, P., Wang, Q., Li, J., Ge, X., Zhang, Q., Zhu, C., Ren, L., Xu, W., Zhao, J., Han, T., Worsnop, D. R., and
Wang, Z.: Primary and secondary aerosols in Beijing in winter: sources, variations and processes, Atmos. Chem. Phys., 16,
8309-8329, doi:10.5194/acp-16-8309-2016, 2016.

Takami, A., Miyoshi, T., Shimono, A., and Hatakeyama, S.: Chemical composition of fine aerosol measured by AMS at Fukue
Island, Japan during APEX period, Atmos. Environ., 39, 4913-4924, doi:10.1016/j.atmosenv.2005.04.038, 2005.

Ulbrich, I. M., Canagaratna, M. R., Zhang, Q., Worsnop, D. R., and Jimenez, J. L.: Interpretation of organic components from
Positive Matrix Factorization of aerosol mass spectrometric data, Atmos. Chem. Phys., 9, 2891-2918, doi:10.5194/acp-9-2891-
2009, 2009.

Wan, X., Kang, S., Li, Q., Rupakheti, D., Zhang, Q., Guo, J., Chen, P., Tripathee, L., Rupakheti, M., Panday, A. K., Wang, W.,
Kawamura, K., Gao, S., Wu, G., and Cong, Z.: Organic molecular tracers in the atmospheric aerosols from Lumbini, Nepal, in
the northern Indo-Gangetic Plain: influence of biomass burning, Atmos. Chem. Phys., 17, 8867-8885, doi:10.5194/acp-17-
8867-2017, 2017.

Wang, J., Ge, X., Chen, Y., Shen, Y., Zhang, Q., Sun, Y., Xu, J., Ge, S., Yu, H., and Chen, M.: Highly time-resolved urban aerosol
characteristics during springtime in Yangtze River Delta, China: insights from soot particle aerosol mass spectrometry, Atmos.

Chem. Phys., 16, 9109-9127, doi:10.5194/acp-16-9109-2016, 2016a.

Wang, J., Onasch, T. B., Ge, X., Collier, S., Zhang, Q., Sun, Y., Yu, H., Chen, M., Prévôt, A. S. H., and Worsnop, D. R.: Observation of Fullerene Soot in Eastern China, Environ. Sci. Technol. Lett., 3, 121-126, doi:10.1021/acs.estlett.6b00044, 2016b.

Wang, J., Zhang, Q., Chen, M., Collier, S., Zhou, S., Ge, X., Xu, J., Shi, J., Xie, C., Hu, J., Ge, S., Sun, Y., and Coe, H.: First Chemical Characterization of Refractory Black Carbon Aerosols and Associated Coatings over the Tibetan Plateau (4730 m a.s.l), Environ. Sci. Technol., 51, 14072-14082, doi:10.1021/acs.est.7b03973, 2017.

Wang, Y., Hu, M., Lin, P., Guo, Q., Wu, Z., Li, M., Zeng, L., Song, Y., Zeng, L., Wu, Y., Guo, S., Huang, X., and He, L.: Molecular Characterization of Nitrogen-Containing Organic Compounds in Humic-like Substances Emitted from Straw Residue Burning, Environ. Sci. Technol., 51, 5951-5961, doi:10.1021/acs.est.7b00248, 2017.

Wu, G., Liu, Y., Zhang, Q., Duan, A., Wang, T., Wan, R., Liu, X., Li, W., Wang, Z., and Liang, X.: The Influence of Mechanical and Thermal Forcing by the Tibetan Plateau on Asian Climate, Journal of Hydrometeorology, 8, 770-789, doi:10.1175/jhm609.1, 2007.

Xia, X., Zong, X., Cong, Z., Chen, H., Kang, S., and Wang, P.: Baseline continental aerosol over the central Tibetan plateau and a case study of aerosol transport from South Asia, Atmos. Environ., 45, 7370-7378, doi:10.1016/j.atmosenv.2011.07.067, 2011.

Xu, B., Cao, J., Hansen, J., Yao, T., Joswia, D. R., Wang, N., Wu, G., Wang, M., Zhao, H., Yang, W., Liu, X., and He, J.: Black soot and the survival of Tibetan glaciers, Proc. Natl. Acad. Sci. USA, 106, 22114-22118, doi:10.1073/pnas.0910444106, 2009.

Xu, J., Zhang, Q., Chen, M., Ge, X., Ren, J., and Qin, D.: Chemical composition, sources, and processes of urban aerosols during summertime in northwest China: insights from high-resolution aerosol mass spectrometry, Atmos. Chem. Phys., 14, 12593-12611, doi:10.5194/acp-14-12593-2014, 2014.

Xu, J., Shi, J., Zhang, Q., Ge, X., Canonaco, F., Prévôt, A. S. H., Vonwiller, M., Szidat, S., Ge, J., Ma, J., An, Y., Kang, S., and Qin, D.: Wintertime organic and inorganic aerosols in Lanzhou, China: sources, processes, and comparison with the results during summer, Atmos. Chem. Phys., 16, 14937-14957, doi:10.5194/acp-16-14937-2016, 2016.

Xu, J., Zhang, Q., Shi, J., Ge, X., Xie, C., Wang, J., Kang, S., Zhang, R., and Wang, Y.: Chemical characteristics of submicron particles at the central Tibet Plateau: influence of long-range transport, Atmos. Chem. Phys. Discuss., 1-32, doi:10.5194/acp-2017-587, 2017.

Xu, L., Suresh, S., Guo, H., Weber, R. J., and Ng, N. L.: Aerosol characterization over the southeastern United States using high-resolution aerosol mass spectrometry: spatial and seasonal variation of aerosol composition and sources with a focus on organic nitrates, Atmos. Chem. Phys., 15, 7307-7336, doi:10.5194/acp-15-7307-2015, 2015.

Yadav, I. C., Devi, N. L., Li, J., Syed, J. H., Zhang, G., and Watanabe, H.: Biomass burning in Indo-China peninsula and its impacts on regional air quality and global climate change-a review, Environmental pollution, 227, 414-427, doi:10.1016/j.envpol.2017.04.085, 2017.

Yang, K., Wu, H., Qin, J., Lin, C., Tang, W., and Chen, Y.: Recent climate changes over the Tibetan Plateau and their impacts on energy and water cycle: A review, Global and Planetary Change, 112, 79-91, doi:10.1016/j.gloplacha.2013.12.001, 2014.

Yao, T., Thompson, L., Mosbrugger, V., Zhang, F., Ma, Y., Luo, T., Xu, B., Yang, X., Joswiak, D. R., Wang, W., Joswiak, M. E., Devkota, L. P., Tayal, S., Jilani, R., and Fayziev, R.: Third Pole Environment (TPE), Environmental Development, 3, 52-64, doi:10.1016/j.envdev.2012.04.002, 2012a.

Yao, T., Thompson, L., Yang, W., Yu, W., Gao, Y., Guo, X., Yang, X., Duan, K., Zhao, H., Xu, B., Pu, J., Lu, A., Xiang, Y., Kattel, D. B., and Joswiak, D.: Different glacier status with atmospheric circulations in Tibetan Plateau and surroundings, Nature Clim. Change, 2, 663-667, doi:10.1038/nclimate1580, 2012b.

Zhang, Q., Alfarra, M. R., Worsnop, D. R., Allan, J. D., Coe, H., Canagaratna, M. R., and Jimenez, J. L.: Deconvolution and quantification of hydrocarbon-like and oxygenated organic aerosols based on aerosol mass spectrometry, Environ. Sci. Technol., 39, 4938-4952, doi:10.1021/es048568l, 2005a.

Zhang, Q., Canagaratna, M. R., Jayne, J. T., Worsnop, D. R., and Jimenez, J. L.: Time- and size-resolved chemical composition of submicron particles in Pittsburgh: Implications for aerosol sources and processes, J. Geophys. Res., 110, doi:10.1029/2004jd004649, 2005b.

Zhang, Q., Jimenez, J. L., Canagaratna, M. R., Allan, J. D., Coe, H., Ulbrich, I., Alfarra, M. R., Takami, A., Middlebrook, A. M., Sun, Y. L., Dzepina, K., Dunlea, E., Docherty, K., DeCarlo, P. F., Salcedo, D., Onasch, T., Jayne, J. T., Miyoshi, T., Shimono, A., Hatakeyama, S., Takegawa, N., Kondo, Y., Schneider, J., Drewnick, F., Borrmann, S., Weimer, S., Demerjian, K., Williams, P., Bower, K., Bahreini, R., Cottrell, L., Griffin, R. J., Rautiainen, J., Sun, J. Y., Zhang, Y. M., and Worsnop, D. R.: Ubiquity and dominance of oxygenated species in organic aerosols in anthropogenically-influenced Northern Hemisphere midlatitudes, Geophys. Res. Lett., 34, doi:10.1029/2007gl029979, 2007a.

Zhang, Q., Jimenez, J. L., Worsnop, D. R., and Canagaratna, M.: A case study of urban particle acidity and its influence on secondary organic aerosol, Environ. Sci. Technol., 41, 3213-3219, doi:10.1021/es061812j, 2007b.

Zhang, Q., Jimenez, J. L., Canagaratna, M. R., Ulbrich, I. M., Ng, N. L., Worsnop, D. R., and Sun, Y.: Understanding atmospheric organic aerosols via factor analysis of aerosol mass spectrometry: a review, Anal. Bioanal. Chem., 401, 3045-3067, doi:10.1007/s00216-011-5355-y, 2011.

Zhang, R., Wang, Y., He, Q., Chen, L., Zhang, Y., Qu, H., Smeltzer, C., Li, J., Alvarado, L. M. A., Vrekoussis, M., Richter, A.,

Wittrock, F., and Burrows, J. P.: Enhanced trans-Himalaya pollution transport to the Tibetan Plateau by cut-off low systems,
Atmos. Chem. Phys., 17, 3083-3095, doi:10.5194/acp-17-3083-2017, 2017a.
Zhang, X., Zhang, Y., Sun, J., Yu, Y., Canonaco, F., Prevot, A. S., and Li, G.: Chemical characterization of submicron aerosol
particles during wintertime in a northwest city of China using an Aerodyne aerosol mass spectrometry, Environ. Pollut., 222,
567-582, doi:10.1016/j.envpol.2016.11.012, 2017b.
Zhao, Z., Cao, J., Shen, Z., Xu, B., Zhu, C., Chen, L. W. A., Su, X., Liu, S., Han, Y., Wang, G., and Ho, K.: Aerosol particles at a
high-altitude site on the Southeast Tibetan Plateau, China: Implications for pollution transport from South Asia, J. Geophys.
Res.-Atmos., 118, 11360-11375, doi:10.1002/jgrd.50599, 2013.
Zheng, J., Hu, M., Du, Z., Shang, D., Gong, Z., Qin, Y., Fang, J., Gu, F., Li, M., Peng, J., Li, J., Zhang, Y., Huang, X., He, L., Wu,
Y., and Guo, S.: Influence of biomass burning from South Asia at a high-altitude mountain receptor site in China, Atmos.
Chem. Phys., 17, 6853-6864, doi:10.5194/acp-17-6853-2017, 2017.
Zhou, S., Collier, S., Jaffe, D. A., Briggs, N. L., Hee, J., Sedlacek Iii, A. J., Kleinman, L., Onasch, T. B., and Zhang, Q.: Regional
influence of wildfires on aerosol chemistry in the western US and insights into atmospheric aging of biomass burning organic
aerosol, Atmos. Chem. Phys., 17, 2477-2493, doi:10.5194/acp-17-2477-2017, 2017.
Zhu, Q., He, L. Y., Huang, X. F., Cao, L. M., Gong, Z. H., Wang, C., Zhuang, X., and Hu, M.: Atmospheric aerosol compositions
and sources at two national background sites in northern and southern China, Atmos. Chem. Phys., 16, 10283-10297,
doi:10.5194/acp-16-10283-2016, 2016.
Zou, H., Zhou, L., Ma, S., Li, P., Wang, W., Li, A., Jia, J., and Gao, D.: Local wind system in the Rongbuk Valley on the northern
slope of Mt. Everest, Geophys. Res. Lett., 35, doi:10.1029/2008gl033466, 2008.

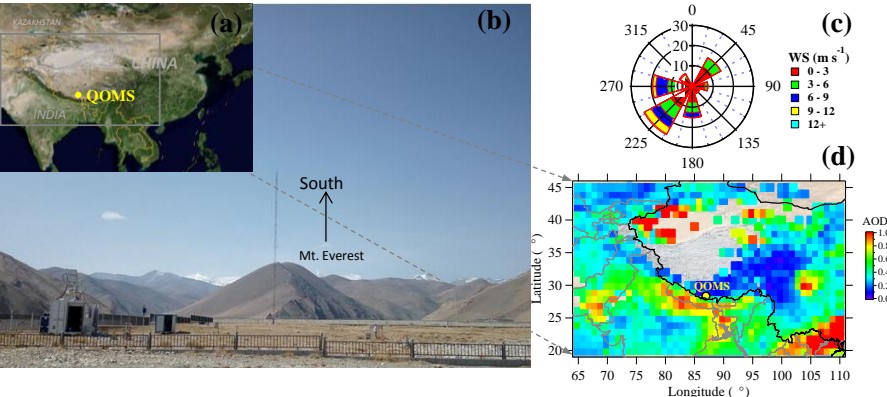

**Figure 1. (a)** Location map for the QOMS, **(b)** picture for the QOMS and its surrounding, **(c)** wind rose plot colored by wind
speed in this study, and **(d)** distribution of the average aerosol optical depth (AOD) around the QOMS retrieved from Terra
MODIS at 550 nm during this study.

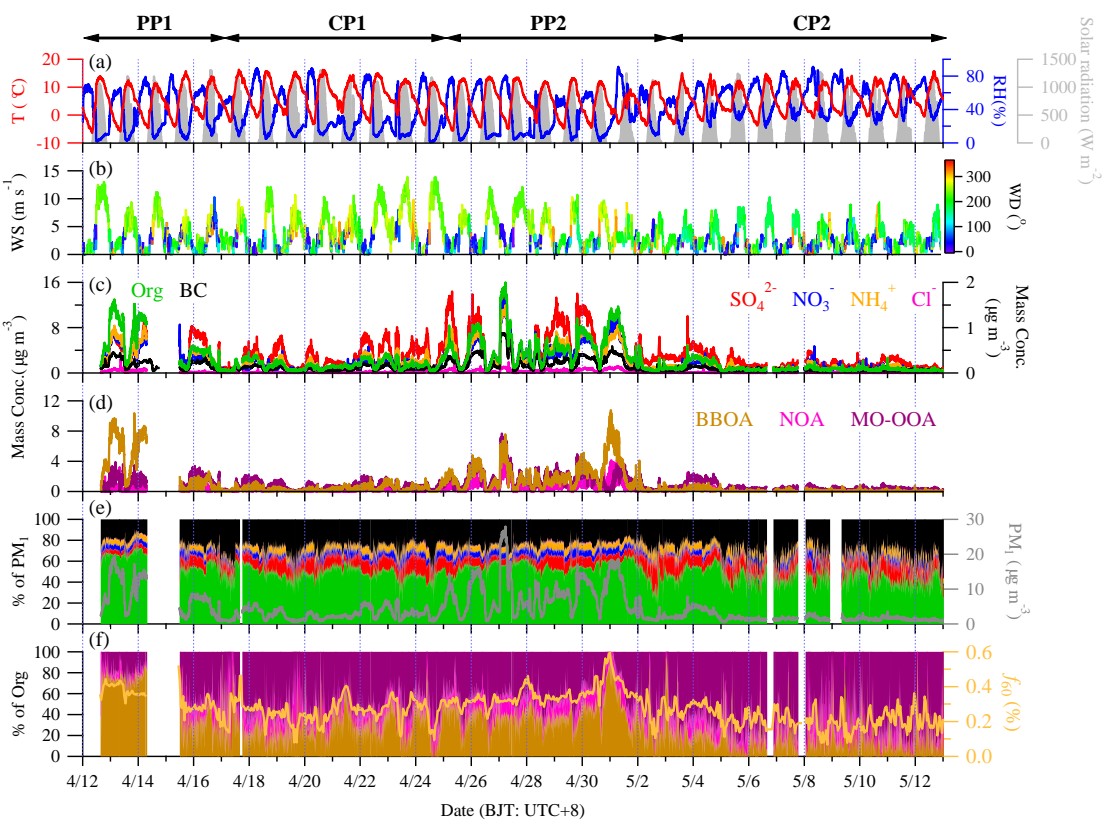

**Figure 2.** Summary of meteorological and HR-ToF-AMS data. The 5-min time series of **(a)** ambient temperature ($T$), relative
humidity (RH), and solar radiation, **(b)** wind speed (WS) colored by wind direction (WD), **(c)** mass concentrations of $PM_1$
species, **(d)** mass concentrations of organic components, **(e)** mass contributions of $PM_1$ species to total $PM_1$ as well as total $PM_1$
mass concentrations, and **(f)** mass contributions of organic components to organics. The time series of hourly average $f_{60}$ (=
$C_2H_4O_2^+$ / OA) values for the entire period is also showed. The markers of PP1 and PP2 represent the two polluted periods while
CP1 and CP2 are clear periods, respectively.

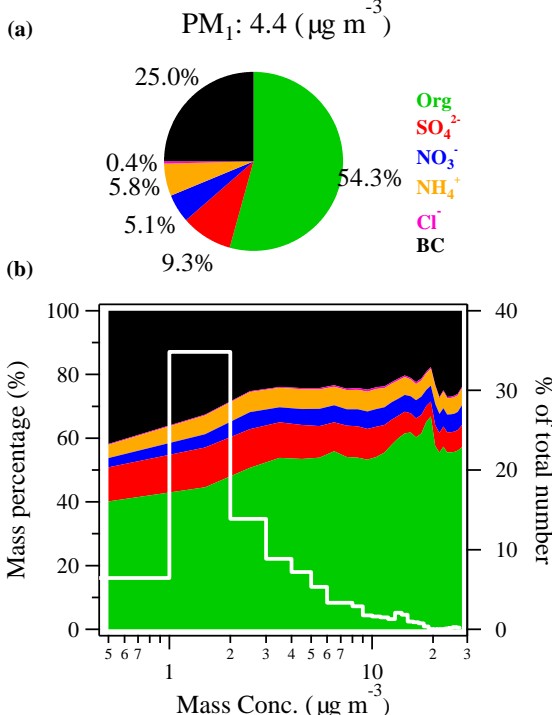

**Figure 3.** The average mass contributions of $PM_1$ (= $NR$-$PM_1$+ BC) species **(a)** during the entire sampling period and **(b)** as a function of the total $PM_1$ mass concentrations. The white solid line in **(b)** shows the percentage of the data number in each mass bins to the total data number.

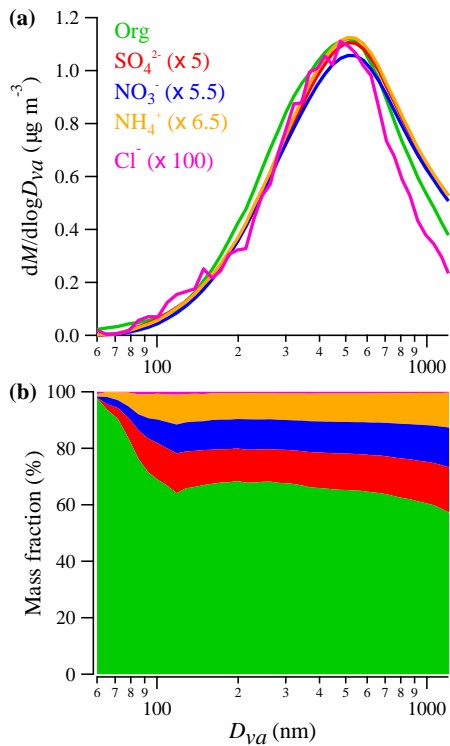

**Figure 4.** The average size distributions of **(a)** mass concentrations and **(b)** mass contributions of $NR$-$PM_1$ species for the entire study.

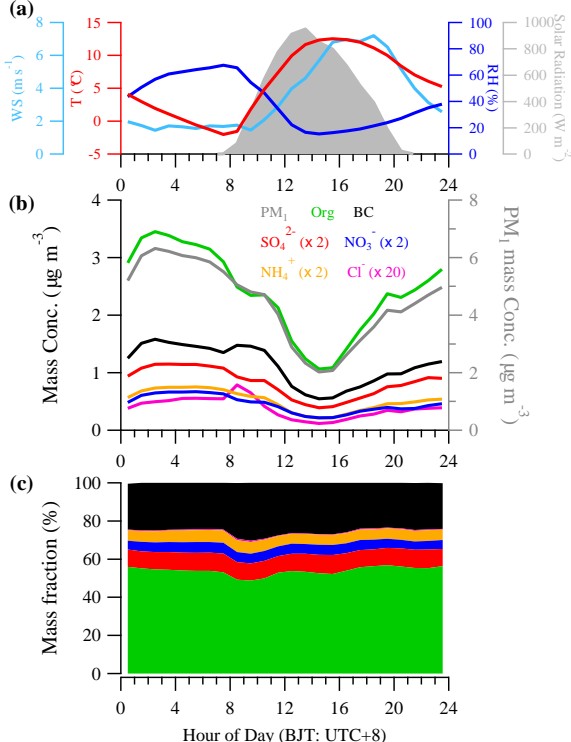

**Figure 5.** The diurnal cycles of **(a)** meteorological parameters (temperature, RH, wind speed, and solar radiation), **(b)** mass
concentrations and **(c)** mass contributions of PM$_1$ chemical species for the entire study.

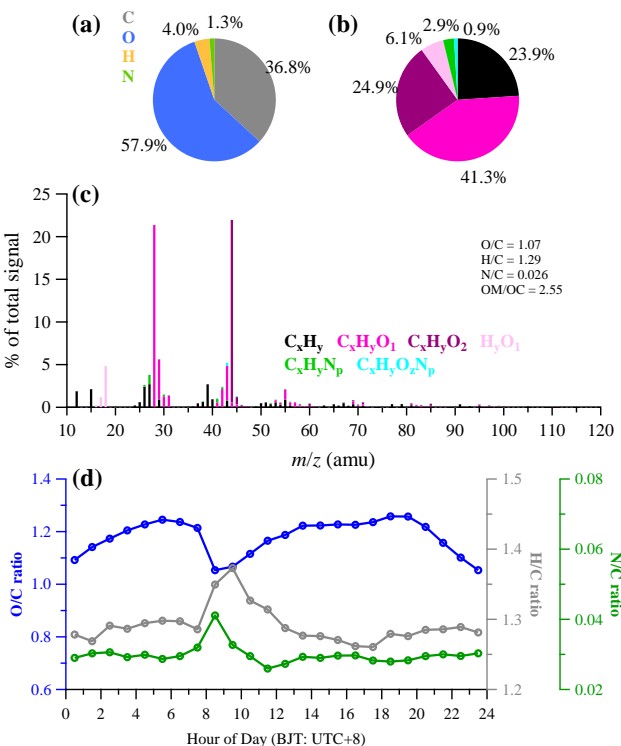

**Figure 6.** The average contributions of **(a)** four elements (C, O, H, and N) and **(b)** six ion categories (colors as in **(c)**) to OA for
the entire study; **(c)** the average high-resolution mass spectrum of OA (colors show six ion categories); **(d)** the diurnal variations
of O/C, H/C, and N/C ratios.

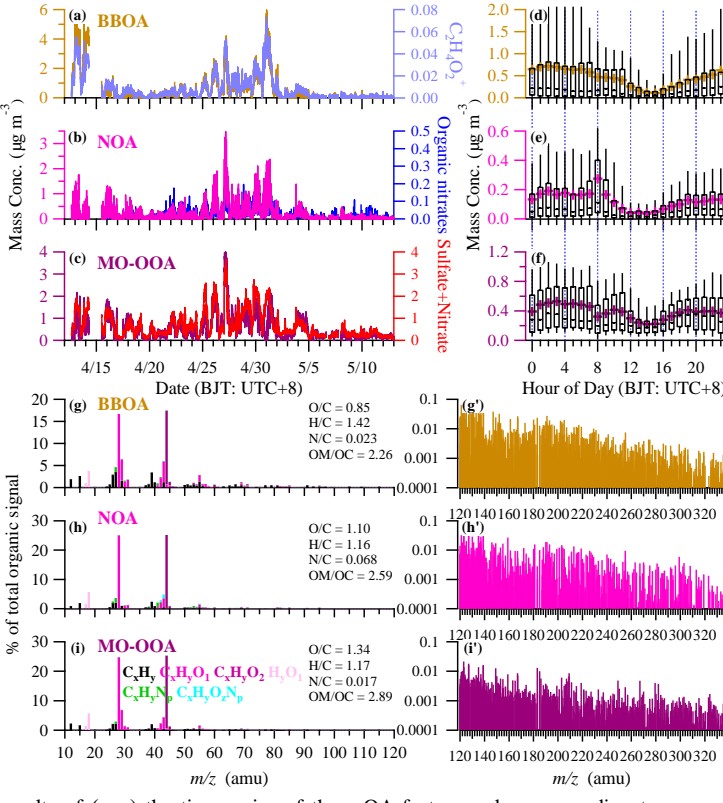

**Figure 7.** The PMF results of **(a–c)** the time series of three OA factors and corresponding tracer species, **(d-f)** the diurnal
variations of the mass concentrations of the three OA factors (the whiskers above and below the boxes indicate the 90th and 10th
percentiles, the upper and lower boundaries respectively indicate the 75th and 25th percentiles, the lines in the boxes indicate the
median values, and the cross symbols indicate the mean values), **(g-i)** high-resolution mass spectra of the three OA factors
colored by six ion families at *m/z* < 120, and **(g'-i')** the unit resolution mass spectra at *m/z* > 120 for each OA factor.

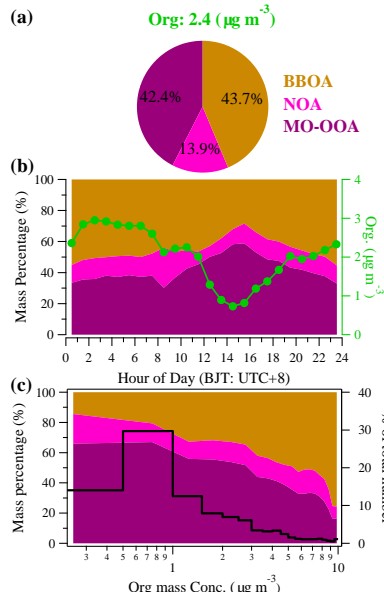

**Figure 8. (a)** The average mass concentration of OA and mass contributions of three OA factors to total OA; **(b)** the diurnal
variations of mass contributions of three OA factors to total OA and the total OA mass concentration; **(c)** The average mass
contributions of three OA factors as a function of total OA mass concentrations. The black solid line in **(c)** shows the percentage
of the data number in each OA mass bins to the total data number.

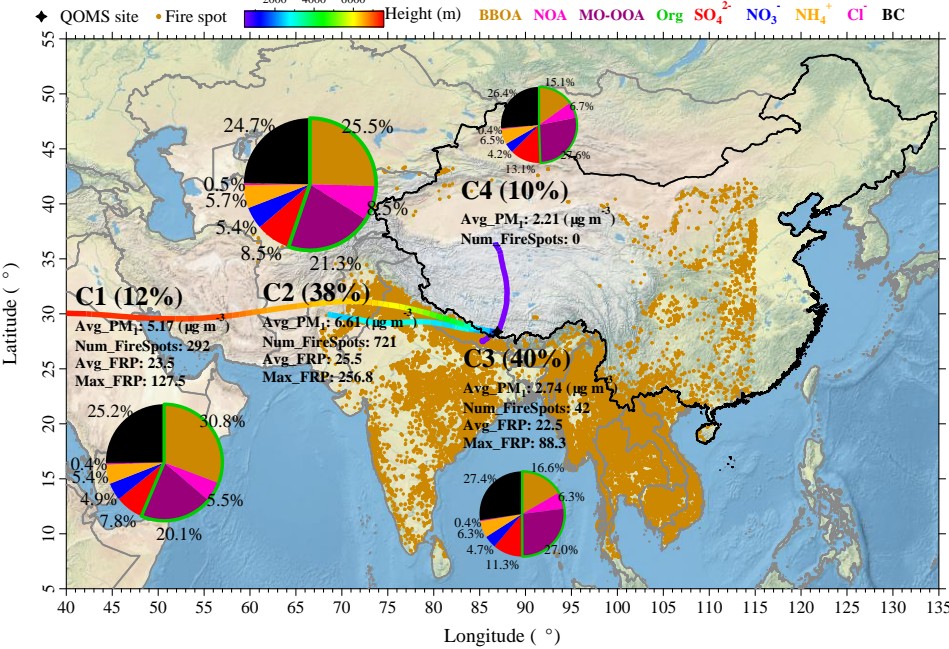

**Figure 9.** The average back trajectory clusters during the entire study and the corresponding mass contributions of PM₁ species
and OA factors to the total PM₁ mass. The areas of each pie charts are scaled by the corresponding average PM₁ mass
concentrations. The average PM₁ mass concentrations, number of fire hotspots as well as the average and maximum fire radiative
powers (FRP) belong to each clusters during the entire measurement period are also given.

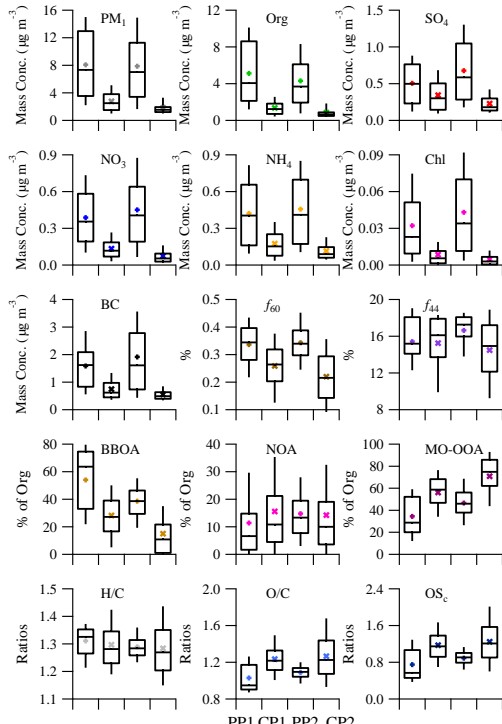

**Figure 10.** Box plots of mass concentrations of total PM₁ and its species, $f_{60}$ and $f_{44}$ values, mass contribution of three OA
components to organics, element ratios (H/C and O/C), and carbon oxidation states (OS$_c$) among the four polluted and clear
periods. The whiskers indicate the 90th and 10th percentiles, the upper and lower boundaries of boxes indicate the 75th and 25th
percentiles, the lines in the boxes indicate the median values, and the markers indicate the mean values.

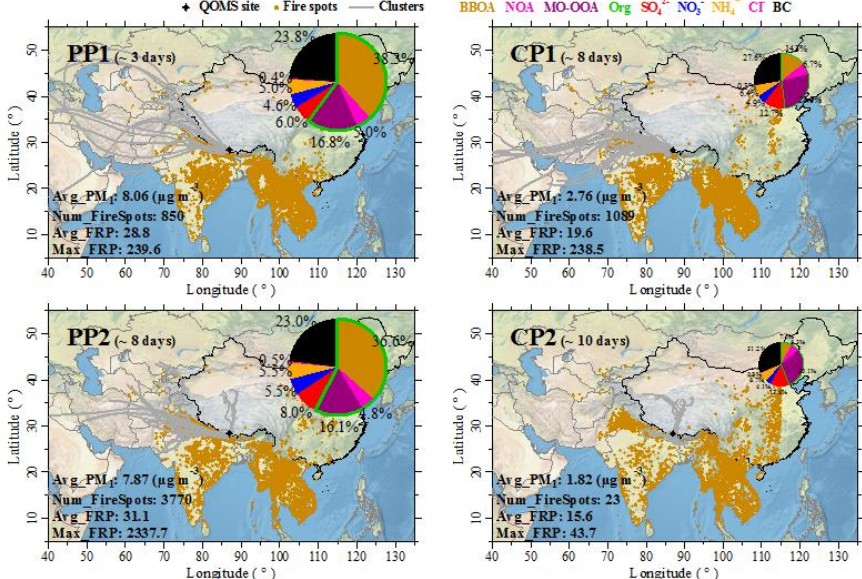

**Figure 11.** The 72-h back trajectories (grey solid lines) calculated every 6 h for the different episodes. Pie charts show the
average mass contributions of PM$_1$ species and OA factors to the total PM$_1$ mass for each episodes (scaled by the corresponding
average PM$_1$ mass concentrations). The average PM$_1$ mass concentrations, number of fire hotspots as well as the average and
maximum fire radiative powers (FRP) belong to all trajectories for the different episodes are also given.

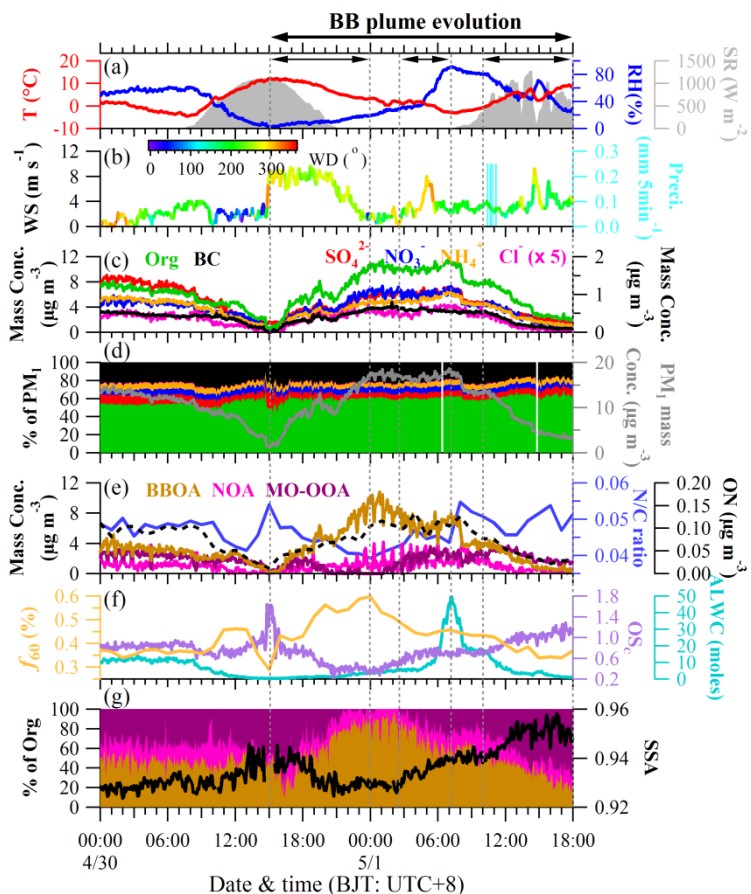

**Figure 12.** The temporal variations of meteorological parameters, mass concentrations and mass contributions of each PM$_1$
species and OA components as well as the N/C ratio, $f_{60}$ values, carbon oxidation states (OS$_c$), aerosol liquid water content
(ALWC) and single scattering albedo (SSA) for the case study period from April 30 at 00:00 to May 1 at 18:00.

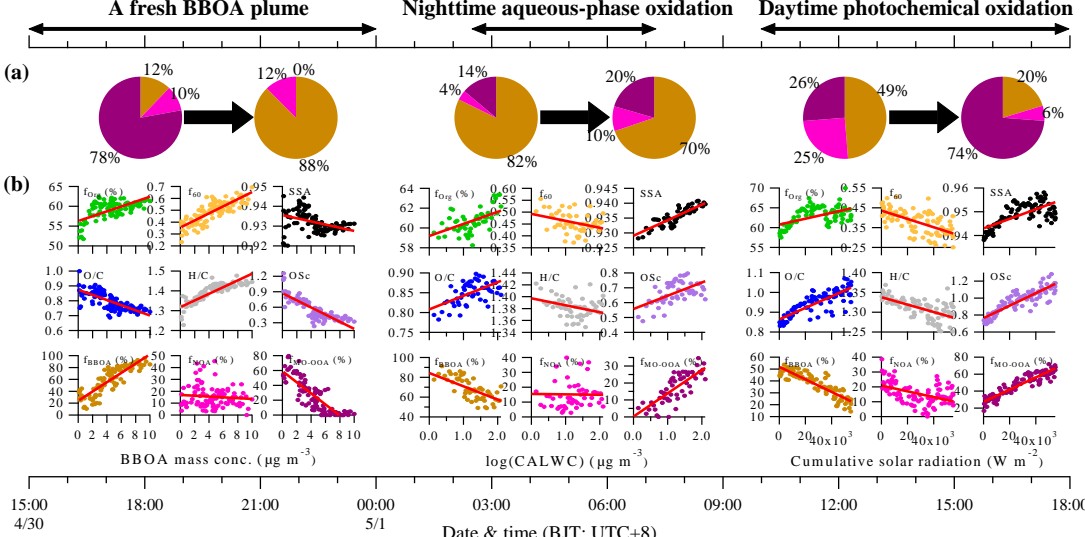

**Figure 13.** Case study of chemical evolution of BB plume from April 30 at 15:00 to May 1 at 18:00. The periods marked with
arrows are three distinct evolution processes. Pie charts in **(a)** are the mass contributions of three OA factors to total OA during
the beginning and end time for each process, respectively. The scattering plots in **(b)** are the aerosol chemistry parameters as a
function of BBOA mass concentration, logarithmic values of cumulative aerosol liquid water content (CALWC), and cumulative
solar radiation for the corresponding process.

# Chemical characterization of long-range transport biomass burning emissions to the Himalayas: insights from high-resolution aerosol mass spectrometry

**Xinghua Zhang[1,2,3], Jianzhong Xu[1], Shichang Kang[1], Yanmei Liu[1,3], Qi Zhang[4]**

[1]State Key Laboratory of Cryospheric Sciences, Northwest Institute of Eco-Environment and Resources, Chinese Academy of Sciences, Lanzhou 730000, China

[2]Key Laboratory of Arid Climatic Change and Reducing Disaster of Gansu Province, Key Laboratory of Arid Climatic Change and Disaster Reduction of CMA, Institute of Arid Meteorology, China Meteorological Administration, Lanzhou 730020, China

[3]University of Chinese Academy of Sciences, Beijing 100049, China

[4]Department of Environmental Toxicology, University of California, Davis, CA 95616, USA

*Correspondence to*: Jianzhong Xu (jzxu@lzb.ac.cn)

## Abstract

An intensive field measurement was conducted at a remote, background, and high-altitude site (Qomolangma station, QOMS, 4276 m a.s.l.) in the northern Himalayas, using an Aerodyne high-resolution time-of-flight aerosol mass spectrometer (HR-ToF-AMS) along with other collocated instruments. The field measurement was performed from April 12 to May 12, 2016 to chemically characterize the high time-resolved submicron particulate matter ($PM_1$) and obtain the dynamic processes (emissions, transport, and chemical evolution) of biomass burning (BB), frequently transported from South Asia to the Himalayas during pre-monsoon season. Overall, the average ($\pm 1\sigma$) $PM_1$ mass concentration was 4.44 ($\pm 4.54$) μg m$^{-3}$ for the entire study, comparable with those observed at other remote sites worldwide. Organic aerosol (OA) was the dominant $PM_1$ species (accounting for 54.3% of total $PM_1$ on average) followed by black carbon (BC) (25.0%), sulfate (9.3%), ammonium (5.8%), nitrate (5.1%), and chloride (0.4%). The average size distributions of $PM_1$ species all peaked at an overlapping accumulation mode (~ 500 nm), suggesting that aerosol particles were internally well-mixed and aged during long-range transport. Positive matrix factorization (PMF) analysis on the high-resolution organic mass spectra identified three distinct OA factors, including a BB-related OA (BBOA, 43.7%), a nitrogen-containing OA (NOA, 13.9%) and a more-oxidized oxygenated OA (MO-OOA, 42.4%). Two polluted episodes with enhanced $PM_1$ mass loadings and elevated BBOA contributions from the west and southwest of QOMS during the study were observed. A typical BB plume was investigated in detail to illustrate the chemical evolution of aerosol characteristics under distinct air mass origins, meteorological conditions and atmospheric oxidation processes.

## 1 Introduction

The Tibetan Plateau and Himalayas (TPH), generally called the "third pole", is the highest (average altitude of more than 4000 m a.s.l.) and largest (~ 2 500 000 km$^2$) plateau in the world. This region has been recognized as one of the most pristine region in the world due to its high altitude, sparse population and minor influence of anthropogenic activities (Yao et al., 2012a).

Consideration on the intense dynamical and thermal forcing effects, the TPH not only plays a key role in the formation of Asian monsoon systems, but also impacts the large-scale atmospheric circulation, hydrological cycle, as well as global climate (Duan and Wu, 2005; Wu et al., 2007). Over the past decades, more attentions have been paid to the environment and climate change in the TPH since this region is very susceptibility to the global climate change such as fast air temperature rise and dramatic glacier shrinkage (Xu et al., 2009; Kang et al., 2010; Yao et al., 2012b; Yang et al., 2014). Atmospheric environment in the TPH, albeit which is one of the most pristine region in the world, has been thought to be influenced variably due to the worse air pollution in its surrounding countries (Hou et al., 2003; Lau et al., 2008). For example, polluted air mass, particularly from South and Southeast Asia regions, had been observed frequently to transport to the Himalayas (Bonasoni et al., 2010; Cong et al., 2015), heat the aloft air masses over the TPH (Lau et al., 2006; Ramanathan and Carmichael, 2008) and decline the surface albedo after its deposition onto snow and glacier (Xu et al., 2009). As a consequence, characterizing the aerosol physicochemical properties in the TPH, including mass loading, chemical composition, size distribution and source, are of great importance to better understand the aerosol chemistry, estimate the aerosol radiative forcing, and finally evaluate the effect of polluted air mass on the ecology and environment in the TPH region.

Numerous aerosol measurements have been conducted in the TPH region in past decades to characterize the physicochemical properties, sources and transport pathways of ambient aerosol (Liu et al., 2008; Decesari et al., 2010; Marcq et al., 2010; Marinoni et al., 2013; Putero et al., 2014; Xu et al., 2017; Zhang et al., 2017a). South and Southeast Asia are two major polluted regions due to their intense biomass burning (BB) activities from natural forest fires and traditional human burning activities for residential heating and cooking (Engling et al., 2011; Yadav et al., 2017). The polluted feature of South and Southeast Asia during April 12 to May 12, 2016 can be further revealed by the distribution of average aerosol optical depth in Fig. 1. During the pre-monsoon period, atmospheric pollutants associated with BB emissions in South Asia are generally advected by regional and long-range transport (e.g., westerlies and South Asian monsoon system) to Himalayas and built up in the southern foothills, then pollutants are lifted up to high altitude by the Himalayan topography and the typical valley wind circulation (Zhao et al., 2013; Cong et al., 2015; Liu et al., 2017). However, the chemical properties of aerosol particles are still not well understood and limited in the Himalayas region due to its remote and harsh environments, challenging weather conditions and logistic difficulties. In addition, most of the available studies are mainly based on the off-line filter sampling of ambient aerosol or snow/ice samples following by laboratory analyses (Decesari et al., 2010; Ram et al., 2010; Li et al., 2016; Wan et al., 2017). These studies usually had a relatively low-time resolution (days to weeks). Therefore, real-time consecutive field measurement, especially focusing on the high-resolution size-resolved chemical characteristics of aerosol particles, is of great importance and necessary to give insight into the sources and the dynamic chemical evolution of ambient aerosol.

Online real-time instrument such as Aerodyne aerosol mass spectrometer (AMS), which can be used to characterize the chemical properties and sources of submicron aerosol particles with high time resolution and sensitivity, has been greatly developed and widely implemented worldwide (Canagaratna et al., 2007; Zhang et al., 2007a; Jimenez et al., 2009; Li et al., 2017). Although the deployments of the AMS in China have started since 2006, most of these studies in

China are conducted in urban areas, including Beijing−Tianjin−Hebei (Sun et al., 2013; Sun et al., 2016), Yangtze River Delta (Wang et al., 2016a; Wang et al., 2016b), Pearl River Delta regions (Huang et al., 2011), and Lanzhou (Xu et al., 2014; Xu et al., 2016; Zhang et al., 2017b) as shown in Fig. S1, whereas just few studies deployed in remote sites so far, such as Menyuan (Du et al., 2015), Mt. Yulong (Zheng et al., 2017), and Nam Co (Xu et al., 2017; Wang et al., 2017). In this paper, an Aerodyne high-resolution time-of-flight mass spectrometer (HR-ToF-AMS) was deployed at the Qomolangma Station for Atmospheric and Environmental Observation and Research (QOMS) in the north slope of the Himalayas to fill the vacancy of real-time mass spectrometer measurement at high elevation site and evaluate the significant impacts of BBs from polluted areas in the South Asia on the TPH aerosol properties during the pre-monsoon season. Here, we report an overview of the 5-min real-time chemical and physical characteristics of submicron aerosols ($PM_1$), including mass loading, composition, size distribution, acidity as well as temporal and diurnal variations. The sources of organic aerosols (OA) are also investigated using positive matrix factorization analysis on the high-resolution OA mass spectrum. BB influence and chemical evolution of aerosols in polluted plume are examined via combining back trajectory analysis of air masses and fire hotspots information, respectively.

## 2 Experimental methods

### 2.1 Sampling site

The QOMS (28.36 °N, 86.95 °E, 4276 m a.s.l.; Fig. 1), which is located in the northern slope of Mt. Everest (~ 30 km away), was established for atmospheric and environmental observation since 2005 (Ma et al., 2008). The geomorphic and climate features around the QOMS are typical alpine cold and arid areas covered by sandy soil with sparse vegetation. The QOMS is located in a long river valley and isolated from residential areas due to its harsh environment with a small village (with a population of ~ 300) to the south (~ 10 km). The closest town, Dingri County, is ~ 100 km south from the QOMS. A freeway is located at the front of the QOMS for tourism with increased tourist during summer. The measurements were conducted from April 12 to May 12, 2016. Since this period was within the typical pre-monsoon season of the TPH, the large-scale atmospheric circulation pattern was dominated by westerly or southwesterly winds with limited precipitation. Owning to a distinct thermal forcing from the southern mountains and glaciers, the QOMS was locally dominated by strongly mountain-valley circulation with down-slope wind prevailing during the daytime, especially in the afternoon (Fig. 1c and S2) (Zou et al., 2008), which would make the valley as an efficient channel for the down transport of air mass form high-altitude troposphere.

### 2.2 Instrumentation

A suite of real-time instruments were co-located to measure the physiochemical properties of fine particles at the QOMS, including an Aerodyne HR-ToF-AMS (Aerodyne Research Inc., Billerica, MA, USA) for 5-min size-resolved chemical compositions (organics, sulfate, nitrate, ammonium, and chloride) of non-refractory submicron particulate matter ($NR-PM_1$), a scanning mobility particle sizer (SMPS, model 3936, TSI Inc., Shoreview, MN, USA) for 5-min particle number concentration and size distribution between 14.6 and 661.2 nm in mobility diameter ($D_m$), and a photoacoustic extinctiometer (PAX, DMT Inc., Boulder, CO, USA) for particle light absorption

and scattering coefficient ($b_{abs}$ and $b_{scat}$) at 405 nm and further deriving black carbon (BC) mass
concentration. All instruments were placed in an air-conditioned room with temperature
maintaining at ~ 20 ℃. Ambient aerosol particles were introduced through a 0.5 inch copper tube
which stemmed out of the rooftop by about 1.5 m. A PM$_{2.5}$ cyclone (model URG-2000-30EH,
URG Corp., Chapel Hill, NC, USA) was used in front of the sampling inlet for removing coarse
particles with size cutoffs of 2.5 μm in aerodynamic diameter ($D_{va}$). A diffusion dryer was placed
following the cyclone to dry the ambient air and eliminate potential humidity effect on particles.
The total length of the sampling line was about 5 m and the retention time of particles was less
than 2.5 s in the whole inlet. The total air flow rate from the sampling inlet was about 10 L min$^{-1}$,
with part of flow shared by the HR-ToF-AMS and the SMPS while the remaining flow exhausted
by an external pump. The meteorology data including wind speed (WS), wind direction (WD),
relative humidity (RH), temperature ($T$), and solar radiation (SR) during this study were obtained
from a Vantage Pro2 weather station (Davis Instruments Corp., Hayward, CA, USA). Note that all
the date and time used in this study are reported in Beijing Time (BJT: UTC + 8 h).
**2.3 HR-ToF-AMS operation and data analysis**
**2.3.1 HR-ToF-AMS operation**
A detailed instrumental description of the Aerodyne HR-ToF-AMS can be found elsewhere
(DeCarlo et al., 2006) and only a brief summary is provided here. Briefly, the HR-ToF-AMS
consists of three main parts: an aerosol sampling inlet, a particle sizing vacuum chamber, and a
particle composition detection section (Jimenez et al., 2003). Ambient particles are sampled into
the instrument through a critical orifice (130 μm in this study for enhancing the transmission
efficiency at the high-altitude area) and focus into a concentrated and narrow beam through an
aerodynamic lens. Then particles are accelerated into the sizing vacuum chamber and obtain
different velocities for particles with different sizes due to the supersonic expansion induced by
different pressure between the two chambers. Meanwhile, a mechanical chopper with two radial
slits located 180 °apart is used to intercept the focused particle, and then the time of flight (P-ToF)
from the chopper to the vaporizer is measured to obtain the aerodynamic size of particles. After
passing through the sizing chamber, particles are directed onto a resistively heated surface (~
600 ℃) under a high vacuum and ionized by a 70 eV electron impact, and finally detect by the
high-resolution time-of-flight mass spectrometer. In this study, the HR-ToF-AMS was only
toggled under the high sensitive V-mode (detection limits ~ 10 ng m$^{-3}$). Under the V-mode
operation, the instrument also switched between the mass spectrum (MS) mode and the particle P-
ToF mode every 15 s, spending 6 and 9 s on each, to obtain the mass concentrations and size
distributions of the non-refractory species, respectively.
The HR-ToF-AMS was calibrated for ionization efficiency (IE) and particle sizing at the
beginning, in the middle, and at the end of this study according to the standard protocols (Jayne et
al., 2000). Both the calibrations of IE and particle sizing were performed using mono-dispersed
ammonium nitrate particles with nominal diameters of 70–300 nm. Default relative ionization
efficiency (RIE) values were assumed in this study as 1.1 for nitrate, 1.3 for chloride, and 1.4 for
organics. The RIE values of 3.9 and 4.2 were used for ammonium based on the results of two IE
calibrations at the beginning and in the middle of this study, while RIE values of 1.6 and 1.4 were
determined similarly for sulfate by using mono-dispersed ammonium sulfate particles,
respectively.

**2.3.2 HR-ToF-AMS data analysis**

The mass concentrations and size distributions of NR-PM$_1$ species and the ion-speciated mass
spectra, composition and elemental composition of organics were determined from the HR-ToF-
AMS data by using the standard ToF-AMS analysis toolkit SQUIRREL (v1.56) and PIKA (v1.15c)
modules written in Igor Pro (Wavemetrics Inc., Lake Oswego, OR, USA). An empirical particle
collection efficiency (CE) of 0.5 was used to compensate for the incomplete transmission and
detection of particles due to particle bouncing at the vaporizer and partial transmission through the
aerodynamic lens, which has been widely used in field studies employing AMS with a dryer
installed in front of the inlet (Xu et al., 2014; Xu et al., 2016). The elemental ratios of oxygen-to-
carbon (O/C), hydrogen-to-carbon (H/C), nitrogen-to-carbon (N/C), and organic mass-to-organic
carbon (OM/OC) for this study were determined using the "improved-ambient" method (referred
as I-A method) (Canagaratna et al., 2015), which increased O/C on average by 34%, H/C on
average by 15%, and OM/OC on average by 17% (Fig. S3) compared with those determined from
the "Aiken ambient" method (referred as A-A method) (Aiken et al., 2008).
Positive matrix factorization (PMF) analysis using the PMF2.exe algorithm (v4.2) (Paatero
and Tapper, 1994) in robust mode was conducted on the high resolution mass spectra (HRMS) to
determine distinct OA components in this study. The analysis was performed using an Igor Pro-
based PMF Evaluation Tool (PET, v2.03) (Ulbrich et al., 2009), downloaded from the webpage
(http://cires.colorado.edu/jimenez-group/wiki/index.php/PMF-AMS_Analysis_Guide). The data
and error matrices input into the PMF analysis were generated from analyzing the V-mode data via
PIKA fitting. Detailed PMF analysis was thoroughly evaluated following the procedures
summarized in Table 1 of Zhang et al. (2011). Isotopic ions were generally excluded and the four
ions of O$^+$, HO$^+$, H$_2$O$^+$, and CO$^+$ were downweighted in PMF analysis, because they were
determined according to the relationship with CO$_2^+$ signal (Ulbrich et al., 2009). The "bad" ions
with $S/N$ less than 0.2 were removed from the HRMS data and error matrices before PMF analysis,
and "weak" ions with $S/N$ between 0.2 and 2 were downweighted by increasing their errors. In
addition, some runs with huge mass loading spikes were also removed from the data and error
matrices. The detailed matrix preparation and data pretreatment can also refer to Xu et al. (2014).
A summary of key diagnostic plots of the PMF results for this study is presented in Fig. S4.
Overall, the PMF solutions were investigated for 1 to 8 factors and for the rotational parameter
(fPeak) varying from –1 to 1 with a step of 0.1. Besides examining the model residuals, scaled
residuals, and the Q/Q$_{exp}$ contributions for each $m/z$ and time following procedures detailed in
Table 1 of Zhang et al. (2011), the optimum solution can also be evaluated via comparing the mass
spectra of individual factors with reference spectra from specific sources or other ambient AMS
measurements, comparing the time series of individual factors with the known external tracers,
and analyzing the diurnal variations of individual factors. Finally, the 3-factor solution with fPeak
= 0 was chosen in this work. The direct comparisons of the mass spectra, time series, and diurnal
variations for 2-factor and 4-factor solution were also shown in Fig. S5 and S6, respectively. The
2-factor solution does not resolve the small, yet distinct nitrogen-containing OA, while the 4-
factor solution shows a splitting factor from the BB OA resolved in the 3-factor solution and
seems just like a simple separation of the two BB polluted episodes.

## 2.4 Other relevant data

The Hybrid Single Particle Lagrangian Integrated Trajectory (HYSPLIT4) model developed by the National Oceanic and Atmospheric Administration (NOAA) (Draxler and Rolph, 2003) was used to investigate the origins of air masses in this study, using the meteorological data from the NOAA Global Data Assimilation System (GDAS). The back trajectories were calculated every 6 h at an ending height of 500 m above ground level at the QOMS during the entire campaign, and then clustered them according to their similarity in spatial distribution. Finally, a four-cluster solution was adopted according to its small total spatial variance.

Aerosol optical depth (AOD) at 550 nm was derived from the observations made by National Aeronautics and Space Administration (NASA) Moderate Resolution Imaging Spectroradiometer (MODIS) onboard the Terra satellite. The distribution of average aerosol optical depth (AOD) in a large range areas (20 °–45 °N, 60 °–110 °E) around the TPH during the entire period of this study is given in Fig. 1d.

Various active fire hotspots were detected over South and Southeast Asia by the Fire Information for Resource Management System (FIRMS) provided by MODIS satellite (https://firms.modaps.eosdis.nasa.gov), demonstrating the possibility that active wildfires or BBs from South and Southeast Asia may have significant impacts on the air conditions in the TPH region.

The aerosol liquid water content (ALWC) was estimated with the Extended AIM (E-AIM) Aerosol Thermodynamics Model (http://www.aim.env.uea.ac.uk/aim/aim.php). The input data included the concentrations of sulfate, nitrate, ammonium, and chloride measured by the HR-ToF-AMS as well as the relative humidity (RH) and temperature of ambient air.

## 3 Results and discussion

## 3.1 Overview of the study

### 3.1.1 Meteorological conditions

The measurement period in our study was within the typical pre-monsoon season of the TPH. The meteorological conditions were therefore characterized by a relatively cold, dry and windy weather, and the westerlies dominated the large-scale atmospheric circulation patterns with little precipitation, as displayed in Fig. 2. During the study, the averaged diurnal air temperature ranged from −2.0 to 12.5 °C with an average ($\pm 1\sigma$) of 5.7 ($\pm 5.0$) °C, and the RH ranged from 15.3 to 67.5% with an average of 39.8 $\pm 18.8$%. Only two light precipitation events (1 and 0.5 mm d$^{-1}$) occurred on 1 and 8 May, respectively. The WDs at QOMS were predominantly by southwesterly, which were mainly associated with the thermally driven mountain-valley winds and glacier winds (Zou et al., 2008). For the diurnal variation of wind conditions, a nearly calm wind period (hourly average WS less than 2 m s$^{-1}$) was observed in the early morning time; after sunrise to noon time, there was a weak up-slope wind period (from the north); the diurnal wind cycles in the rest time were dominated by the down-slope wind (from the southwest) with the maximum value of hourly average WS up to 7 m s$^{-1}$ (Fig. 2b and S2).

### 3.1.2 Inter-comparisons between different instruments

An inter-comparison of the total $PM_1$ (NR-$PM_1$ + BC) mass concentrations measured by the HR-
ToF-AMS (CE = 0.5) and the PAX with particle volumes (assuming spherical particles)
determined from the SMPS is shown in Fig. S7. Overall, the $PM_1$ mass is closely correlated ($R^2$ =
0.97) with that of SMPS particle volume during the entire campaign, with a linear regression slope
of 2.86. This slope is significantly higher than the estimated average $PM_1$ density of 1.44 g cm$^{-3}$,
which is calculated based on the measured particle compositions in this study and the assumed
particle densities of 1.2 for organics, 1.78 for $(NH_4)_2SO_4$, 1.72 for $NH_4NO_3$, 1.52 for $NH_4Cl$ and
1.8 g cm$^{-3}$ for BC (Zhang et al., 2005b; Xu et al., 2016). This discrepancy is likely introduced by
various factors, including different transmission sizes between HR-ToF-AMS and SMPS (up to ~
1.0 μm in $D_{va}$ for AMS vs. limited size range of 14.6−661.2 nm in $D_m$ for SMPS), rough
calculation of $PM_1$ density using assumed composition densities and spherical shape without
consideration the particle porosity, as well as the using of empirical and constant CE value of 0.5
in this study. This phenomenon was also observed at other sites in previous studies (Ge et al., 2012;
Huang et al., 2012; Xu et al., 2014; Du et al., 2015).

**3.1.3 Mass concentration and chemical composition of $PM_1$**

As shown in Fig. 2, the mass concentrations of $PM_1$ and all $PM_1$ species, as well as their mass
fractions in $PM_1$ varied dynamically throughout this study. Two polluted periods (PP1 and PP2)
were identified according to their high $PM_1$ mass concentrations (daily average $PM_1$ mass is larger
than 5 μg m$^{-3}$), high contributions from BBOA and unique back trajectories. The rest periods
characterized by low $PM_1$ mass concentrations were considered as clear periods (CP1 and CP2).
The 5-min total $PM_1$ mass concentration ranged from 0.18 to 27.97 μg m$^{-3}$ for the study, with an
average ($\pm 1\sigma$) value of 4.44 $\pm$ 4.54 μg m$^{-3}$. This average value was more than two times lower
than most of the $PM_1$ mass concentrations measured with Aerodyne AMS or aerosol chemical
speciation monitor (ACSM) instruments at various urban, suburban, rural or background sites in
China (10.9−138.8 μg m$^{-3}$) (Fig. S1), except slightly lower than that at Mt. Yulong (5.7 μg m$^{-3}$)
located at the southeastern edge of the TPH, whereas higher than that at Nam Co Station (2.0 μg
m$^{-3}$) located in the central of the TPH. Moreover, as shown in table S1, the $PM_1$ mass
concentration in this study was also lower than those measured at the three remote island sites in
Asia which were frequently influenced by outflow from China, Korea and Japan (i.e., 7.9 μg m$^{-3}$
for Okinawa island, 12.0 μg m$^{-3}$ for Fukue island in Japan, and 10.7 μg m$^{-3}$ for Jeju island in
Korea) (Takami et al., 2005; Jimenez et al., 2009), as well as the $PM_1$ mass concentration (15.1 μg
m$^{-3}$) obtained at the Bachelor mountain in United States which was heavily impacted by wildfire
smoke plumes (Zhou et al., 2017). However, it was higher than those reported at other coastal,
high elevation, forest or remote background sites in North America and Europe (0.55−2.91 μg m$^{-3}$)
(Zhang et al., 2007a; Sun et al., 2009; Fröhlich et al., 2015). Although these measurements
mentioned above were conducted at various sites worldwide during different seasons, these
comparisons further demonstrate that QOMS is a typical high elevation and remote background
site in Asia.
Overall, organics and BC were the two dominant $PM_1$ species (averagely contributed 54.3%
and 25.0% to the total $PM_1$ mass, respectively) followed by sulfate (9.3%), ammonium (5.8%),
nitrate (5.1%), and chloride (0.4%) (Fig. 3a). The high contributions of organics and BC at QOMS
were significantly associated with the active BB emissions by long-range transport from polluted

areas in South Asia. Organic compounds and BC have been revealed as two dominant components of BB aerosols and generally used to identify BB events in previous studies (Bond et al., 2004; Bougiatioti et al., 2014). In addition, biomass burning at high elevation regions of Himalayas and south Asia was more incomplete burning and could emit amount of BC. This conclusion can be further revealed by their enhanced mass concentrations and contributions, especially for organics, during the two distinct polluted episodes influenced by active BB plumes. Figure 3b showed the mass contributions of $PM_1$ species as a function of total $PM_1$ mass concentrations. The $PM_1$ mass loadings in this study were mostly below 6 μg m$^{-3}$ (accounted for ~ 77%); The mass contribution of organics increased significantly with the increase of total $PM_1$ mass loading whereas the rest species showed relatively stable or decrease trends, suggesting the dominant contributions of organics in the polluted episodes at QOMS.

### 3.1.4 Acidity and size distributions of submicron aerosols

To evaluate the bulk acidity of NR-$PM_1$ in this study, we calculated the $NH_4^+$ concentration ($NH_{4\,calc}^+$) based on the mass concentrations of sulfate, nitrate and chloride measured by the HR-ToF-AMS and assumed full neutralization of these anions by ammonium (Zhang et al., 2007b). The scatter plot of the measured $NH_4^+$ ($NH_{4\,meas}^+$) concentration versus the $NH_{4\,calc}^+$ concentration for the entire campaign was shown in Fig. S7. A tight correlation ($R^2 = 0.97$) existed between $NH_{4\,meas}^+$ and $NH_{4\,calc}^+$ with a linear regression slope of 1.2, indicating that there were excess of ammonium in the submicron particle. This slightly high $NH_{4\,meas}^+/NH_{4\,calc}^+$ ratio was quite different with those results from various urban and rural sites in China, where bulk aerosols were overall neutralized or acidic due to the enrich gaseous precursors of $SO_2$ and $NO_x$ that could be further oxidized to sulfate and nitrate (Sun et al., 2013; Xu et al., 2014; Du et al., 2015; Zhang et al., 2017b). The excess ammonium at QOMS might relate to the important contributions of organic acids in this area (Cong et al., 2015), which could underestimate the $NH_{4\,calc}^+$ due to the neglect of organic acids in the ion-balance calculation, and the non-negligible contributions of nitrogen-containing organic compounds to $NH_x^+$ which finally overestimated the $NH_{4\,meas}^+$ (Sun et al., 2009; Ge et al., 2012). As mentioned above, atmospheric aerosols in the TPH region were significantly influenced by BB emissions from South Asia during the sampling periods. BBs would emit large amounts of nitrogen-containing organic compounds (Fleming et al., 2017; Zhou et al., 2017) and as discussed in section 3.2.

Figure 4 shows the average size distributions of NR-$PM_1$ species and their mass contributions as the function of size distribution. Overall, all chemical species showed a nearly consistent but narrow accumulation mode peaking at ~ 500 nm in $D_{va}$, indicating the well internal-mixed and aged aerosol particles at QOMS. Ultrafine particles (particles with diameter less than 100 nm) were dominated by organics (more than 70%), while the mass contributions of chemical species at the major peak (~ 500 nm) were organics (~ 65%), sulfate (~ 13%), nitrate (~ 11%), ammonium (~ 10%), and chloride (~ 1%). The contribution of organics decreased with the increase of size mode, while the contributions of three major inorganic species (sulfate, nitrate and ammonium) slightly increased with the increasing sizes (Fig. 4b).

### 3.1.5 Diurnal variations of chemical species

The average diurnal cycles of meteorological parameters as well as the $PM_1$ species and their mass fractions for the entire campaign were shown in Fig. 5. All $PM_1$ species presented a similar diurnal

pattern with lower concentrations in the daytime whereas higher concentrations in the nighttime.
The mass concentrations reached the minimum values at around 15:00. This pattern was
accompanied with the enhanced wind speed and the increased air temperature in the afternoon
which could related with the dynamics of planetary boundary layer (PBL). After that, the mass
concentrations began to build up and reached to high levels in the nighttime. Note that the mass
concentrations of chloride and BC also existed a slight peak during the early morning, which
corresponded with the calm wind conditions and the lowest air temperature of the day and could
associated with the enhanced local emissions at QOMS in the morning. The diurnal cycles of mass
contributions of each $PM_1$ species were relatively stable for the entire campaign, besides the slight
increase of BC from 24% at ~ 08:00 to 30% at ~ 10:00. Overall, organics dominated $PM_1$
throughout the day (49−57%), followed by BC (23−30%), sulfate (9−10%), ammonium (5−6%),
nitrate (4−6%), and chloride (0.3−0.8%).

## 3.2 Bulk characteristics of OA

Figure 6a and b showed the average mass contributions of the four elements and the six ion
categories to total organics, respectively. The organic mass was on average composed of 36.8%
carbon, 57.9% oxygen, 4.0% hydrogen, and 1.3% nitrogen. For ionic categories, $C_xH_yO_1^+$ ions
dominated the total OA accounting for 41.3%, followed by $C_xH_yO_2^+$ (24.9%), $C_xH_y^+$ (23.9%),
$H_yO_1^+$ (6.1%), $C_xH_yN_p^+$ (2.9%) and $C_xH_yO_zN_p^+$ (0.9%). The contributions of oxygen and the two
major oxygenated ion fragments ($C_xH_yO_z^+ = C_xH_yO_1^+ + C_xH_yO_2^+$) at QOMS were quite higher than
those obtained at other urban or rural sites in China, whereas carbon and $C_xH_y^+$ ions had relative
lower contributions, e.g., 38% of $C_xH_yO_z^+$ and 21% of oxygen versus 56% of $C_xH_y^+$ and 70% of
carbon in urban Lanzhou (Xu et al., 2014), and 37.4% of $C_xH_yO_z^+$ versus 51.2% of $C_xH_y^+$ in urban
Nanjing (Wang et al., 2016a), suggesting that OA at QOMS were highly aged. Correspondingly,
the average high-resolution OA mass spectrum (Fig. 6c) also showed significantly high
contribution (~ 25%) at $m/z$ 44 signal (one of the most reliable marker of oxygenated OA)
compared with other ion fragments, e.g., 5% at $m/z$ 43 (indicator for less oxidized compounds),
1.7% at $m/z$ 55 (important COA fragment), and 0.4% at $m/z$ 57 (tracer for traffic-related emission)
(Alfarra et al., 2004; Zhang et al., 2005a). The average O/C ratio was 1.07 during this study, which
was much higher than those observed at various urban and rural sites in China using the I-A
method, e.g., 0.37 in Beijing (Sun et al., 2016), 0.36 in Lanzhou (Xu et al., 2016), 0.35 in Nanjing
(Wang et al., 2016a), and 0.65 in Ziyang (Hu et al., 2016)). Moreover, the average O/C ratio was
even higher than that of 0.98 at the background site of Mt. Wuzhi in southern China (Zhu et al.,
2016), indicating that OA at QOMS was more oxidized and aged during long-range transport. The
average H/C, N/C and OM/OC ratios were on average 1.29, 0.026 and 2.55 in this study,
determined a nominal chemical formula of OA as $C_1H_{1.29}O_{1.07}N_{0.026}$.
For the diurnal cycles, O/C ratio had two peaks in the early morning and late afternoon, likely
related to the production of secondary organic aerosol (SOA) via aqueous-phase reactions or
photochemical oxidation processes during these two periods. H/C and N/C ratios yet showed
inverse diurnal cycles with that of O/C, namely peaked at around 08:00−10:00 in the morning.
The Van Krevelen diagram (H/C versus O/C), which had been used widely to probe the oxidation
reaction mechanisms for bulk OA (Heald et al., 2010), showed an apparent anticorrelation ($R^2$ =
0.57) with a slope of −0.48 at QOMS (Fig. S8). Ng et al. (2011b) have suggested that a slope of −

0.5 indicate a net change in chemical composition from the addition of both acid and
alcohol/peroxide functional groups without fragmentation, and/or carboxylic acid groups with
fragmentation.

### 3.3 Organic aerosol source apportionment

Source apportionment via PMF analysis on the high-resolution OA mass spectrum identified three
distinct factors in this campaign according to their unique temporary variations, mass spectrum
(MS) profiles, element ratios, correlations with tracers, and diurnal patterns, i.e., a BB-related OA
(BBOA), a nitrogen-containing OA (NOA) and a more-oxidized oxygenated OA (MO-OOA).
Detailed discussion on each factor is given in the following subsections.

### 3.3.1 BBOA

Although significant high contribution at $m/z$ 44 (mostly $CO_2^+$) was found in all of the three OA
components, the BBOA MS was also characterized by contributions at $m/z$ 60 (mainly $C_2H_4O_2^+$)
and tiny $m/z$ 73 (mainly $C_3H_5O_2^+$) (Fig. 7g), which were generally regarded as well-known tracers
for BB emissions (Alfarra et al., 2007). The average fraction of the signal at $m/z$ 60 (referred as $f_{60}$)
in the BBOA mass spectrum was 0.61%, which was higher than the typical value of $\sim$ 0.3% in the
absence of BB impacts (Cubison et al., 2011). The time series of BBOA correlated tightly with
those of $C_2H_4O_2^+$ ($R^2 = 0.91$) and $C_3H_5O_2^+$ ($R^2 = 0.87$) as well as BC ($R^2 = 0.72$) and nitrate ($R^2 =$
0.75) (Fig. 7a and Table S2). If ignoring the influence of high contribution at $m/z$ 44, the BBOA
mass spectrum in this study correlated well ($R^2 = 0.5$–$0.9$) with those BBOA mass spectrum
identified at other sites worldwide (Ng et al., 2011a; Mohr et al., 2012; Saarikoski et al., 2012;
Crippa et al., 2013; Crippa et al., 2014; Xu et al., 2016), as shown in Fig. S9. The average mass
concentration of BBOA was 1.05 $\mu$g m$^{-3}$ for the entire study and contributed a large fraction
(43.7%) of the total OA mass on average (Fig. 8a), indicating that BBOA was an important
components of OA during the pre-monsoon season at the QOMS. The diurnal cycle of BBOA
showed high concentrations during nighttime whereas relatively low concentrations during
daytime (Fig. 7d). Correspondingly, the mass contributions of BBOA to total OA mass decreased
distinctly from $\sim$ 55% at 00:00 to 28% at 15:00 (Fig. 8b). In addition, higher mass concentrations
and contributions of BBOA were found during the two polluted episodes (PP1 and PP2) than those
during the clear periods, further indicating the important contribution of BBOA to OA in this
region. Figure 8c showed the mass fractions of the three OA components as a function of total OA
mass during the entire campaign. A continuously increased trend was found for the BBOA
contributions with the increasing OA mass, which contributed $\sim$ 15% when the total OA mass was
less than 0.3 $\mu$g m$^{-3}$, whereas it reached up to more than 75% with the OA mass increased to 9 $\mu$g
m$^{-3}$. This dominant contribution of BBOA during the polluted periods was consistent with those
results in previous studies that BB emission were an important source of aerosol to the southern
TPH (Engling et al., 2011; Xia et al., 2011; Putero et al., 2014; Cong et al., 2015). The O/C ratio
(0.85) of BBOA in this study was quite higher than those BBOA factors identified at other
urban/rural sites in previous studies (Aiken et al., 2009; Huang et al., 2011; Mohr et al., 2012; Sun
et al., 2016; Xu et al., 2016), suggesting its long-range transport feature. This aged BBOA feature
was similar with those obtained at other remote sites worldwide, such as a remote forest site in
Finland (Raatikainen et al., 2010), a remote background site in Greece (Bougiatioti et al., 2014),

and a national air quality background sites in southern China (Zhu et al., 2016), where OA were
generally highly oxidized.

### 3.3.2 NOA

Besides the two highest signals at $m/z$ 43.99 ($CO_2^+$) and 27.995 ($CO^+$) which together contributed
half of the total NOA signal due to the highly aged OA nature at QOMS, the NOA MS was also
characterized by some nitrogen-containing fragments, such as $m/z$ 27.011 ($CHN^+$), 41.027
($C_2H_3N^+$), and 43.006 ($CHON^+$). In total, these three fragments could comprise nearly half of the
nitrogen-containing signals in the NOA factor and finally contributed 5% of the total NOA signal.
The average O/C ratio of NOA for the entire campaign was 1.10 with the highest N/C ratio (0.068)
among the three OA components. This high N/C ratio at QOMS was comparable with those
nitrogen-containing OA factor identified in previous studies, such as 0.06 in Mexico City (Aiken
et al., 2009), 0.078 in Po Valley, Italy (Saarikoski et al., 2012), and 0.053 in New York (Sun et al.,
2012). The time series of NOA showed tightly correlation ($R^2 = 0.62$) with that of estimated
organic nitrates, whereas relatively weak correlations with $PM_1$ species and OA ions (Table S2). In
addition, the $f_{60}$ value (~ 0.37%) was also slightly higher than the background $f_{60}$ (0.3%) of BB
aerosols (Fig. 7h). These results together suggested that this oxygenated OA factor was likely a
nitrogen-containing OA and might be related to the aged BB emissions, consistent with the results
in previous studies that large amounts of nitrogen-containing organic compounds were found from
BB aerosols (Laskin et al., 2009; Gautam et al., 2016; Wang et al., 2017). In recently, Fleming et
al. (2017) found dung burning, a very popular activities in Himalayas and India for residential
cooking and heating, could emit much more nitrogen-containing OA than wood burning. Our filter
samples during high BBOA period analyzed by Fourier Transform Ion Cyclotron Resonance Mass
Spectrometry (FTICR-MS) also found amount of ON molecular (in preparation). As shown in Fig.
7e and 8b, both the diurnal cycles of mass concentrations and fractions of NOA had distinct
increase in the morning, similar with the diurnal patterns of chloride, element ratios of H/C and
N/C, and the estimated organic nitrates. This diurnal feature of NOA at QOMS was quite
consistent with those NOA factors identified in Po Valley, Italy (Saarikoski et al., 2012) and in
Mexico City (Aiken et al., 2009), or less-oxidized oxygenated OA (LO-OOA) in southeastern
USA (Xu et al., 2015) where have active BB emissions. NOA contributed ~ 14% of the total OA
mass on average, with an average mass concentration of 0.34 μg m$^{-3}$ for the entire study (Fig. 8a).

### 3.3.3 MO-OOA

An obvious more oxygenated OA factor was also identified in this study according to its
significant high signal at $m/z$ 44 (~ 25%) and the high average O/C ratio of 1.34 (Fig. 7i). The time
series of MO-OOA correlated closely ($R^2 = 0.7$) with sulfate and nitrate (Fig. 7c and Table S2).
Moreover, the mass spectrum of MO-OOA in this study resembled tightly to those more aged and
low-volatility oxygenated OA (LV-OOA) observed using AMS instruments at various sites
worldwide (Fig. S9), e.g., with $R^2$ of 0.89 and 0.97 to those in Lanzhou, China (Xu et al., 2014;
Zhang et al., 2017b), 0.96 to that in Paris, France (Crippa et al., 2013), 0.95 to that in Barcelona,
Spain (Mohr et al., 2012), as well as 0.70 and 0.71 to the standard LV-OOA mass spectrums
obtained from abundant AMS data sets by Ng et al. (2011a) and Crippa et al. (2014). The diurnal
variation of MO-OOA was mainly driven by the dynamic of PBL height, with high concentrations
during the nighttime yet relatively low concentrations during the daytime (Fig. 7f). This pattern

was quite different with those observed in previous studies that LV-OOA generally showed elevated concentrations during the afternoon in accordance with strong photochemical activities, suggesting that SOA at QOMS were mainly oxidized and aged during the long-range transport. On average, MO-OOA contributed by 42.4% of the total OA mass, with an average mass concentration of 1.02 μg m$^{-3}$ for the entire study (Fig. 8a). As shown in Fig. 2f and 8c, MO-OOA also displayed enhanced mass contributions during the clear periods, especially for period after May 2 when the average mass fraction of MO-OOA increased up to ~ 68% of the total OA mass.

**3.4 Impact of BB emissions on aerosol characteristics**

**3.4.1 Sources of BB aerosols**

In order to understand the transport pathways and the potential source areas of aerosol, 3-day back trajectories of air mass were calculated at an ending height of 500 m above ground level every 6 h at the QOMS from April 12 to May 12, 2016. A four-cluster solution and the wildfire hotspots around the QOMS during the entire measurement period were presented in Fig. 9. Cluster 1 and 2 (C1 and C2), which originated from the west of the QOMS and passed over many hotspot areas (e.g., Indo-Gangetic Plain and Nepal), represented two polluted clusters. On the contrary, C3 and C4, which accounted for half of the total back trajectories, were identified as clear clusters. C3 traveled a short distance from the southwest of the QOMS, whereas C4 was from the north of the QOMS and passed over the inland of the TPH. The average PM$_1$ mass concentrations for C1 and C2 were 5.17 and 6.61 μg m$^{-3}$, respectively, which were 2−3 times higher than those for the two clear clusters (2.74 and 2.21 μg m$^{-3}$). The mass contributions of OA and BBOA during C1 and C2 were up to more than 55% and 25% of the total PM$_1$ mass on average, whereas weak contributions were found for the clear clusters (C3 and C4), indicating the significant impacts of BB emissions from South Asia on aerosol loadings at QOMS.

**3.4.2 Comparison of aerosol characteristics and air mass origins during different episodes**

As shown in Fig. 2, the mass concentrations and compositions of PM$_1$ varied dynamically during the entire sampling period. Two polluted periods (PP1 and PP2) and two clear periods (CP1 and CP2) were identified. The comparisons of average mass concentrations and other indicators for the four different episodes were presented in box plots in Fig. 10, whereas the corresponding back trajectories of air masses and MODIS fire hotspots belong to each episode period were given in Fig. 11, respectively.

During the two polluted periods, PM$_1$ mass concentrations were much higher than those in clear periods (8.06 and 7.87 μg m$^{-3}$ for PP1 and PP2 vs. 2.76 and 1.82 μg m$^{-3}$ for CP1 and CP2; similarly hereinafter), with higher contributions from OA (60.1% and 57.5% vs. 48.1% and 43.9%) and BBOA (38.3% and 36.6% vs. 14.3% and 7.5%) (Fig. 10 and 11). In addition, $f_{60}$ were also higher during polluted periods than those for clear periods (0.34% and 0.34% vs. 0.26% and 0.22% on average) (Fig. 10). Air masses during PP1 and PP2 generally originated form long-range transport to the west of the QOMS, which would pass through intense wildfires areas in South Asia (e.g., Indo-Gangetic Plain and Nepal where showed high AOD values in Fig. 1d and active fire hotspots in Fig. 9 and 11). The fire hotspot number around the air mass trajectories during PP2 was more than three times higher than those during other periods. Although the hotspot number around the air mass trajectories during PP1 was not as abundant as that during PP2 and even

slightly lower than that during CP1, it was just collected within 3 days for PP1 whereas 8-10 days for another periods. Hence, the BB activities were also more frequent and intense during the short PP1 and finally resulted in the highest average $PM_1$ mass concentration among these periods. Back trajectories in CP1 also originated from the west of QOMS and passed over the northern India and Nepal, however, both the intensity of fire hotspot number (1089 hotspots in ~ 8 days) and average FRP (19.6) were obvious lower than that in PP2. CP2 was the most clear period, of which average $PM_1$ mass concentration was more than four times lower than those in polluted periods. Back trajectories during CP2 period were from either the north of QOMS which passed over inland areas of the TPH or the south of QOMS with quite short distance and low WS. These results together suggested the significant roles of air mass sources and BB emissions to aerosol characteristics at QOMS.

### 3.4.3 Case study on the chemical evolution of BB emission aerosols

In order to examine how atmospheric aging affects the aerosol chemistry characteristics at QOMS, a typical evolution process of BB aerosol plume (referred as BB evolution case) was analyzed from April 30 at 15:00 when a fresh BB plume occurred to May 1 at 18:00 when the BB plume was highly aged after undergoing various atmospheric oxidation processes. The temporal variations of meteorological parameters, mass concentrations and mass contributions of each $PM_1$ species and OA components as well as other chemistry parameters before and during this BB evolution case were all shown in Fig. 12.

Before the BB evolution case, all the mass concentrations decreased slowly and synchronously from 00:00 to 10:00 on April 30, which were consistent with the nearly stable trends of mass contributions and other chemistry parameters, indicating the relatively unified air mass sources and stable atmospheric conditions. After that, the wind circulations changed from the thermally-driven down-slope winds (mostly southwest) to the weak up-slope winds (northeast). In this period, BBOA and $f_{60}$ values kept relatively stable in contrast to other species likely due to the weak of air dilution and local sources. All the species reached the minimum at around 15:00 due to the lift of PBL.

The BB evolution case in this study was further divided into three different situations (as marked with arrows in Fig. 12 and 13), including the arriving of the fresh BBOA plume (from 15:00 to 24:00 on April 30), followed by the aqueous-phase oxidation in the nighttime (from 2:30 to 7:10 on May 1) and photochemical oxidation in the daytime (from 10:00 to 18:00 on May 1). All the mass concentrations began to increase from 15:00 and finally reached the maximum $PM_1$ mass loading of 18.4 μg m$^{-3}$ at 24:00, which was about four times higher than the average $PM_1$ mass during the entire campaign. Thus continuous increase was mainly dominated by the dramatic increase of BBOA, which reached up to 10.8 μg m$^{-3}$ and contributed 88% of the total OA mass and 50% of the total $PM_1$ at 24:00 (Fig. 13a), suggesting a distinct presence of BB emissions during this period. In contrast, the total OA mass was comprised by 78% of MO-OOA and 12% of BBOA at 15:00. Similar continuous increase trend could also be found for the mass concentration of calculated organic nitrate in this stage. In addition, nine aerosol chemistry parameters were presented as a function of BBOA mass concentrations during this period (Fig. 13b). The mass contributions of OA to $PM_1$ ($f_{Org}$) and BBOA to total OA ($f_{BBOA}$), $f_{60}$, and H/C ratio were all increased with the increasing BBOA mass, whereas the mass contribution of MO-OOA to total

OA ($f_{MO-OOA}$), O/C ratio, carbon oxidation state ($OS_c = 2 \times O/C - H/C$) of OA, and aerosol single scattering albedo (SSA) were decreased obviously, indicating the fresh nature of this BB plume. The significant impacts of fresh BB plume during this period was mainly associated with the unique wind circulation and the long-range transport of air masses. As displayed in Fig. 12b, the wind circulation changed from the weak up-slope winds to the strong down-slope glacier winds on April 30 at 15:00, with the WS increased from ~ 2 to 8 m s$^{-1}$. Meanwhile, the back trajectories in this period also presented that the long-range transport of air masses passed over the northern India and Nepal where active wildfires occurred, then air masses would accumulate and uplift to cross the Himalayas and finally downward to QOMS with the strong glacier winds.

A distinct aqueous-phase oxidation process was found in the nighttime from 02:30 to 07:10 on May 1. Although the total PM$_1$ and its species showed nearly stable mass concentrations during this period, the BBOA mass decreased gradually (from 82% to 70%) whereas MO-OOA increased constantly (from 14% to 20%) with the significant increase of RH (up to 91%) and aerosol liquid water content (ALWC) (Fig. 12). The scattering plots of the aerosol chemistry parameters versus the logarithmic values of cumulative ALWC, which could be used for the aqueous-phase oxidation during transport, also showed apparent increase trends for $f_{MO-OOA}$, O/C ratio, $OS_c$, and SSA that generally indicated the aerosol aging extent. All of these together suggested a distinct aqueous-phase oxidation of BBOA in the nighttime.

Since sunrise, all the mass concentrations decreased gradually, mainly related to the increasing PBL height and the clear air mass dilution. The back trajectories indicated that air masses during this period firstly went into the inland of the north of QOMS where had rare wildfires. Moreover, the BB plume would further undergo strong photochemical oxidation in the daytime due to the strong solar radiation. MO-OOA just contributed 26% of the total OA mass at 10:00, but it could increase to 74% at 18:00 after long-time photochemical oxidation. In contrast, BBOA mass contribution decreased from 49% to 20%. The cumulative solar radiation, which denoted the total amount of solar radiation that the plumes were exposed to during transport, could be used as an indicator for the extent of photochemical aging in the daytime (Zhou et al., 2017). Clear increased trend were found for $f_{MO-OOA}$, O/C ratio, $OS_c$, and SSA values with the increasing of cumulative solar radiation, whereas decreased trend in $f_{BBOA}$, $f_{NOA}$, H/C ratio, and $f_{60}$ values, suggesting a possible oxidation mechanism that the relatively fresh BBOA and NOA oxidized to aged MO-OOA in the daytime. Another interesting phenomenon was the continuous increase of SSA during both the aqueous-phase and photochemical oxidation periods on May 1 (Fig. 12e and 13b), indicating the potential influence of atmospheric aging to aerosol optical property at QOMS.

## 4 Conclusions

A comprehensive characterization of submicron aerosol chemical compositions and sources was investigated at the QOMS during the pre-monsoon season in 2016. The average mass concentration of PM$_1$ (NR-PM$_1$ + BC) was 4.44 ($\pm$ 4.54) μg m$^{-3}$ for the entire study, which was much lower than those observed in various sites in China. OA was the dominant PM$_1$ species (accounted for 54.3% of the total mass on average) and its contributions increased with the increase PM$_1$ mass loading. The average size distributions of all PM$_1$ species displayed an overlapping and narrow accumulation mode at ~ 500 nm, indicating the internally well-mixed and aged aerosol particles at QOMS. All species presented similar diurnal cycles, with lower

concentrations in the daytime whereas higher concentrations at the nighttime, mainly attributed to the dynamic variations of PBL height. Three OA factors were identified by PMF analysis on the high-resolution OA mass spectrum, including a relatively fresh BB-related OA (BBOA), a nitrogen-containing OA (NOA) and a more-oxidized oxygenated OA (MO-OOA). BBOA and MO-OOA could respectively account for 43.7% and 42.4% of OA mass on average, however, their contributions to OA showed completely opposite variation trends with the increase of OA mass. A continuously increased trend could be found for BBOA with the increasing OA, suggesting the key role of BBOA during polluted periods when frequent and intense wildfires were observed in South Asia. The significant impact of BB emissions on aerosol characteristics at QOMS have been also illustrated for different air mass origins and periods, respectively. Elevated $PM_1$ mass concentrations and high contributions of BBOA were found for both polluted clusters and polluted periods. A case study of typical evolution process of BB aerosol plume was investigated in detail to illustrate the chemical evolution of aerosol characteristics at QOMS. The fresh BB plume occurred in the afternoon on April 30 and finally resulted in highly $PM_1$ mass loading of 18.4 μg m$^{-3}$, which was about four times higher than the average $PM_1$ mass during the entire campaign. Obvious aqueous-phase oxidation and photochemical oxidation processes were analyzed in the nighttime and daytime on May 1, respectively, both suggesting the oxidation mechanism that fresh BBOA to aged MO-OOA. The continuous increase of SSA during the two oxidation periods suggested the potential influence of atmospheric aging to aerosol optical property at QOMS.

*Acknowledgements.* This research was supported by grants from the National Natural Science Foundation of China (41771079, 41421061), the Key Laboratory of Cryospheric Sciences Scientific Research Foundation (SKLCS-ZZ-2017-01), and the Chinese Academy of Sciences Hundred Talents Program. The authors thank the colleagues for continuing support and discussion, and thank the NOAA Air Resources Laboratory, NASA MODIS and FIRMS teams for providing the HYSPLIT trajectory model, AOD and fire hotspots datasets.

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

Background aerosol over the Himalayas and Tibetan Plateau: observed characteristics of aerosol mass loading, Atmos. Chem.
Phys., 17, 449-463, doi:10.5194/acp-17-449-2017, 2017.

Liu, Z., Liu, D., Huang, J., Vaughan, M., Uno, I., Sugimoto, N., Kittaka, C., Trepte, C., Wang, Z., Hostetler, C., and Winker, D.:
Airborne dust distributions over the Tibetan Plateau and surrounding areas derived from the first year of CALIPSO lidar
observations, Atmos. Chem. Phys., 8, 5045-5060, doi:10.5194/acp-8-5045-2008, 2008.

Ma, Y., Kang, S., Zhu, L., Xu, B., Tian, L., and Yao, T.: ROOF OF THE WORLD: Tibetan Observation and Research Platform,
Bulletin of the American Meteorological Society, 89, 1487-1492, doi:10.1175/2008bams2545.1, 2008.
Marcq, S., Laj, P., Roger, J. C., Villani, P., Sellegri, K., Bonasoni, P., Marinoni, A., Cristofanelli, P., Verza, G. P., and Bergin, M.:
Aerosol optical properties and radiative forcing in the high Himalaya based on measurements at the Nepal Climate
Observatory-Pyramid site (5079 m a.s.l.), Atmos. Chem. Phys., 10, 5859-5872, doi:10.5194/acp-10-5859-2010, 2010.
Marinoni, A., Cristofanelli, P., Laj, P., Duchi, R., Putero, D., Calzolari, F., Landi, T. C., Vuillermoz, E., Maione, M., and
Bonasoni, P.: High black carbon and ozone concentrations during pollution transport in the Himalayas: Five years of
continuous observations at NCO-P global GAW station, J. Environ. Sci., 25, 1618-1625, doi:10.1016/S1001-0742(12)60242-3,
2013.
Mohr, C., DeCarlo, P. F., Heringa, M. F., Chirico, R., Slowik, J. G., Richter, R., Reche, C., Alastuey, A., Querol, X., Seco, R.,
Peñuelas, J., Jiménez, J. L., Crippa, M., Zimmermann, R., Baltensperger, U., and Prévôt, A. S. H.: Identification and
quantification of organic aerosol from cooking and other sources in Barcelona using aerosol mass spectrometer data, Atmos.
Chem. Phys., 12, 1649-1665, doi:10.5194/acp-12-1649-2012, 2012.
Ng, N., Canagaratna, M., Jimenez, J., Zhang, Q., Ulbrich, I., and Worsnop, D.: Real-time methods for estimating organic
component mass concentrations from aerosol mass spectrometer data, Environ. Sci. Technol., 45, 910-916,
doi:10.1021/es102951k, 2011a.
Ng, N. L., Canagaratna, M. R., Jimenez, J. L., Chhabra, P. S., Seinfeld, J. H., and Worsnop, D. R.: Changes in organic aerosol
composition with aging inferred from aerosol mass spectra, Atmos. Chem. Phys., 11, 6465-6474, doi:10.5194/acp-11-6465-
2011, 2011b.
Paatero, P., and Tapper, U.: Positive matrix factorization: A non-negative factor model with optimal utilization of error estimates
of data values, Environmetrics, 5, 111-126, doi:10.1002/env.3170050203, 1994.
Putero, D., Landi, T. C., Cristofanelli, P., Marinoni, A., Laj, P., Duchi, R., Calzolari, F., Verza, G. P., and Bonasoni, P.: Influence
of open vegetation fires on black carbon and ozone variability in the southern Himalayas (NCO-P, 5079 m a.s.l.), Environ.
Pollut., 184, 597-604, doi:10.1016/j.envpol.2013.09.035, 2014.
Raatikainen, T., Vaattovaara, P., Tiitta, P., Miettinen, P., Rautiainen, J., Ehn, M., Kulmala, M., Laaksonen, A., and Worsnop, D. R.:
Physicochemical properties and origin of organic groups detected in boreal forest using an aerosol mass spectrometer, Atmos.
Chem. Phys., 10, 2063-2077, doi:10.5194/acp-10-2063-2010, 2010.
Ram, K., Sarin, M. M., and Hegde, P.: Long-term record of aerosol optical properties and chemical composition from a high-
altitude site (Manora Peak) in Central Himalaya, Atmos. Chem. Phys., 10, 11791-11803, doi:10.5194/acp-10-11791-2010,
2010.
Ramanathan, V., and Carmichael, G.: Global and regional climate changes due to black carbon, Nature Geoscience, 1, 221-227,
doi:10.1038/ngeo156, 2008.
Saarikoski, S., Carbone, S., Decesari, S., Giulianelli, L., Angelini, F., Canagaratna, M., Ng, N. L., Trimborn, A., Facchini, M. C.,
Fuzzi, S., Hillamo, R., and Worsnop, D.: Chemical characterization of springtime submicrometer aerosol in Po Valley, Italy,
Atmos. Chem. Phys., 12, 8401-8421, doi:10.5194/acp-12-8401-2012, 2012.
Sun, Y., Zhang, Q., Macdonald, A., Hayden, K., Li, S., Liggio, J., Liu, P., Anlauf, K., Leaitch, W., and Steffen, A.: Size-resolved
aerosol chemistry on Whistler Mountain, Canada with a high-resolution aerosol mass spectrometer during INTEX-B, Atmos.
Chem. Phys., 9, 3095-3111, doi:10.5194/acp-9-3095-2009, 2009.
Sun, Y. L., Zhang, Q., Schwab, J. J., Chen, W. N., Bae, M. S., Hung, H. M., Lin, Y. C., Ng, N. L., Jayne, J., Massoli, P., Williams,
870        L. R., and Demerjian, K. L.: Characterization of near-highway submicron aerosols in New York City with a high-resolution
aerosol mass spectrometer, Atmos. Chem. Phys., 12, 2215-2227, doi:10.5194/acp-12-2215-2012, 2012.
Sun, Y., Wang, Z., Fu, P., Jiang, Q., Yang, T., Li, J., and Ge, X.: The impact of relative humidity on aerosol composition and
evolution processes during wintertime in Beijing, China, Atmos. Environ., 77, 927-934, doi:10.1016/j.atmosenv.2013.06.019,
2013.
Sun, Y., Du, W., Fu, P., Wang, Q., Li, J., Ge, X., Zhang, Q., Zhu, C., Ren, L., Xu, W., Zhao, J., Han, T., Worsnop, D. R., and
Wang, Z.: Primary and secondary aerosols in Beijing in winter: sources, variations and processes, Atmos. Chem. Phys., 16,
8309-8329, doi:10.5194/acp-16-8309-2016, 2016.
Takami, A., Miyoshi, T., Shimono, A., and Hatakeyama, S.: Chemical composition of fine aerosol measured by AMS at Fukue
Island, Japan during APEX period, Atmos. Environ., 39, 4913-4924, doi:10.1016/j.atmosenv.2005.04.038, 2005.
Ulbrich, I. M., Canagaratna, M. R., Zhang, Q., Worsnop, D. R., and Jimenez, J. L.: Interpretation of organic components from
Positive Matrix Factorization of aerosol mass spectrometric data, Atmos. Chem. Phys., 9, 2891-2918, doi:10.5194/acp-9-2891-
2009, 2009.
Wan, X., Kang, S., Li, Q., Rupakheti, D., Zhang, Q., Guo, J., Chen, P., Tripathee, L., Rupakheti, M., Panday, A. K., Wang, W.,
Kawamura, K., Gao, S., Wu, G., and Cong, Z.: Organic molecular tracers in the atmospheric aerosols from Lumbini, Nepal, in
the northern Indo-Gangetic Plain: influence of biomass burning, Atmos. Chem. Phys., 17, 8867-8885, doi:10.5194/acp-17-
8867-2017, 2017.
Wang, J., Ge, X., Chen, Y., Shen, Y., Zhang, Q., Sun, Y., Xu, J., Ge, S., Yu, H., and Chen, M.: Highly time-resolved urban aerosol
characteristics during springtime in Yangtze River Delta, China: insights from soot particle aerosol mass spectrometry, Atmos.

Chem. Phys., 16, 9109-9127, doi:10.5194/acp-16-9109-2016, 2016a.

Wang, J., Onasch, T. B., Ge, X., Collier, S., Zhang, Q., Sun, Y., Yu, H., Chen, M., Prévôt, A. S. H., and Worsnop, D. R.: Observation of Fullerene Soot in Eastern China, Environ. Sci. Technol. Lett., 3, 121-126, doi:10.1021/acs.estlett.6b00044, 2016b.

Wang, J., Zhang, Q., Chen, M., Collier, S., Zhou, S., Ge, X., Xu, J., Shi, J., Xie, C., Hu, J., Ge, S., Sun, Y., and Coe, H.: First Chemical Characterization of Refractory Black Carbon Aerosols and Associated Coatings over the Tibetan Plateau (4730 m a.s.l), Environ. Sci. Technol., 51, 14072-14082, doi:10.1021/acs.est.7b03973, 2017.

Wang, Y., Hu, M., Lin, P., Guo, Q., Wu, Z., Li, M., Zeng, L., Song, Y., Zeng, L., Wu, Y., Guo, S., Huang, X., and He, L.: Molecular Characterization of Nitrogen-Containing Organic Compounds in Humic-like Substances Emitted from Straw Residue Burning, Environ. Sci. Technol., 51, 5951-5961, doi:10.1021/acs.est.7b00248, 2017.

Wu, G., Liu, Y., Zhang, Q., Duan, A., Wang, T., Wan, R., Liu, X., Li, W., Wang, Z., and Liang, X.: The Influence of Mechanical and Thermal Forcing by the Tibetan Plateau on Asian Climate, Journal of Hydrometeorology, 8, 770-789, doi:10.1175/jhm609.1, 2007.

Xia, X., Zong, X., Cong, Z., Chen, H., Kang, S., and Wang, P.: Baseline continental aerosol over the central Tibetan plateau and a case study of aerosol transport from South Asia, Atmos. Environ., 45, 7370-7378, doi:10.1016/j.atmosenv.2011.07.067, 2011.

Xu, B., Cao, J., Hansen, J., Yao, T., Joswia, D. R., Wang, N., Wu, G., Wang, M., Zhao, H., Yang, W., Liu, X., and He, J.: Black soot and the survival of Tibetan glaciers, Proc. Natl. Acad. Sci. USA, 106, 22114-22118, doi:10.1073/pnas.0910444106, 2009.

Xu, J., Zhang, Q., Chen, M., Ge, X., Ren, J., and Qin, D.: Chemical composition, sources, and processes of urban aerosols during summertime in northwest China: insights from high-resolution aerosol mass spectrometry, Atmos. Chem. Phys., 14, 12593-12611, doi:10.5194/acp-14-12593-2014, 2014.

Xu, J., Shi, J., Zhang, Q., Ge, X., Canonaco, F., Prévôt, A. S. H., Vonwiller, M., Szidat, S., Ge, J., Ma, J., An, Y., Kang, S., and Qin, D.: Wintertime organic and inorganic aerosols in Lanzhou, China: sources, processes, and comparison with the results during summer, Atmos. Chem. Phys., 16, 14937-14957, doi:10.5194/acp-16-14937-2016, 2016.

Xu, J., Zhang, Q., Shi, J., Ge, X., Xie, C., Wang, J., Kang, S., Zhang, R., and Wang, Y.: Chemical characteristics of submicron particles at the central Tibet Plateau: influence of long-range transport, Atmos. Chem. Phys. Discuss., 1-32, doi:10.5194/acp-2017-587, 2017.

Xu, L., Suresh, S., Guo, H., Weber, R. J., and Ng, N. L.: Aerosol characterization over the southeastern United States using high-resolution aerosol mass spectrometry: spatial and seasonal variation of aerosol composition and sources with a focus on organic nitrates, Atmos. Chem. Phys., 15, 7307-7336, doi:10.5194/acp-15-7307-2015, 2015.

Yadav, I. C., Devi, N. L., Li, J., Syed, J. H., Zhang, G., and Watanabe, H.: Biomass burning in Indo-China peninsula and its impacts on regional air quality and global climate change-a review, Environmental pollution, 227, 414-427, doi:10.1016/j.envpol.2017.04.085, 2017.

Yang, K., Wu, H., Qin, J., Lin, C., Tang, W., and Chen, Y.: Recent climate changes over the Tibetan Plateau and their impacts on energy and water cycle: A review, Global and Planetary Change, 112, 79-91, doi:10.1016/j.gloplacha.2013.12.001, 2014.

Yao, T., Thompson, L., Mosbrugger, V., Zhang, F., Ma, Y., Luo, T., Xu, B., Yang, X., Joswiak, D. R., Wang, W., Joswiak, M. E., Devkota, L. P., Tayal, S., Jilani, R., and Fayziev, R.: Third Pole Environment (TPE), Environmental Development, 3, 52-64, doi:10.1016/j.envdev.2012.04.002, 2012a.

Yao, T., Thompson, L., Yang, W., Yu, W., Gao, Y., Guo, X., Yang, X., Duan, K., Zhao, H., Xu, B., Pu, J., Lu, A., Xiang, Y., Kattel, D. B., and Joswiak, D.: Different glacier status with atmospheric circulations in Tibetan Plateau and surroundings, Nature Clim. Change, 2, 663-667, doi:10.1038/nclimate1580, 2012b.

Zhang, Q., Alfarra, M. R., Worsnop, D. R., Allan, J. D., Coe, H., Canagaratna, M. R., and Jimenez, J. L.: Deconvolution and quantification of hydrocarbon-like and oxygenated organic aerosols based on aerosol mass spectrometry, Environ. Sci. Technol., 39, 4938-4952, doi:10.1021/es048568l, 2005a.

Zhang, Q., Canagaratna, M. R., Jayne, J. T., Worsnop, D. R., and Jimenez, J. L.: Time- and size-resolved chemical composition of submicron particles in Pittsburgh: Implications for aerosol sources and processes, J. Geophys. Res., 110, doi:10.1029/2004jd004649, 2005b.

Zhang, Q., Jimenez, J. L., Canagaratna, M. R., Allan, J. D., Coe, H., Ulbrich, I., Alfarra, M. R., Takami, A., Middlebrook, A. M., Sun, Y. L., Dzepina, K., Dunlea, E., Docherty, K., DeCarlo, P. F., Salcedo, D., Onasch, T., Jayne, J. T., Miyoshi, T., Shimono, A., Hatakeyama, S., Takegawa, N., Kondo, Y., Schneider, J., Drewnick, F., Borrmann, S., Weimer, S., Demerjian, K., Williams, P., Bower, K., Bahreini, R., Cottrell, L., Griffin, R. J., Rautiainen, J., Sun, J. Y., Zhang, Y. M., and Worsnop, D. R.: Ubiquity and dominance of oxygenated species in organic aerosols in anthropogenically-influenced Northern Hemisphere midlatitudes, Geophys. Res. Lett., 34, doi:10.1029/2007gl029979, 2007a.

Zhang, Q., Jimenez, J. L., Worsnop, D. R., and Canagaratna, M.: A case study of urban particle acidity and its influence on secondary organic aerosol, Environ. Sci. Technol., 41, 3213-3219, doi:10.1021/es061812j, 2007b.

Zhang, Q., Jimenez, J. L., Canagaratna, M. R., Ulbrich, I. M., Ng, N. L., Worsnop, D. R., and Sun, Y.: Understanding atmospheric organic aerosols via factor analysis of aerosol mass spectrometry: a review, Anal. Bioanal. Chem., 401, 3045-3067, doi:10.1007/s00216-011-5355-y, 2011.

Zhang, R., Wang, Y., He, Q., Chen, L., Zhang, Y., Qu, H., Smeltzer, C., Li, J., Alvarado, L. M. A., Vrekoussis, M., Richter, A.,

Wittrock, F., and Burrows, J. P.: Enhanced trans-Himalaya pollution transport to the Tibetan Plateau by cut-off low systems,
Atmos. Chem. Phys., 17, 3083-3095, doi:10.5194/acp-17-3083-2017, 2017a.
Zhang, X., Zhang, Y., Sun, J., Yu, Y., Canonaco, F., Prevot, A. S., and Li, G.: Chemical characterization of submicron aerosol
particles during wintertime in a northwest city of China using an Aerodyne aerosol mass spectrometry, Environ. Pollut., 222,
567-582, doi:10.1016/j.envpol.2016.11.012, 2017b.
Zhao, Z., Cao, J., Shen, Z., Xu, B., Zhu, C., Chen, L. W. A., Su, X., Liu, S., Han, Y., Wang, G., and Ho, K.: Aerosol particles at a
high-altitude site on the Southeast Tibetan Plateau, China: Implications for pollution transport from South Asia, J. Geophys.
Res.-Atmos., 118, 11360-11375, doi:10.1002/jgrd.50599, 2013.
Zheng, J., Hu, M., Du, Z., Shang, D., Gong, Z., Qin, Y., Fang, J., Gu, F., Li, M., Peng, J., Li, J., Zhang, Y., Huang, X., He, L., Wu,
Y., and Guo, S.: Influence of biomass burning from South Asia at a high-altitude mountain receptor site in China, Atmos.
Chem. Phys., 17, 6853-6864, doi:10.5194/acp-17-6853-2017, 2017.
Zhou, S., Collier, S., Jaffe, D. A., Briggs, N. L., Hee, J., Sedlacek Iii, A. J., Kleinman, L., Onasch, T. B., and Zhang, Q.: Regional
influence of wildfires on aerosol chemistry in the western US and insights into atmospheric aging of biomass burning organic
aerosol, Atmos. Chem. Phys., 17, 2477-2493, doi:10.5194/acp-17-2477-2017, 2017.
Zhu, Q., He, L. Y., Huang, X. F., Cao, L. M., Gong, Z. H., Wang, C., Zhuang, X., and Hu, M.: Atmospheric aerosol compositions
and sources at two national background sites in northern and southern China, Atmos. Chem. Phys., 16, 10283-10297,
doi:10.5194/acp-16-10283-2016, 2016.
Zou, H., Zhou, L., Ma, S., Li, P., Wang, W., Li, A., Jia, J., and Gao, D.: Local wind system in the Rongbuk Valley on the northern
slope of Mt. Everest, Geophys. Res. Lett., 35, doi:10.1029/2008gl033466, 2008.

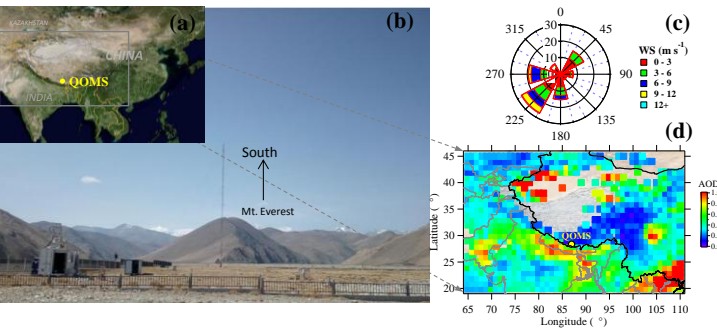

**Figure 1. (a)** Location map for the QOMS, **(b)** picture for the QOMS and its surrounding, **(c)** wind rose plot colored by wind
speed in this study, and **(d)** distribution of the average aerosol optical depth (AOD) around the QOMS retrieved from Terra
MODIS at 550 nm during this study.

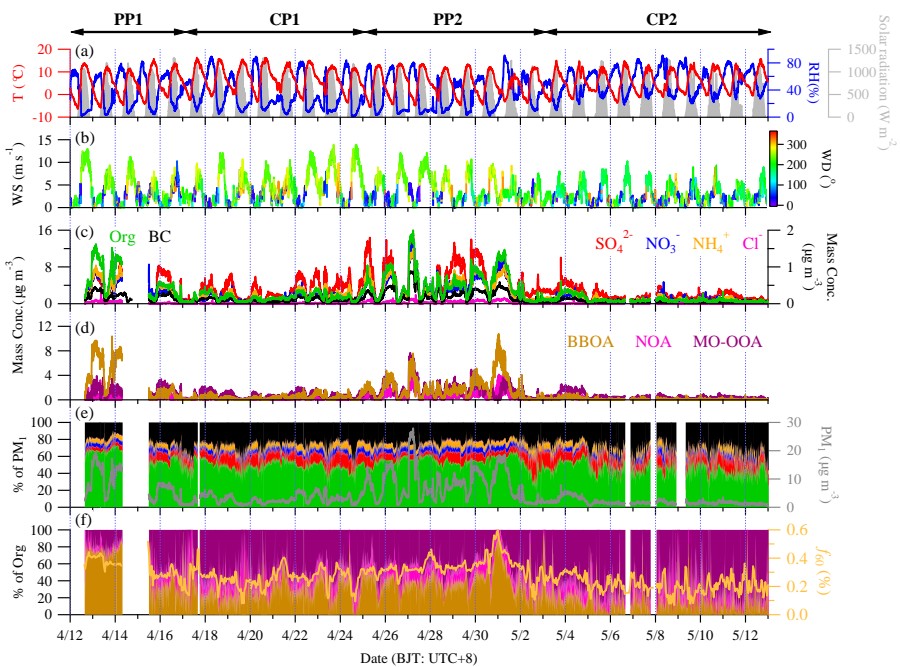

**Figure 2.** Summary of meteorological and HR-ToF-AMS data. The 5-min time series of **(a)** ambient temperature ($T$), relative
humidity (RH), and solar radiation, **(b)** wind speed (WS) colored by wind direction (WD), **(c)** mass concentrations of $PM_1$
species, **(d)** mass concentrations of organic components, **(e)** mass contributions of $PM_1$ species to total $PM_1$ as well as total $PM_1$
mass concentrations, and **(f)** mass contributions of organic components to organics. The time series of hourly average $f_{60}$ (=
$C_2H_4O_2^+$ / OA) values for the entire period is also showed. The markers of PP1 and PP2 represent the two polluted periods while
CP1 and CP2 are clear periods, respectively.

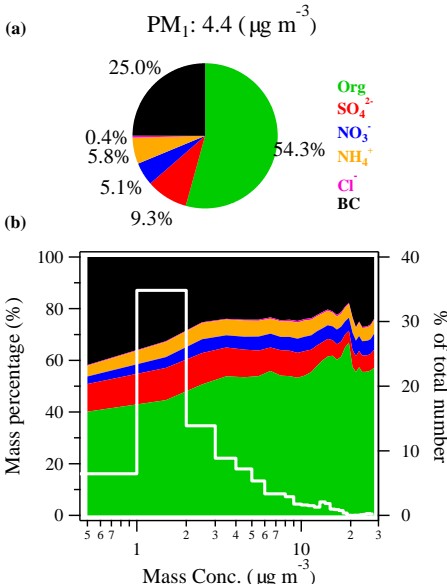

**Figure 3.** The average mass contributions of $PM_1$ (= NR-PM$_1$+ BC) species **(a)** during the entire sampling period and **(b)** as a
function of the total $PM_1$ mass concentrations. The white solid line in **(b)** shows the percentage of the data number in each mass
bins to the total data number.

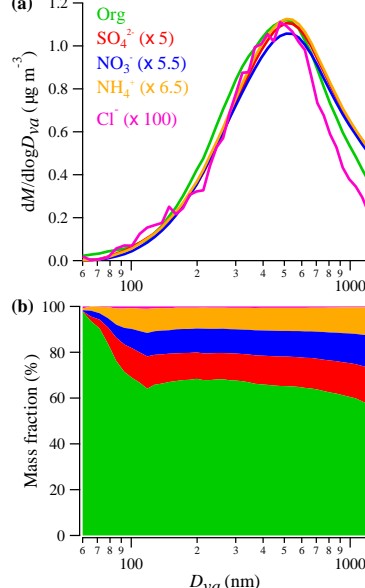

**Figure 4.** The average size distributions of **(a)** mass concentrations and **(b)** mass contributions of NR-PM$_1$ species for the entire
study.

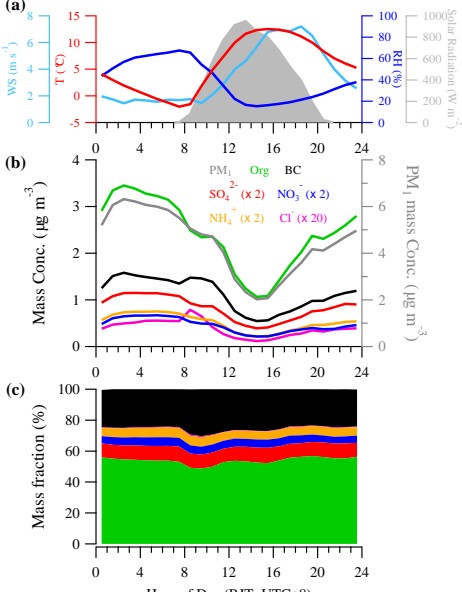

**Figure 5.** The diurnal cycles of **(a)** meteorological parameters (temperature, RH, wind speed, and solar radiation), **(b)** mass concentrations and **(c)** mass contributions of PM$_1$ chemical species for the entire study.

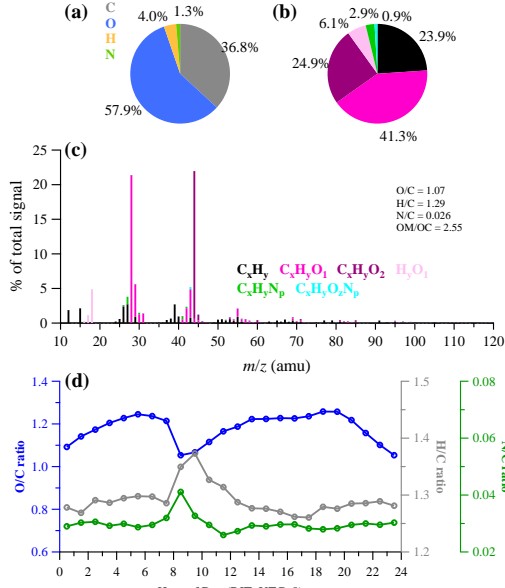

**Figure 6.** The average contributions of **(a)** four elements (C, O, H, and N) and **(b)** six ion categories (colors as in **(c)**) to OA for the entire study; **(c)** the average high-resolution mass spectrum of OA (colors show six ion categories); **(d)** the diurnal variations of O/C, H/C, and N/C ratios.

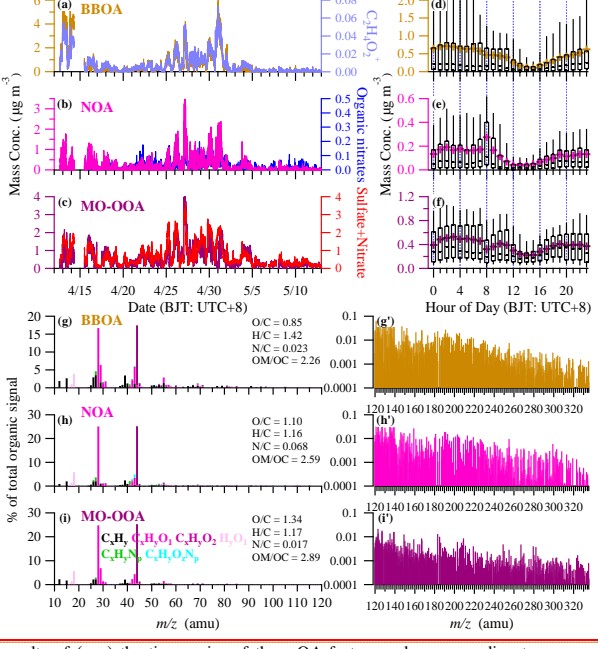

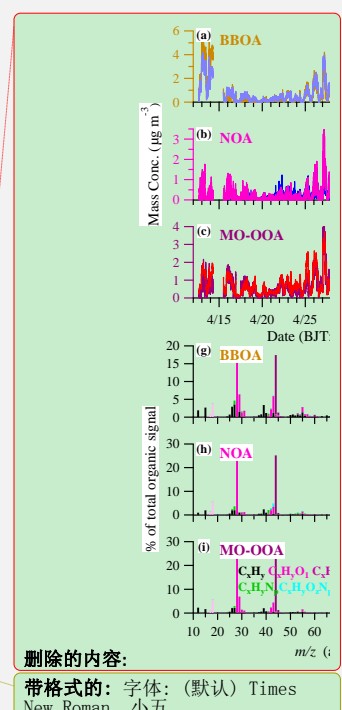

**Figure 7.** The PMF results of **(a–c)** the time series of three OA factors and corresponding tracer species, **(d-f)** the diurnal
variations of the mass concentrations of the three OA factors (the whiskers above and below the boxes indicate the 90th and 10th
percentiles, the upper and lower boundaries respectively indicate the 75th and 25th percentiles, the lines in the boxes indicate the
median values, and the cross symbols indicate the mean values), **(g-i)** high-resolution mass spectra of the three OA factors
colored by six ion families at *m/z* < 120, and **(g'-i')** the unit resolution mass spectra at *m/z* > 120 for each OA factor.

New Roman, 小五

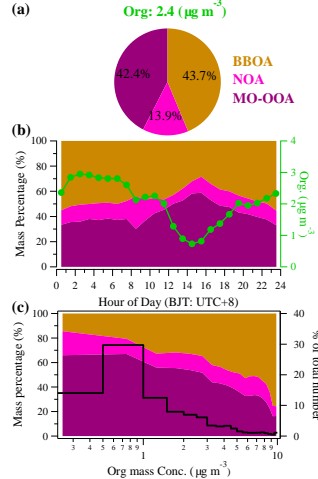

**Figure 8. (a)** The average mass concentration of OA and mass contributions of three OA factors to total OA; **(b)** the diurnal
variations of mass contributions of three OA factors to total OA and the total OA mass concentration; **(c)** The average mass
contributions of three OA factors as a function of total OA mass concentrations. The black solid line in **(c)** shows the percentage
of the data number in each OA mass bins to the total data number.

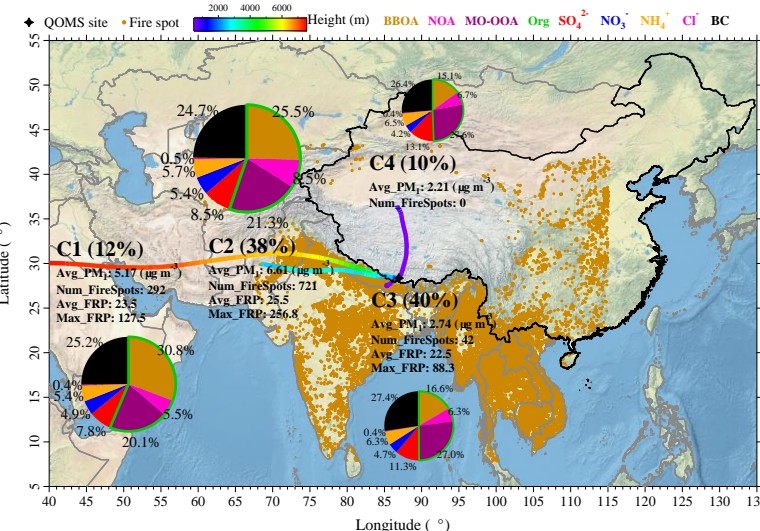

**Figure 9.** The average back trajectory clusters during the entire study and the corresponding mass contributions of PM₁ species
and OA factors to the total PM₁ mass. The areas of each pie charts are scaled by the corresponding average PM₁ mass
concentrations. The average PM₁ mass concentrations, number of fire hotspots as well as the average and maximum fire radiative
powers (FRP) belong to each clusters during the entire measurement period are also given.

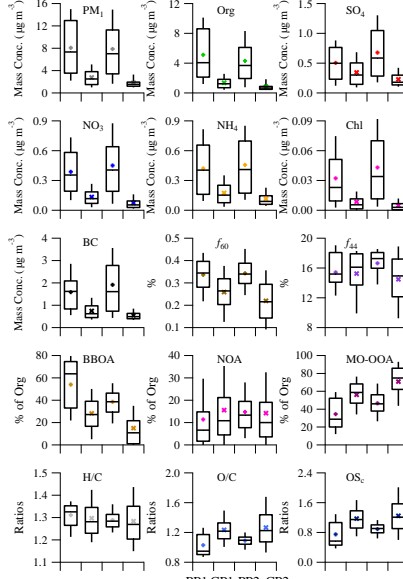

**Figure 10.** Box plots of mass concentrations of total PM₁ and its species, $f_{60}$ and $f_{44}$ values, mass contribution of three OA
components to organics, element ratios (H/C and O/C), and carbon oxidation states (OS$_c$) among the four polluted and clear
periods. The whiskers indicate the 90th and 10th percentiles, the upper and lower boundaries of boxes indicate the 75th and 25th
percentiles, the lines in the boxes indicate the median values, and the markers indicate the mean values.

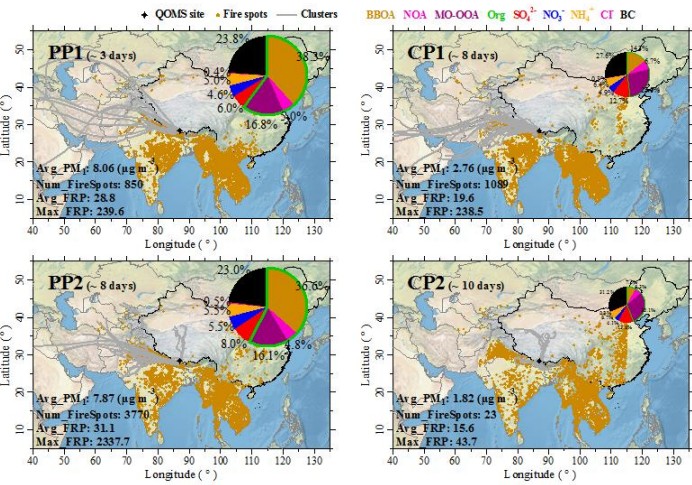

**Figure 11.** The 72-h back trajectories (grey solid lines) calculated every 6 h for the different episodes. Pie charts show the average mass contributions of PM$_1$ species and OA factors to the total PM$_1$ mass for each episodes (scaled by the corresponding average PM$_1$ mass concentrations). The average PM$_1$ mass concentrations, number of fire hotspots as well as the average and maximum fire radiative powers (FRP) belong to all trajectories for the different episodes are also given.

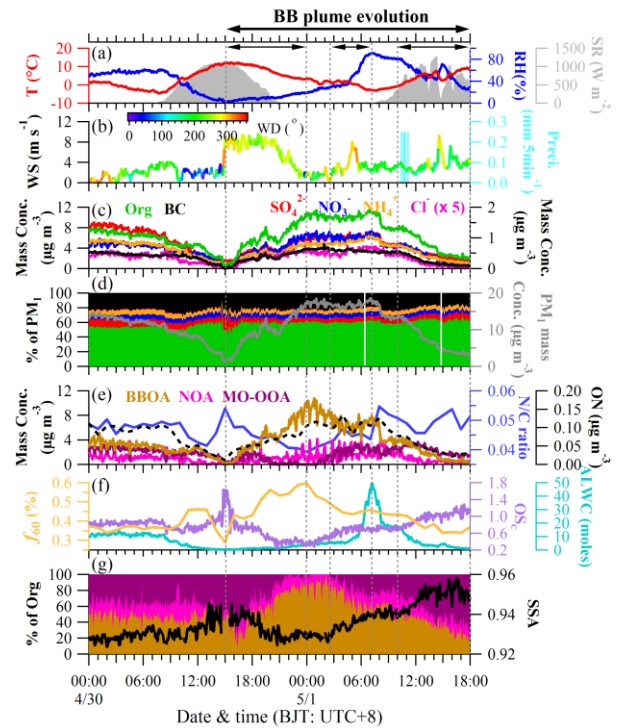

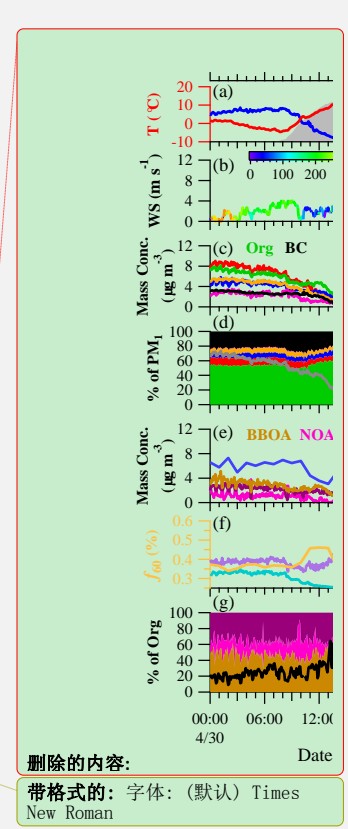

**Figure 12.** The temporal variations of meteorological parameters, mass concentrations and mass contributions of each PM$_1$ species and OA components as well as the N/C ratio, $f_{60}$ values, carbon oxidation states (OS$_c$), aerosol liquid water content (ALWC) and single scattering albedo (SSA) for the case study period from April 30 at 00:00 to May 1 at 18:00.

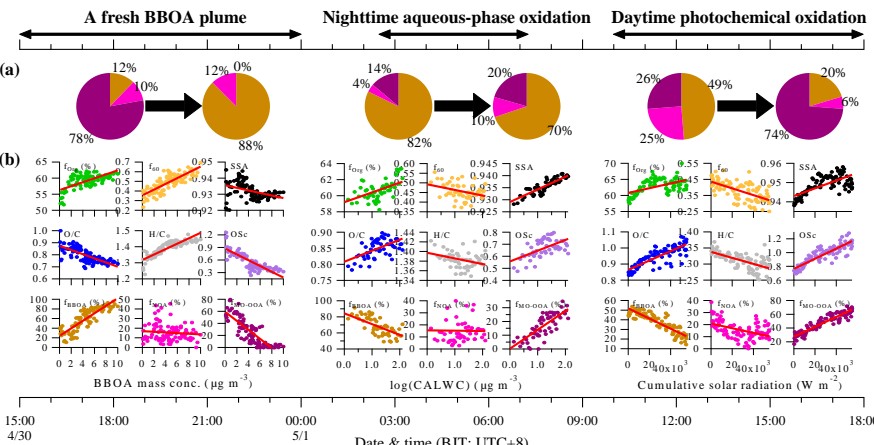

**Figure 13.** Case study of chemical evolution of BB plume from April 30 at 15:00 to May 1 at 18:00. The periods marked with
arrows are three distinct evolution processes. Pie charts in **(a)** are the mass contributions of three OA factors to total OA during
the beginning and end time for each process, respectively. The scattering plots in **(b)** are the aerosol chemistry parameters as a
function of BBOA mass concentration, logarithmic values of cumulative aerosol liquid water content (CALWC), and cumulative
solar radiation for the corresponding process.