# Peer review of "Chemical characterization of long-range transport biomass burning"

_Atmospheric Chemistry and Physics, 2017_

## Referee Comment (RC1) · Anonymous Referee #1 · 5 Dec 2017

The manuscript by Zhang et al. presents a detailed chemical characterization of $PM_1$ at a remote and high-altitude site in the northern Himalayas during pre-monsoon season by using HR-ToF-AMS measurements. Although the transport of biomass burning aerosol from the South Asia to the Tibet Plateau was reported previously, this study is unique in terms of real-time characterization and organic aerosol source apportionment. This study showed that organic aerosol (OA) was the dominant species in $PM_1$ and was highly aged during the long-range transport. Consistently, PMF of OA spectra showed that BBOA and OOA are two dominant OA factors. In addition, a detailed analysis of biomass burning plume was used to illustrate the impacts and chemical evolution of biomass burning aerosols. This manuscript is generally well written and I recommend it for publication after minor revisions.

Comments:

1. The BC contribution is unexpectedly high, up to 25% in this study. It is also much higher than that measured at another site in Tibetan Plateau, e.g. 8% by the same group. Any more explanation for the high BC contribution? For example, the absorption coefficient was measured at 405 nm which can be significantly affected by brown carbon from biomass burning emissions. Are there additional measurements to verify the BC data?

2. Line 426-428: Can the authors show some plots of FTICR-MS results to support the conclusions here?

3. The high resolution mass spectra in the figures appear to miss $m/z$ 13 ($CH^+$) and 14 ($CH_2^+$), any explanations?

4. Line 342: "...whereas carbon and $C_xH_y^+$ ions had relative higher contributions". "higher" should to be "lower".

5. The distribution of average aerosol optical depth (AOD) derived from MODIS is mentioned in Section 2.4 and presented in Fig. 1d in this study, however, no analysis or usage for this information in current version of this paper.

6. Figure 12, "polume" is a typo

7. Line 28, change "transportation" to "transport"

8. Figures 7g – 7i : m/z 28 was buried behind the OA names

9. Line 141: the size of critical orifice is 130 μm or 120 μm?

---

## Referee Comment (RC2) · Anonymous Referee #2 · 10 Dec 2017

General comments:

This paper reports for the first time data of PM1 species observed at the Qomolangma site (a high-elevation site north of Mt. Everest) during a pre-monsoon period using a high-resolution time-of-flight aerosol mass spectrometer (HR-ToF-AMS). Some ancillary data (meteorology, BC, size distribution, etc.) are also presented and analyzed together with the HR-ToF-AMS measurements. Mass concentrations and fractions of PM1 species are characterized. Impacts from local meteorological meteorology, long-range transport and biomass burning on the variations of PM1 species are investigated

and discussed. The measurements are also compared with those from some other remote and urban sites. The results indicate that the PM1 level at this remote site was not as low as expected. The mass concentration and compositions of PM1 are found to be influenced by biomass burning plumes that were transported from South Asian countries to the site. Significant biomass burning impacts caused higher fractions of organic aerosol (OA) and black carbon. Oxidation of OA during the transport enhanced the fraction of more-oxidized oxygenated OA (MO-OOA) and aerosol single scattering albedo (SSA). This paper adds new and valuable measurements of aerosol compositions and concentrations from the Tibetan Plateau, one of the less studied key regions. The analysis and the results of this paper are generally sound. The paper is within the scope of ACP and generally well written. I recommend publication of this paper in ACP after revisions. I only have some minor points for the author to consider in the revision.

Specific comments:

(1) Line 36, the highest altitude of the TPH should be over 8800 m asl but the average altitude is about 4000 m asl.

(2) Line 43, change "focused on" to "paid to".

(3) Line 70, change "heavily" to "mainly" or "exclusively".

(4) Line 185, change "The detailed analysis were" to "Detailed analysis was".

(5) Line 213, I think "starting" should be changed to "ending".

(6) Line 223, deleting "potential".

(7) Line 233, brackets are needed for the sigma value.

(8) Line 235, change "southwest" to "southwesterly".

(9) Line 265, a site cannot be both in the southeastern edge and in the central of a region.

(10) Lines 292-293 and elsewhere, "predicted" means something different. I think you calculated NH4 and compared it with the observed one. This sentence should be reworded and NH4+_pred should be changed to NH4+_calc.

(11) Lines 549-550, "Noting the N/C ratio also displayed constantly increased trend at night, probably associated with nitrate radical oxidation". I feel this is too speculative. You are talking about aqueous-phase oxidation. You have no measurements of nitrate radical in both gaseous and liquid phase.

(12) Fig. 9, are the fire hotspots annual averages or statistics for the measurement period?

(13) Fig. 12, why has the ALWC a unit "moles"?

(14) Fig. 13, if you intend to compare the correlations in different periods or conditions, you compare the some correlations. However, the x-axes are different for different periods. Such comparisons do not make sense.

(15) Figs. S5 and S6, give more details in figure captions.

(16) Fig. S12, you are not using discrete colors, so the color bar is not appropriate.

(17) This paper includes totally 26 figures and many figures contain several plots (partly too tiny to be easily read). When reading the paper, I felt sometimes lost jumping between the text and the cited figures. And I think many figures are not discussed to certain degree. I think the authors should show only figures that are really necessary and discussed them in detail. The presentation of the figures and their order should be improved.

---

## Referee Comment (RC3) · Anonymous Referee #3 · 20 Dec 2017

In this study, a filed study was performed from April 12 to May 12, 2016 at a high-altitude site (QOMS) in the northern Himalayas using an Aerodyne high-resolution time-of-flight aerosol mass spectrometer (HR-ToF-AMS) along with other collocated instruments, with the target to characterize the chemical composition, sources, and transport mechanism of polluted biomass burning aerosols from South Asia to the Himalayas during pre-monsoon season. As highly-time resolved aerosol measurement in such high altitude regions are very rare and important, the data set provided by this work is thus very valuable. The authors also performed a comprehensive analysis on

this dataset, and the findings, conclusions are well supported by such analyses. Overall, the manuscript is also well written and documented. The topic also fits well in the scope of ACP. I thus recommend this manuscript can be published after some revisions. General comments 1. In Fig. 2, the summary of temporal variation of meteorological parameters and various HR-ToF-AMS data are present. Dose all parameters show consistent time solution, e.g., 5-min? Especially f60 values, which shows unexpectedly smooth trend. Please give clear introduction.

2. The average mass contributions of PM1 species during the entire sampling period and as a function of the total PM1 mass concentrations was analyzed in section 3.1.3 and Fig. 3. Organic aerosol and black carbon are the two dominant PM1 species in this study. In addition, the author suggested that organic compounds and BC have been revealed as two dominant components of BB aerosols. However, with the continuously enhancement of pollution mainly due to the biomass burning emissions, why the contribution of BC does not increase as that of organics?

3. In this study, the diurnal variations of all PM1 species showed similar patterns that related with the dynamics of planetary boundary layer (PBL). Is there any measurement of PBL height at QOMS?

4. In the section 3.4.2 or Fig. 9 and 11, the author give the fire hotspots statistics, e.g., number, average and max FRP for each cluster or periods, how did the author calculated? The total statistics for the whole areas (5-55° N, 40-135° E) that you selected or other approach?

5. In the section 3.4.3, you discussed the evolution of BB emission aerosol, it will be very good if you could show behavior of organic nitrate during this period.

6. The author give the identification of the two polluted periods (PP1 and PP2) and two clear periods (CP1 and CP2) in section 3.4.2, however, the initial usage of these distinct periods are in Fig. 2 and section 3.3.1, thus make reader more confuse.

7. Change "transportation" to "transport" in the whole manuscript.

8. Line 43, "...on the environment and climate change in the TPH region since they are very susceptibility to the global climate" changed to "...on the environment and climate change in the TPH since this region is very susceptibility to the global climate".

9. Line 49, "had frequently been observed " changed to "had been observed frequently"

10. Line 64, "south Asia" changed to "South Asia", and similarly anywhere else.

11. Line 213, 387, and 500, "QOMS" was the abbreviation of "Qomolangma Station ", so change "QOMS site" to just "QOMS".

12. A recent study in the TP region can be included in Line 84 of the introduction part (Wang et al., Environ Sci Technol 2017, 51, (24), 14072-14082). In addition, a few new studies and reviews that summarizes the application of advanced AMS techniques can be added in Line 77-83 (Li et al., Atmos. Environ. 2017, 158, 270-304; Wang et al., Environ. Sci. Technol. Lett. 2016, 3, (4), 121-126; Wang et al., Atmos. Chem. Phys. 2016, 16, (14), 9109-9127.)
* * *

---

## Author Comment (AC1) · 7 Feb 2018

**Long-range transport biomass burning emissions to the Himalayas: insights from high-resolution aerosol mass spectrometry**

**Xinghua Zhang et al.**

We appreciate the reviewers for their constructive comments and suggestions. The manuscript has been revised accordingly. Our point-by-point responses to the comments are presented below. The comments are in **black** and responses in **blue**. Changes made to the manuscript are in **red**.

**Response to reviewer #1**

The manuscript by Zhang et al. presents a detailed chemical characterization of $PM_1$ at a remote and high-altitude site in the northern Himalayas during pre-monsoon season by using HR-ToF-AMS measurements. Although the transport of biomass burning aerosol from the South Asia to the Tibet Plateau was reported previously, this study is unique in terms of real-time characterization and organic aerosol source apportionment. This study showed that organic aerosol (OA) was the dominant species in $PM_1$ and was highly aged during the long-range transport. Consistently, PMF of OA spectra showed that BBOA and OOA are two dominant OA factors. In addition, a detailed analysis of biomass burning plume was used to illustrate the impacts and chemical evolution of biomass burning aerosols. This manuscript is generally well written and I recommend it for publication after minor revisions.

Thank you very much for your insightful suggestion and positive comments.

**Comments:**
(1) The BC contribution is unexpectedly high, up to 25% in this study. It is also much higher than that measured at another site in Tibetan Plateau, e.g. 8% by the same group. Any more explanation for the high BC contribution? For example, the absorption coefficient was measured at 405 nm which can be significantly affected by brown carbon from biomass burning emissions. Are there additional measurements to verify the BC data?

We agree the reviewer's comment. The BC contribution to $PM_1$ was indeed high (up to 25%) in this study and even three times higher than that (8%) measured at Nam Co station in the central of Tibetan Plateau by our group (Xu et al., 2017).

The BC concentrations were derived from a photoacoustic extinctiometer (PAX) in this study, which firstly measured the particle light absorption and scattering coefficient at 405 nm, then achieved the extinction coefficient and BC mass concentration. In order to verify the BC data from the PAX in this study, we compared it with that the BC data from a seven-wavelength aethalometer (model AE33, Magee Scientific, Berkeley, CA, USA) during the same period at QOMS. The time series of BC data from the two instruments correlated well, with a correlation coefficient of 0.94 and a slope of 1.1 (Fig. R1). The slight high BC data from PAX mainly due to the measurement of absorption coefficient at 405 nm that can be significantly affected by brown carbon from biomass burning emissions. In addition, the different measurement principles, sensitivities, and data processing methods for the two instruments might be another reasons for the slight difference of BC datasets.

As we stated in the manuscript, the high contributions of organics and BC at QOMS were significantly associated with the active biomass burning (BB) emissions by long-range transport from polluted areas in South Asia. Organic compounds and BC have been revealed as two dominant components of BB aerosols and generally used to identify BB events in previous studies (Bond et al., 2004; Bougiatioti et al., 2014). In addition, biomass burning at high elevation regions of Himalayas and south Asia could be more incomplete burning and emit amount of BC. Comparing the two field studies by our group at QOMS and Nam Co station, the intense of biomass burning emission aerosols from the polluted areas in South Asia could be easier to transport to QOMS than to Nam Co station due to the relative shorter transport distance of QOMS. Moreover, the measurement in this study was from April 12 to May 12, 2016 during the pre-monsoon season which included two intense BB polluted periods, whereas it was during both pre-monsoon and monsoon seasons (between May 31 and July 1, 2015) at Nam Co station, especially for monsoon season which showed much lower concentrations. In conclusion, the high BC contribution, which associated with the intense long-range transport of biomass burning emission during pre-monsoon season in the QOMS study, was reasonable.

[Figure]

Fig. R1. The time series (left) and comparison (right) of BC concentrations from PAX and AE33 measurements between April 12 and May 12, 2016 at QOMS.

(2) Line 426-428: Can the authors show some plots of FTICR-MS results to support the conclusions here?

The mass spectrum of one typical filter sample measured by a Fourier transform ion cyclotron resonance mass spectrometer (FTICR-MS) was shown below (Fig. R2). The assigned molecular formulas were divided into two groups based on their elemental composition including CHO and CHON which all had important contribution in molecular number. The high contribution of CHON compounds could from the specific bio-fuels in south Asia such as dung-cake using for residential cooking and warming. Fleming et al., (2017) show a very high nitrogen-containing compound contribution in dung-cake emission. Due to the comment from the reviewer#2 that too many figures are included in our manuscript, so we cannot add this figure finally to either manuscript or supplementary.

[Figure]

Fig. R2. The mass spectrum of one typical filter sample measured by Fourier transform ion cyclotron resonance mass spectrometer (FTICR-MS).

(3) The high resolution mass spectra in the figures appear to miss *m/z* 13 (CH+) and 14 (CH2+), any explanations?

As we mentioned in Sect. 2.3.2 for the HR-ToF-AMS data process in this study, the "bad" ions with S/N less than 0.2 were removed from the HRMS data and error matrices before PMF analysis. The *m/z* 13 ($CH^+$) and 14 ($CH_2^+$) were the two "bad" ions when performing PMF analysis, so they missed in the figures of high resolution mass spectra in this study.

(4) Line 342: "...whereas carbon and CxHy+ ions had relative higher contributions". "higher" should to be "lower".

We made a mistake and have corrected it as follows in the revised version.

"...whereas carbon and $C_xH_y^+$ ions had relative  lower contributions..."

(5) The distribution of average aerosol optical depth (AOD) derived from MODIS is mentioned in Section 2.4 and presented in Fig. 1d in this study, however, no analysis or usage for this information in current version of this paper.

The distribution of average aerosol optical depth (AOD) derived from MODIS in Fig. 1d was planned to show the serious polluted feature of aerosols in South and Southeast Asia. We have added a few sentences to describe the AOD information in the revised version as following.

"South and Southeast Asia are two major polluted regions due to their intense biomass burning (BB) activities from natural forest fires and traditional human burning activities for residential heating and cooking (Engling et al., 2011; Yadav et al., 2017). The polluted feature of South and Southeast Asia during April 12 to May 12, 2016 can be further revealed by the distribution of average aerosol optical depth in Fig. 1." in Introduction.

"Air masses during PP1 and PP2 generally originated form long-range transportation to the west of the QOMS, which would pass through intense wildfires areas in South Asia (e.g., Indo-Gangetic Plain and Nepal where showed high AOD values in Fig. 1d and active fire hotspots in Fig. 9 and 11)." in Sect. 3.4.2

(6) Figure 12, "polume" is a typo

We made a mistake on this word and have corrected "polume" to "plume".

(7) Line 28, change "transportation" to "transport"

Corrected.

"...aerosol particles were internally well-mixed and aged during long-range transport."

(8) Figures 7g – 7i : m/z 28 was buried behind the OA names

Thanks a lot for your suggestion. We have make the mass spectrum more clear in the revised version.

(9) Line 141: the size of critical orifice is 130 μm or 120 μm?

We used a 130 μm critical orifice in this study for the purpose to enhance the transmission efficiency and flow rate of HR-ToF-AMS as mentioned in Sect. 2.3.1 in the manuscript.

**Response to reviewer#2**

This paper reports for the first time data of $PM_1$ species observed at the Qomolangma site (a high-elevation site north of Mt. Everest) during a pre-monsoon period using a high-resolution time-of-flight aerosol mass spectrometer (HR-ToF-AMS). Some ancillary data (meteorology, BC, size distribution, etc.) are also presented and analyzed together with the HR-ToF-AMS measurements. Mass concentrations and fractions of $PM_1$ species are characterized. Impacts from local meteorological meteorology, long-range transport and biomass burning on the variations of $PM_1$ species are investigated and discussed. The measurements are also compared with those from some other remote and urban sites. The results indicate that the $PM_1$ level at this remote site was not as low as expected. The mass concentration and compositions of $PM_1$ are found to be influenced by biomass burning plumes that were transported from South Asian countries to the site. Significant biomass burning impacts caused higher fractions of organic aerosol (OA) and black carbon. Oxidation of OA during the transport enhanced the fraction of more-oxidized oxygenated OA (MO-OOA) and aerosol single scattering albedo (SSA). This paper adds new and valuable measurements of aerosol compositions and concentrations from the Tibetan Plateau, one of the less studied key regions. The analysis and the results of this paper are generally sound. The paper is within the scope of ACP and generally well written. I recommend publication of this paper in ACP after revisions. I only have some minor points for the author to consider in the revision.

We thank the reviewer for his/her careful review of the manuscript.

**Specific comments:**
(1) Line 36, the highest altitude of the TPH should be over 8800 m asl but the average altitude is about 4000 m asl.

We agree the reviewer's suggestion and have correction this sentence as follows.

"The Tibetan Plateau and Himalayas (TPH), generally called the "third pole", is the highest (average altitude of more than 4000 m a.s.l.) and largest ($\sim 2\,500\,000$ km$^2$) plateau in the world."

(2) Line 43, change "focused on" to "paid to".

Corrected.

"...more attentions have been paid to the environment and climate change in the TPH..."

(3) Line 70, change "heavily" to "mainly" or "exclusively".

Thanks for the suggestion. We have changed "heavily" to "mainly" in the revised version.

"...most of the available studies are mainly based on the off-line filter sampling..."

(4) Line 185, change "The detailed analysis were" to "Detailed analysis was".

We have corrected it as follows in the revised version.

"...Detailed PMF analysis was thoroughly evaluated following..."

(5) Line 213, I think "starting" should be changed to "ending".

We have changed "starting" to "ending" in the revised manuscript as follows.

"The back trajectories were calculated every 6 h at an ending height of 500 m..."

(6) Line 223, deleting "potential".

According to the reviewer's suggestion, we have removed it in the revised version.

(7) Line 233, brackets are needed for the sigma value.

Agree. We have added brackets for the sigma value as follows. "...the averaged diurnal air temperature ranged from −2.0 to 12.5 °C with an average ($\pm 1\sigma$) of 5.7 ($\pm 5.0$) °C..."

(8) Line 235, change "southwest" to "southwesterly".

Corrected.

"The WDs at QOMS were predominantly by southwesterly..."

(9) Line 265, a site cannot be both in the southeastern edge and in the central of a region.

This sentence is used to state both the locations of Mt. Yulong and Nam Co station. We have made it more clear in the in the revised version.

"..., except slightly lower than that at Mt. Yulong (5.7 $\mu g\ m^{-3}$) located at the southeastern edge of the TPH, whereas higher than that at Nam Co Station (2.0 $\mu g\ m^{-3}$) located in the central of the TPH."

(10) Lines 292-293 and elsewhere, "predicted" means something different. I think you calculated NH4 and compared it with the observed one. This sentence should be reworded and NH4+_pred should be changed to NH4+_calc.

Thanks a lot for the reviewer's suggestion. We have reworded this paragraph and changed $NH_{4\ pred}^{+}$ to $NH_{4\ calc}^{+}$. The specific correction in the revised version is following.

"To evaluate the bulk acidity of NR-PM$_1$ in this study, we calculated the $NH_4^{+}$ concentration ($NH_{4\ calc}^{+}$) based on the mass concentrations of sulfate, nitrate and chloride measured by the HR-ToF-AMS and assumed full neutralization of these anions by ammonium (Zhang et al., 2007b).

The scatter plot of the measured $NH_4^+$ ($NH_{4\,meas}^+$) concentration versus the $NH_{4\,calc}^+$ concentration for the entire campaign was shown in Fig. S7. A tight correlation ($R^2 = 0.97$) existed between $NH_{4\,meas}^+$ and $NH_{4\,calc}^+$ with a linear regression slope of 1.2, indicating that there were excess of ammonium in the submicron particle. This slightly high $NH_{4\,meas}^+/NH_{4\,calc}^+$ ratio was quite different with those results from various urban and rural sites in China, where bulk aerosols were overall neutralized or acidic due to the enrich gaseous precursors of $SO_2$ and $NO_x$ that could be further oxidized to sulfate and nitrate (Sun et al., 2013; Xu et al., 2014; Du et al., 2015; Zhang et al., 2017b). The excess ammonium at QOMS might relate to the important contributions of organic acids in this area (Cong et al., 2015), which could underestimate the $NH_{4\,calc}^+$ due to the neglect of organic acids in the ion-balance calculation, and the non-negligible contributions of nitrogen-containing organic compounds to $NH_x^+$ which finally overestimated the $NH_{4\,meas}^+$ (Sun et al., 2009; Ge et al., 2012). As mentioned above, atmospheric aerosols in the TPH region were significantly influenced by BB emissions from South Asia during the sampling periods. BBs would emit large amounts of nitrogen-containing organic compounds (Fleming et al., 2017; Zhou et al., 2017) and as discussed in section 3.2."

(11) Lines 549-550, "Noting the N/C ratio also displayed constantly increased trend at night, probably associated with nitrate radical oxidation". I feel this is too speculative. You are talking about aqueous-phase oxidation. You have no measurements of nitrate radical in both gaseous and liquid phase.

We agree with the reviewer's comment. In order to avoid misunderstanding, this speculation has been removed in the revised version.

(12) Fig. 9, are the fire hotspots annual averages or statistics for the measurement period?

The fire hotspots information used in this study are from the statistics of the entire measurement period (from April 12 to May 12, 2016) or each episode period (e.g., PP1, CP1), rather than the annual average values. We have added a few sentences to clarify this information in the revised version as follows.

"A four-cluster solution and the wildfire hotspots around the QOMS during the entire measurement period were presented in Fig. 9." in Sect. 3.4.1.

"...whereas the corresponding back trajectories of air masses and MODIS fire hotspots belong to each episode period were given in Fig. 11, respectively." in Sect. 3.4.2.

"**Fig. 9** ...The average PM1 mass concentrations, number of fire hotspots as well as the average and maximum fire radiative powers (FRP) belong to each clusters during the entire measurement period are also given." in Fig. 9.

(13) Fig. 12, why has the ALWC a unit "moles"?

The aerosol liquid water content (ALWC) used in this study was calculated from the online E-AIM model. The detailed introduction of this model has been added in Sect. 2.4 in the revised version as follows. The model output included the aqueous phase water content with unit of moles,

grams, molality, etc., as well as the estimated density and total volume of the aqueous phase. The model outputs with different unit are actually consistent for the temporal variation

"The aerosol liquid water content (ALWC) was estimated with the Extended AIM (E-AIM) Aerosol Thermodynamics Model (http://www.aim.env.uea.ac.uk/aim/aim.php). The input data included the concentrations of sulfate, nitrate, ammonium, and chloride measured by the HR-ToF-AMS as well as the relative humidity (RH) and temperature of ambient air." in Sect. 2.4.

(14) Fig. 13, if you intend to compare the correlations in different periods or conditions, you compare the some correlations. However, the x-axes are different for different periods. Such comparisons do not make sense.

Thanks for the reviewer's suggestion. A typical case study was investigated in this study to examine the chemical evolution of biomass burning emission aerosols undergoing various atmospheric oxidation processes. The temporal variation of aerosol chemical characteristics was given in Fig. 12 and three distinct chemical processes were identify, namely an obvious emission of fresh biomass burning plume followed by aqueous-phase oxidation in the nighttime and photochemical oxidation in the daytime. In fact, the purpose of Fig. 13 was not to simply compare the correlations in different periods, but to give a comprehensive evidence to support the different processes. We finally select nine parameters in each process. The first stage was an obvious biomass burning emission with a steep enhancement of BBOA, so we used the BBOA as the x-axes to indicate how the other chemical parameters varied with the fresh BBOA emission. We can see that the fraction of OA, BBOA, and $m/z$ 60 as well as H/C ratio were all increased whereas the parameters such as O/C, $OS_c$, etc. were decreased continuously. The second stage was an obvious aqueous-phase oxidation process which showed apparent chemical formation from fresh to aged BBOA with a steep increase of ALWC (Fig. 12), so the cumulative ALWC value was used as the x-axes to examine how aerosol characteristics changed during the nighttime oxidation process (Fig. 13). Similarly, distinct photochemical oxidation process occurred in the daytime with the increase of solar radiation. Although the $PM_1$ mass concentration decreased, the mass fraction of MO-OOA increased apparently in this period with continuous increase of $OS_c$ and SSA values. So the cumulative solar radiation was also used as the x-axes in this stage. In the two oxidation processes, the fraction of BBOA, $m/z$ 60 and H/C ratio decreased, whereas the fraction of MO-OOA, O/C, $OS_c$, SSA were all increased, suggesting the obvious chemical evolution of fresh BBOA.

(15) Figs. S5 and S6, give more details in figure captions.

We have added more information in the figure captions of Figs. S5 and S6 in the revised version as follows.

"**Fig. S5.** The 2-factor solution PMF results of **(left)** the time series of the two OA factors, **(middle)** the high-resolution mass spectra of the two OA factors colored by six ion families at m/z < 120, and **(right)** the diurnal variations of the mass concentrations of the two OA factors (the whiskers above and below the boxes indicate the 90th and 10th percentiles, the upper and lower boundaries respectively indicate the 75th and 25th percentiles, the lines in the boxes indicate the median values, and the cross symbols indicate the mean values)."

"**Fig. S6.** The 4-factor solution PMF results of **(left)** the time series of the four OA factors, **(middle)** the high-resolution mass spectra of the four OA factors colored by six ion families at m/z < 120, and **(right)** the diurnal variations of the mass concentrations of the four OA factors (the whiskers above and below the boxes indicate the 90th and 10th percentiles, the upper and lower boundaries respectively indicate the 75th and 25th percentiles, the lines in the boxes indicate the median values, and the cross symbols indicate the mean values)."

(16) Fig. S12, you are not using discrete colors, so the color bar is not appropriate.

The figure S12 in the supplement have been removed in the revised version according to your next comment.

(17) This paper includes totally 26 figures and many figures contain several plots (partly too tiny to be easily read). When reading the paper, I felt sometimes lost jumping between the text and the cited figures. And I think many figures are not discussed to certain degree. I think the authors should show only figures that are really necessary and discussed them in detail. The presentation of the figures and their order should be improved.

Thanks a lot for the reviewer's insightful suggestion. We have re-arranged the figures in both the manuscript and supplement. Finally, a total of 22 figures are included in the revised version and 4 figures are removed. In addition, we have modified some figures according to the three reviewers' suggestion and make them more clear to be read.

**Response to reviewer#3**

In this study, a filed study was performed from April 12 to May 12, 2016 at a high-altitude site (QOMS) in the northern Himalayas using an Aerodyne high-resolution time-of-flight aerosol mass spectrometer (HR-ToF-AMS) along with other collocated instruments, with the target to characterize the chemical composition, sources, and transport mechanism of polluted biomass burning aerosols from South Asia to the Himalayas during pre-monsoon season. As highly-time resolved aerosol measurement in such high altitude regions are very rare and important, the data set provided by this work is thus very valuable. The authors also performed a comprehensive analysis on this dataset, and the findings, conclusions are well supported by such analyses. Overall, the manuscript is also well written and documented. The topic also fits well in the scope of ACP. I thus recommend this manuscript can be published after some revisions.

Thank you very much for your careful review of the manuscript.

**General comments:**

1. In Fig. 2, the summary of temporal variation of meteorological parameters and various HR-ToF-AMS data are present. Dose all parameters show consistent time solution, e.g., 5-min? Especially $f_{60}$ values, which shows unexpectedly smooth trend. Please give clear introduction.

The meteorological parameters and the HR-ToF-AMS data in Fig.2 are all showed at 5-min time solutions except the $f_{60}$ values which used the hourly average results. Due to the $f_{60}$ values were

calculated from the ratios of mass concentrations of $C_2H_4O_2^+$ to that of total organics, the temporal variation at 5-min resolution showed much noise, so the hourly average were used here. We have added this information in the caption of figure 2 as follows.

"**Figure 2.** Summary of meteorological and HR-ToF-AMS data. The 5-min time series of **(a)** ambient temperature ($T$), relative humidity (RH), and solar radiation, **(b)** wind speed (WS) colored by wind direction (WD), **(c)** mass concentrations of $PM_1$ species, **(d)** mass concentrations of organic components, **(e)** mass contributions of $PM_1$ species to total $PM_1$ as well as total $PM_1$ mass concentrations, and **(f)** mass contributions of organic components to organics. The time series of hourly average $f_{60}$ (= $C_2H_4O_2^+$ / OA) values for the entire period is also showed. The markers of PP1 and PP2 represent the two polluted periods while CP1 and CP2 are clear periods, respectively."

2. The average mass contributions of $PM_1$ species during the entire sampling period and as a function of the total $PM_1$ mass concentrations was analyzed in section 3.1.3 and Fig. 3. Organic aerosol and black carbon are the two dominant $PM_1$ species in this study. In addition, the author suggested that organic compounds and BC have been revealed as two dominant components of BB aerosols. However, with the continuously enhancement of pollution mainly due to the biomass burning emissions, why the contribution of BC does not increase as that of organics?

Organic aerosol and black carbon were the two dominant $PM_1$ species in this study which averagely contributed 54.3% and 25.0% to the total $PM_1$ mass, respectively, due to the significant impacts of biomass burning emissions. In fact, with the continuously enhancement of the pollution associated with biomass burning emissions, the mass concentrations of black carbon increased apparently in this study as that of organics. This trend could be clearly seen from the PP1 and PP2 periods in figure 2 and 10, respectively. The mass contributions displayed in figure 3 were the relative fractions of each species to total $PM_1$ rather than the absolute mass concentrations. During the long-range transport of pollutants from South Asia to high-altitude Tibetan Plateau and Himalayas areas, an amount of volatile and semi-volatile organic vapors condensed rapidly to particle phase due to the lower air temperature in the high elevation area, so the enhancement of organic mass was much faster than that of BC, which finally led to a weak increase of BC mass fraction.

3. In this study, the diurnal variations of all $PM_1$ species showed similar patterns that related with the dynamics of planetary boundary layer (PBL). Is there any measurement of PBL height at QOMS?

We have no measurement of PBL height at QOMS in this field study. However, we could infer the dynamical variation of PBL height indirectly from other meteorological parameters, such as the diurnal variations of ambient temperature, relative humidity as well as solar radiation, of which maximum or minimum periods in the afternoon were quite consistent with the period of low mass concentrations of $PM_1$ species in figure 5.

4. In the section 3.4.2 or Fig. 9 and 11, the author give the fire hotspots statistics, e.g., number, average and max FRP for each cluster or periods, how did the author calculated? The total statistics for the whole areas (5-55 °N, 40-135 °E) that you selected or other approach?

The fire hotspots statistics, including hotspots number, the average and maximum fire radiative powers (FRP), used in this study were all calculated from each cluster during either the entire period (figure 9) or different episode periods (figure 11), rather than the total statistics of the whole areas. Namely, for a certain 72-h back trajectory cluster, a total of 73 points with different latitude and longitude were achieved from the HYSPLIT4 mode. Then the fire hotspots which were located around these location points (within $\pm 0.1°$ for both latitude and longitude) were selected for the calculation of fire hotspots statistics in this study. In order to make more clear, we have revised the captions of Fig. 9 and 11 as follows in the revised version.

"Figure 9. ...The average $PM_1$ mass concentrations, number of fire hotspots as well as the average and maximum fire radiative powers (FRP) belong to each clusters during the entire measurement period are also given."

"Figure 11. ...The average $PM_1$ mass concentrations, number of fire hotspots as well as the average and maximum fire radiative powers (FRP) belong to all trajectories for the different episodes are also given."

5. In the section 3.4.3, you discussed the evolution of BB emission aerosol, it will be very good if you could show behavior of organic nitrate during this period.

According to the reviewer's comment, we have added the temporal variation of calculated organic nitrate in figure 12e and given some discussion in the section 3.4.3 as follows.

"Similar continuous increase trend could also be found for the mass concentration of calculated organic nitrate in this stage"

6. The author give the identification of the two polluted periods (PP1 and PP2) and two clear periods (CP1 and CP2) in section 3.4.2, however, the initial usage of these distinct periods are in Fig. 2 and section 3.3.1, thus make reader more confuse.

We have removed the identification of the two polluted periods and clear periods from the section 3.4.2 to 3.1.3 in the revised version; the specific changes can be seen as following.

"As shown in Fig. 2, the mass concentrations of $PM_1$ and all $PM_1$ species, as well as their mass fractions in $PM_1$ varied dynamically throughout this study. Two polluted periods (PP1 and PP2) were identified according to their high $PM_1$ mass concentrations (daily average $PM_1$ mass is larger than 5 μg m$^{-3}$), high contributions from BBOA and unique back trajectories. The rest periods characterized by low $PM_1$ mass concentrations were considered as clear periods (CP1 and CP2)." in section 3.1.3.

7. Change "transportation" to "transport" in the whole manuscript.

We have corrected all "transportation" to "transport" in the revised version according to the reviewer's suggestion.

8. Line 43, "...on the environment and climate change in the TPH region since they are very susceptibility to the global climate" changed to "...on the environment and climate change in the TPH since this region is very susceptibility to the global climate".

Agree. We have rewritten this sentence as follows.

"Over the past decades, more attentions have been paid to the environment and climate change in the TPH since this region is very susceptibility to the global climate change..."

9. Line 49, "had frequently been observed " changed to "had been observed frequently"

Corrected.

"For example, polluted air mass, particularly from South and Southeast Asia regions, had been observed frequently to transport to the Himalayas..."

10. Line 64, "south Asia" changed to "South Asia", and similarly anywhere else.

We have corrected the "south Asia" or "south and southeast Asia" to "South Asia" or "South and Southeast Asia" in the revised version according to reviewer's comment.

11. Line 213, 387, and 500, "QOMS" was the abbreviation of "Qomolangma Station ", so change "QOMS site" to just "QOMS".

Agree. The expressions of "QOMS site" were all changed to "QOMS".

12. A recent study in the TP region can be included in Line 84 of the introduction part (Wang et al., Environ Sci Technol 2017, 51, (24), 14072-14082). In addition, a few new studies and reviews that summarizes the application of advanced AMS techniques can be added in Line 77-83 (Li et al., Atmos. Environ. 2017, 158, 270-304; Wang et al., Environ. Sci. Technol. Lett. 2016, 3, (4), 121-126; Wang et al., Atmos. Chem. Phys. 2016, 16, (14), 9109-9127.)

We have added these studies to the revised manuscript and the detailed correction can be seen as following.

[revised manuscript text omitted]